# Counterfactual Image Editing
# with Disentangled Causal Latent Space

**Yushu Pan** and **Elias Bareinboim**

Causal Artificial Intelligence Lab
Columbia University
{yushupan, eb}@cs.columbia.edu

## Abstract

The process of editing an image can be naturally modeled as evaluating a counterfactual query: "What would an image look like if a particular feature had changed?" While recent advances in text-guided image editing leverage powerful pre-trained models to produce visually appealing images, they often lack counterfactual consistency – ignoring how features are causally related and how changing one may affect others. In contrast, existing causal-based editing approaches offer solid theoretical foundations and perform well in specific settings, but remain limited in scalability and often rely on labeled data. In this work, we aim to bridge the gap between causal editing and large-scale text-to-image generation through two main contributions. First, we introduce Backdoor Disentangled Causal Latent Space (BD-CLS), a new class of latent spaces that allows for the encoding of causal inductive biases. One desirable property of this latent space is that, even under weak supervision, it can be shown to exhibit counterfactual consistency. Second, and building on this result, we develop BD-CLS-Edit, an algorithm capable of learning a BD-CLS from a (non-causal) pre-trained Stable Diffusion model. This enables counterfactual image editing without retraining. Our method ensures that edits respect the causal relationships among features, even when some features are unlabeled or unprompted and the original latent space is oblivious to the environment's underlying cause-and-effect relationships.

## 1 Introduction

Image editing is an important task in computer vision, which enables a counterfactual question: "What would a given image be had a feature $X$ changed from $x$ to $x'$?" Addressing such questions benefits generative models by realism, interpretability, fairness, generalizability, and transportability [41, 6, 42, 5, 51, 63, 23, 31]. Earlier approaches to solving this problem typically consider inverting images into a *Latent Space (LS)* and manipulating latent vectors leveraging correlations between the labels of intervened features and images [53, 17, 26, 8]. Recently, text-guided image editing methods leverage large-scale pre-trained models, such as CLIP, Stable Diffusion, and Rectified Flows [45, 48, 15], to enable edits that align with general human common sense concepts, without requiring model retraining [12, 18, 19, 7, 33, 28, 49]. These methods prioritize keeping the edited image as close as possible to the original and maintaining non-edited features unchanged (called as *semantic invariance*), but are often oblivious to the effects of the edited features on other features.

More recently, causal generative models have been proposed to capture causal effects in data [59, 60, 40, 57, 58]. By integrating Structural Causal Models (SCMs) with modern deep generative architectures, these models can practically achieve high-quality causal image editing given observational image samples, corresponding feature labels, and a specified causal diagram under restricted assumptions [47, 46, 13]. From a theoretical standpoint, these methods often (point-)estimate

39th Conference on Neural Information Processing Systems (NeurIPS 2025).

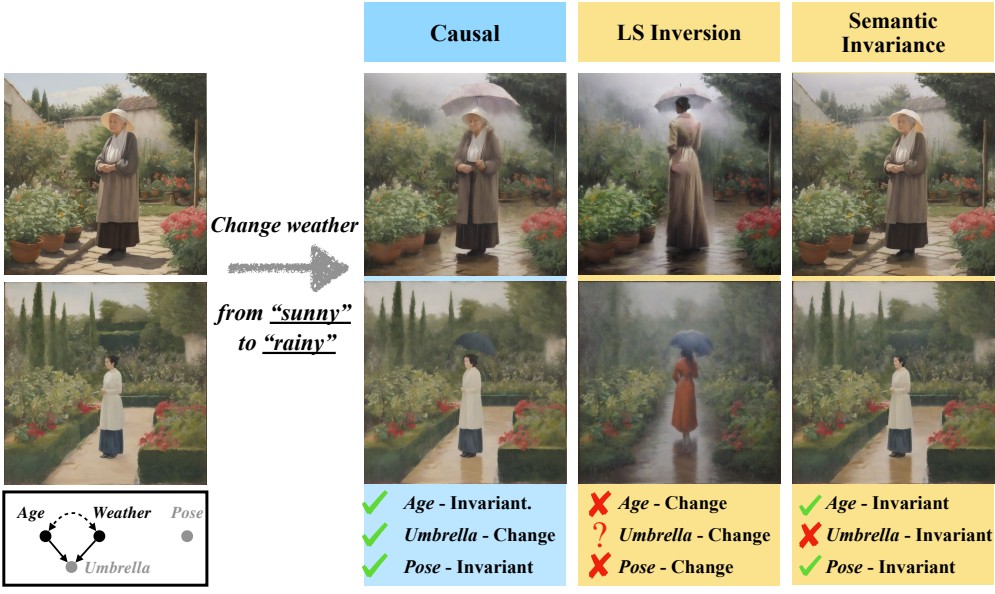

Figure 1: Editing results from Example 1: causal editings (blue); non-causal editings (yellow).

counterfactual queries under the assumption of identifiability, without formal guarantees on the validity of their outputs. [36] addresses this gap and introduces the notation of *counterfactual consistency*, a criterion that offers formal guarantees for the causal effect of editing specific features. Despite these recent advances, existing causal editing methods fall short in terms of scalability compared to large-language vision models and struggle to produce state-of-the-art image generation results in broader, more diverse real-world contexts. In addition, these methods generally require explicit annotations of features to perform counterfactual edits. However, in many real-world scenarios, obtaining these labels can be challenging due to time, cost, or feasibility constraints.

In this work, our aim is to combine the best of both worlds, having methods that are counterfactually consistent while generating high-quality images with only partial annotated data. Figure 2 summarizes our contributions along three key axes, (1) realism, (2) efficiency and scale, and (3) weak supervision. **Realism.** In contrast to current large-scale text-to-image editing methods, our method enables causal editing, which aligns more closely with the goal of realistic image manipulation by respecting underlying causal relationships among features; **Efficiency and Scale.** Unlike prior causal editing approaches, which typically require full model retraining and operate on narrowly scoped

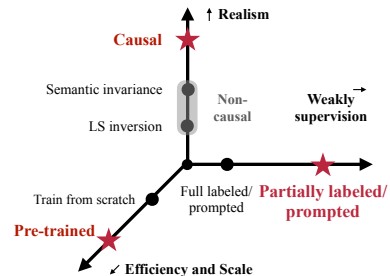

Figure 2: Three axes of contributions.

datasets, our method leverages pre-trained language-vision models to enable efficient editing without retraining. **Weak Supervision**. We address the weakly supervised setting, where only a subset of generative factors is labeled/prompted. The next example illustrates these challenges.

**Example 1.** Consider images describing "a lady is standing in a garden in a sunny day". Human common sense suggests that weather and age are not causally related; however, spurious correlations exist in the training dataset, e.g., young women appear more often in rainy scenes. Beyond weather and age, the images include other generative factors, such as the presence of an umbrella (causally influenced by both age and weather) and the pose of the lady (independent of weather, age, and umbrella use). The causal relationships are shown in the diagram Fig. 1 (bottom left). Age and weather are labeled or prompted; other features, such as a umbrella and pose, are unlabeled (gray).

Now, we consider an image editing task "change the weather from sunny to rainy". If one naively edits the initial image by LS inversion and alters the weather based purely on correlations, features like the age or pose may also change undesirably, despite the nonexistence of a causal link from weather to them. Furthermore, while such methods may raise an umbrella in the edited image, there is no causal guarantee of the probability of it being raised. On the other hand, approaches that prioritize semantic invariance aim to keep features unchanged, e.g., age and pose in this case. However, this

may also result in the umbrella never being raised. Fig. 1 (yellow) illustrates the edited images following non-casual methods. Editing images with causality, the effects of target interventions on the other features in the image are guaranteed to be carried over from the factual to the proper corresponding counterfactual world. To illustrate, the age and pose of the lady should be invariant and the effect of the rainy weather should be correctly reflected such that an umbrella is likely to be raised. See the editing results shown in the figure(blue). ∎

To enable causal image editing under weak supervision and at scale, we formalize the image generation process using an Augmented Structural Causal Model (ASCM) (Def.1) that allows for both labeled and unlabeled generative factors with proper causal semantics. We then introduce the Backdoor Disentangled Causal Latent Space (BD-CLS) (Def.4) as a modified latent space that serves as a proxy for the true generative processes. Building on ASCM and BD-CLS, our main contributions are:

1. We formally study which features should be invariant and which should change, and how these features change when causally editing images (Thm. 1). We then show that BD-CLS provides causal guarantees for both changed and invariant features, even when they are unlabeled (Thm. 2).

2. We develop **BD-CLS-Edit** (Alg. 1), a post-training algorithm that learns a BD-CLS from a pre-trained Stable Diffusion model and enables counterfactual image editing. Extensive experiments are conducted to demonstrate the effectiveness of the proposed framework.

All supplementary material (including proofs) is provided in the full technical report [37].

**Preliminaries.** An uppercase letter $X$ indicates a random variable and a lowercase letter $x$ indicates its corresponding value; bold uppercase $\mathbf{X}$ denotes a set of random variables, lowercase letter $\mathbf{x}$ is its corresponding values and $\mathcal{X}_{\mathbf{X}}$ to denote its domain. We denote $P(\mathbf{X})$ as a probability distribution over a set of random variables $\mathbf{X}$ and $P(\mathbf{x})$ as the probability of $\mathbf{X}$ being equal to the value of $\mathbf{x}$ under the distribution $P(\mathbf{X})$. Our work uses Structural Causal Models (SCM) as the underlying semantic framework [41, Ch. 7], and we follow the presentation provided in [4]. A Structure Causal Model (for short, SCM) is a 4-tuple $< \mathbf{U}, \mathbf{V}, \mathcal{F}, P(\mathbf{U}) >$, where (1) $\mathbf{U}$ is a set of background variables, also called exogenous variables, that are determined by factors outside the model; (2) $\mathbf{V} = \{V_1, V_2, \ldots, V_d\}$ is the set of endogenous variables that are determined by other variables in the model; (3) $\mathcal{F}$ is the set of functions $\{f_{V_1}, f_{V_2} \ldots, f_{V_d}\}$ mapping $\mathbf{U}_{V_j} \cup \mathbf{Pa}_{V_j}$ to $V_j$, where $\mathbf{U}_{V_j} \subseteq \mathbf{U}$ and $\mathbf{Pa}_{V_j} \subseteq \mathbf{V} \backslash V_j$; (4) $P(\mathbf{U})$ is a probability function over the domain of $\mathbf{U}$.

Each $\mathcal{M}$ induces a causal diagram $\mathcal{G}$, a directed acyclic graph (DAG) where each $V_i \in \mathbf{V}$ corresponds to a vertex. There is a directed arrow $(V_j \rightarrow V_i)$ for every $V_i \in \mathbf{V}$ and $V_j \in \mathbf{Pa}_{V_i}$, and there is a dashed-bidirected arrow $(V_j \leftarrow\!-\!-\!-\!\rightarrow V_i)$ for every pair $V_i, V_j \in \mathbf{V}$ such that $\mathbf{U}_{V_i}$ and $\mathbf{U}_{V_j}$ are not independent. We denote $\mathcal{G}_{\tilde{\mathbf{V}}}$ as the causal diagram $\mathcal{G}$ over $\mathbf{V}$ after $\mathbf{V} \backslash \tilde{\mathbf{V}}$ is marginalized out, where $\tilde{\mathbf{V}} \subseteq \mathbf{V}$. For example, for $\mathcal{G} = \{Z \rightarrow X, Z \rightarrow Y, X \rightarrow Y\}$, $\mathcal{G}_{X,Y} = \{X \leftarrow\!-\!-\!-\!\rightarrow Y, X \leftarrow Y\}$. A set of variables $\mathbf{B}$ is said to be a *backdoor set* relative to the pair $(\mathbf{X}, \mathbf{Y})$ if no node in $\mathbf{Z}$ is a descendant of $\mathbf{X}$, and $\mathbf{B}$ blocks every path between $\mathbf{X}$ and $\mathbf{Y}$ that contains an arrow into $\mathbf{X}$.

The counterfactual quantities induced by an SCM $\mathcal{M}$ are defined as [4, Def. 7]:

$$P^{\mathcal{M}}(\mathbf{y}_{1[\mathbf{x}]}, \mathbf{y}_{2[\mathbf{x}_2]}, \ldots,) = \int_{\mathcal{X}_{\mathbf{U}}} \mathbb{1}[\mathbf{Y}_{\mathbf{x}_1}(\mathbf{u}) = \mathbf{y}_1, \mathbf{Y}_{\mathbf{x}_2}(\mathbf{u}) = \mathbf{y}_2, \ldots] dP(\mathbf{u}), \quad (1)$$

where $\mathbf{Y}_{i[\mathbf{x}_i]}(\mathbf{u})$ is evaluated under $\mathcal{F}_{\mathbf{x}_i} := \{f_{V_j} : V_j \in \mathbf{V} \setminus \mathbf{X}_i\} \cup \{f_X \leftarrow x : X \in \mathbf{X}_i\}$. Each $\mathbf{Y}_i$ corresponds to a set of variables where the original mechanisms $f_X$ are replaced with constants $\mathbf{x}_i$ for each $X \in \mathbf{X}_i$. This procedure corresponds to interventions, and we use subscripts to denote the intervening variables (e.g. $\mathbf{Y}_{\mathbf{x}}$) or subscripts with brackets when the variables are indexed (e.g. $\mathbf{Y}_{1[\mathbf{x}_1]}$). We refer to App. A for more background on causal models and diffusion models.

## 2 Augmented SCMs and Causal Consistency

We begin by defining a class of SCMs that models the ground-truth image generation process, incorporating both labeled and unlabeled generative factors.

**Definition 1** (Augmented Structure Causal Model)**.** An Augmented Structure Causal Model (for short, ASCM) over a generative level SCM $\mathcal{M}_0 = \langle \{\mathbf{U}_0, \mathbf{V}_0, \mathcal{F}_0, P^0(\mathbf{U}_0)\} \rangle$ is a tuple $\mathcal{M} = \langle \mathbf{U}, \{\mathbf{V}, \mathbf{L}, \mathbf{I}\}, \mathcal{F}, P(\mathbf{U}) \rangle$ such that (1) $\mathbf{U} = \mathbf{U}_0$; (2) $\{\mathbf{V}, \mathbf{L}\}$ is a partition of all generative factors $\mathbf{V}_0$, where $\mathbf{V}$ are labeled factors; $\mathbf{L} = \mathbf{V}_0 \backslash \mathbf{V}$ are unlabeled factors; $\mathbf{I}$ is an image variable; (3) $\mathcal{F} = \{\mathcal{F}_0, f_{\mathbf{I}}\}$, where $f_{\mathbf{I}}$ maps from (the respective domains of) $\mathbf{V} \cup \mathbf{L}$ to $\mathbf{I}$, which is an invertible function. Namely, there exists a function $h$ such that $\{\mathbf{V}, \mathbf{L}\} = h(\mathbf{I})$. (4) $P(\mathbf{U}_0) = P^0(\mathbf{U}_0)$. ∎

In words, an ASCM $\mathcal{M}$ is a "larger" SCM describing a two-stage generative process of the image variable $\mathbf{I}$. First, high-level factors $\mathbf{V}_0$ are generated at the generative level $\mathcal{M}_0$. Labels (e.g., annotations, text descriptions, etc.) are given only on the part of factors $\mathbf{V} \in \mathbf{V}_0$. The remaining factors $\mathbf{L}$ are unlabeled. Second, all factors are mapped into pixel spaces to form the image. This mapping is invertible, which means that, given an image instance $\mathbf{x}$, all factors can be recognized. That is, there exists a function $h$ that maps from $\mathbf{I}$ to $\{\mathbf{V}, \mathbf{L}\}$. See further discussion in App. E.1.

Equipped with ASCMs (Def. 1), our task to edit the concepts $\mathbf{X}$ in an original image $\mathbf{i}$ from $\mathbf{X} = \mathbf{x}$ to $\mathbf{X} = \mathbf{x}'^{[1]}$ can be formalized as querying an *Image counterfactual distribution (I-ctf)* $P^*(\mathbf{I}_{x'} = \mathbf{i}' \mid \mathbf{I} = \mathbf{i})$ induced by the true underlying model $\mathcal{M}^*$. In addition, ASCMs can formalize the counterfactual effect in editing between generative factors. Formally, given factual factors $\mathbf{W}_1 = \mathbf{w}_1$ ($\mathbf{W}_1 \subseteq \mathbf{V}$), the probability that factors $\mathbf{W}_2$ will be $\mathbf{w}_2$ after the edit $do(\mathbf{X} = \mathbf{x}')$ is formalized by a counterfactual quantity $P^*(\mathbf{W}_{2[\mathbf{X}=\mathbf{x}']} = \mathbf{w}_2 \mid \mathbf{w}_1)$ at the generative level $\mathcal{M}_0$.

## 2.1 Proxy model for ground-truth ASCM

Consider an underlying ground-truth ASCM $\mathcal{M}^*$ and image $\mathbf{I}$ generated from $\mathcal{M}^*$. Generative models such as Stable Diffusion, VAEs, and GANs learn a latent space $\mathbf{Z}$ along with a mapping function $f : \mathbf{Z} \to \mathbf{I}$, which enables the generation of synthetic images $\mathbf{I}$. Formally, this synthetic generation process can be regarded as a *proxy* SCM $\widehat{\mathcal{M}}$ over variable $\mathbf{Z}, \mathbf{I}$ such that $\widehat{\mathcal{M}}$ approximates the image distribution induced by $\mathcal{M}^*$, i.e., $P^{\mathcal{M}^*}(\mathbf{I}) = P^{\widehat{\mathcal{M}}}(\mathbf{I})$. Thus, editing a given image $\mathbf{i}$ by alternating latent vectors ($do(\mathbf{T} = \mathbf{t}'), \mathbf{T} \subseteq \mathbf{Z}$) in proxy models can be modeled as a counterfactual query $P^{\widehat{\mathcal{M}}}(\mathbf{I}_{\mathbf{T}=\mathbf{t}'} \mid \mathbf{i})$ (See Ex. 3 for illustration). Then the following quantity is defined to capture the counterfactuals over features when editing images in such ways.

**Definition 2** (Feature Counterfactual Query). Consider an underlying true ASCM over generative factors $\mathbf{V}$ and $\mathbf{L}$, a proxy model $\widehat{\mathcal{M}}$ over $\{\mathbf{Z}, \mathbf{I}\}$, a set of factual features $\mathbf{W}_2 \subseteq \{\mathbf{V}, \mathbf{L}\}$, and a set of counterfactual features $\mathbf{W}_1 \subseteq \{\mathbf{V}, \mathbf{L}\}$. A feature counterfactual (F-ctf) query is defined as:

$$
P^{\widehat{\mathcal{M}}}(\mathbf{W}_{1[\mathbf{T}=\mathbf{t}']} = \mathbf{w}_1 \mid \mathbf{W}_2 = \mathbf{w}_2) := \frac{\int_{\mathbf{i},\mathbf{i}' \in \mathcal{X}_\mathbf{I}} \mathbf{1}\left[h^*_{\mathbf{W}_1}(\mathbf{i}') = \mathbf{w}_1, h^*_{\mathbf{W}_2}(\mathbf{i}) = \mathbf{w}_2\right] dP^{\widehat{\mathcal{M}}}(\mathbf{i}, \mathbf{i}'_{[\mathbf{T}=\mathbf{t}']})}{\int_{\mathbf{i} \in \mathcal{X}_\mathbf{I}} \mathbf{1}\left[h^*_{\mathbf{W}_2}(\mathbf{i}) = \mathbf{w}_2\right] dP^{\widehat{\mathcal{M}}}(\mathbf{i})}
$$

(2)

where $h^*_{\mathbf{W}_1}$ and $h^*_{\mathbf{W}_2}$ are the mappings from $\mathbf{I}$ to $\mathbf{W}_1$ and $\mathbf{W}_2$. ∎

In words, $P^{\widehat{\mathcal{M}}}(\mathbf{W}_{1[\mathbf{T}=\mathbf{t}']} = \mathbf{w}_1 \mid \mathbf{W}_2 = \mathbf{w}_2)$ is the probability that the feature $\mathbf{W}_1$ would take value $\mathbf{w}_1$ had latent vector $\mathbf{T} = \mathbf{t}'$, given that the features $\mathbf{W}_2$ are currently equal to $\mathbf{w}_2$. The denominator integrates all images $\mathbf{i}_1$ such that $\mathbf{i}_1$ has features $\mathbf{w}_1$ in factual worlds; the numerator integrates over counterfactual worlds $P(\mathbf{i}, \mathbf{i}'_{[\mathbf{T}=\mathbf{t}']})$ such that $\{\mathbf{i}, \mathbf{i}'\}$ has features $\{\mathbf{w}_1, \mathbf{w}_2\}$. Def. 2 provides a way to describe counterfactual quantities over features $\mathbf{W}_1$ and $\mathbf{W}_2$ even when $\mathbf{W}_1$ and $\mathbf{W}_2$ are not necessarily endogenous variables in the proxy model $\widehat{\mathcal{M}}$. See more discussion in Ex. 4 and App. E.3.

Next, we establish a concept to evaluate whether an F-ctf query $P^{\widehat{\mathcal{M}}}(\mathbf{w}_{1[\mathbf{t}']} \mid \mathbf{w}_2)$ induced by the proxy model constitutes a reliable approximation to a counterfactual quantity $P^*(\mathbf{w}_{1[\mathbf{x}']} \mid \mathbf{w}_2)$ induced by the ground truth model.

**Definition 3** (Ctf-consistency). Consider an ASCM $\mathcal{M}^*$ over factors $\{\mathbf{V}, \mathbf{L}\}$ and a proxy model $\widehat{\mathcal{M}}$. An F-ctf query, $P^{\widehat{\mathcal{M}}}(\mathbf{w}_{1[\mathbf{t}']} \mid \mathbf{w}_2)$, is said to be counterfactually consistent with the ground truth $P^*(\mathbf{w}_{1[\mathbf{x}']} \mid \mathbf{w}_2)$ under $\langle \mathcal{Q}, \mathcal{G} \rangle$, if $P^{\widehat{\mathcal{M}}}(\mathbf{w}_{1[\mathbf{t}']} \mid \mathbf{w}_2) \in [l, r]$, where $[l, r]$ is the union of optimal bounds of $P^*(\mathbf{w}_{1[\mathbf{x}']} \mid \mathbf{w}_2)$ given the observational quantities $\mathcal{Q} = \{P^*(\mathbf{v}, \mathbf{l})\}_{\mathbf{v} \in \mathcal{X}'_\mathbf{V}, \mathbf{l} \in \mathcal{X}'_\mathbf{L}}$ and the causal diagram $\mathcal{G}_\mathbf{R}$, where $\mathcal{X}'_\mathbf{V} \subseteq \mathcal{X}_\mathbf{V}$, $\mathcal{X}'_\mathbf{L} \subseteq \mathcal{X}_\mathbf{L}$, and $\mathbf{R} \subseteq \{\mathbf{V}, \mathbf{L}, \mathbf{I}\}$. ∎

This definition offers a principled way to evaluate the estimate produced by a proxy model against the ground truth counterfactual quantity. It extends the formulation of [36, Def. 4.4]. This is needed since given the observational distribution and the causal diagram, the target counterfactual query is not always uniquely computable but some possibly informative bound Def. 10 can be obtained and serve as a natural measure of distance from the data and the true, yet unobserved, counterfactual

---

[1]We assume $\mathbf{x} \neq \mathbf{x}'$ throughout this work, implying that the concepts indeed change.

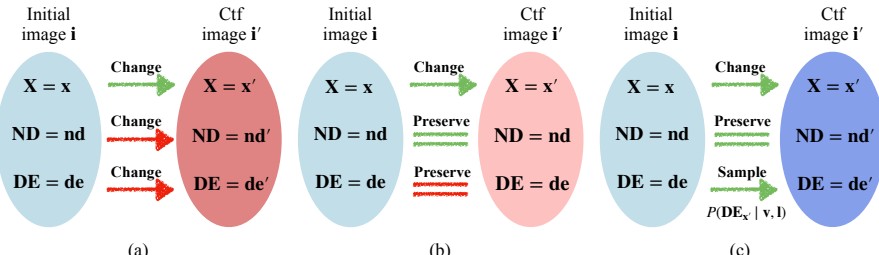

Figure 3: Invariance relationships between features in original image and counterfactual images (images after editing) cross noncausal methods (a-b) and the newly proposed causal method (c).

distribution. Def. 3 says that any value that is out of this bound is regarded as invalid estimations, and any value within this bound is acceptable. See Ex. 5 for an illustration.

# 3 Causal Estimator for Image Editing

In this section, we present our main theoretical results. We first factorize the target I-ctf query $P^{\mathcal{M}^*}(\mathbf{i}'_{\mathbf{x}'} \mid \mathbf{i})$, identifying which generative factors should change or remain invariant (Sec.3.1) across the factual and counterfactual worlds (Sec. 3.1). Then we introduce Backdoor Disentangled Causal Latent Space, and establish the causal guarantees it provides for evaluating I-ctf queries (Sec. 3.2).

## 3.1 Factorization of I-ctf query

We begin by expressing the I-ctf query in terms of generative factors and factorizing it based on their descendant relationships.

**Theorem 1.** *Consider the true underlying ASCM $\mathcal{M}^*$ over $\{\mathbf{V}, \mathbf{I}\}$. Let $\mathbf{ND}$ denote $\cap_{X_i} \mathbf{ND}(X_i) \backslash \mathbf{X}$ (non-descendants of $\mathbf{X}$) in $\mathcal{G}_{\mathbf{V}, \mathbf{L}}$ and $\mathbf{DE}$ denote $\cup_{X_i} \mathbf{DE}(X_i) \backslash \mathbf{X}$ (descendants of $\mathbf{X}$) in $\mathcal{G}_{\mathbf{V}, \mathbf{L}}$. The target I-ctf query $P^*(\mathbf{I}_{\mathbf{x}'} = \mathbf{i}' \mid \mathbf{I} = \mathbf{i})$ can be factorized as*

$$P^*(\mathbf{I}_{\mathbf{x}'} = \mathbf{i}' \mid \mathbf{I} = \mathbf{i}) = \underbrace{\mathbf{1}[h_{\mathbf{X}}^*(\mathbf{i}') = \mathbf{x}']}_{\textit{Intervention Consistency}} \cdot \underbrace{\mathbf{1}[\mathbf{nd}' = \mathbf{nd}]}_{\textit{Non-descendants Invariance}} \cdot \underbrace{P^*(\mathbf{De}_{\mathbf{x}'} = \mathbf{de}' \mid \mathbf{v}, \mathbf{l})}_{\textit{Amount of Descendant Changing}} \quad (3)$$

*where $\mathbf{nd} = h_{\mathbf{ND}}^*(\mathbf{i})$, $\mathbf{nd}' = h_{\mathbf{ND}}^*(\mathbf{i}')$, $\mathbf{de} = h_{\mathbf{DE}}^*(\mathbf{i})$, and $\{\mathbf{v}, \mathbf{l}\} = h_{\mathbf{V}, \mathbf{L}}^*(\mathbf{i})$.* ∎

This result circumscribes which features should remain invariant and which should change through the editing process. Specifically, the first term in the r.h.s. of Eq. 3 corresponds to the notation of **interventional consistency** - the edit should effectively change the features $\mathbf{X}$ in the counterfactual image such that these features are equal to $\mathbf{x}'$; the second term corresponds to **non-descendants invariance** - the non-descendant features $\mathbf{ND}$ must remain invariant across factual and counterfactual images; the third term corresponds to **descendant delta** - the descendants of $\mathbf{X}$ are possibly affected and the probability of changes should be consistent with the counterfactual distribution $P^*(\mathbf{De}_{\mathbf{x}'} \mid \mathbf{v}, \mathbf{l})$. These feature invariance/variance between pre- and post-intervention worlds are shown in Fig. 3(c) (see also Ex. 6 for further grounding).

Using this result, we identify key limitations in the current evaluation of image editing methods. A common approach, LS inversion, edits images by inverting images into an LS and sampling from $P(\mathbf{I} \mid \mathbf{x}')$, enforcing the target feature value $x'$. While this ensures interventional consistency, it often violates non-descendant invariance and descendant changes. Specifically, editing $\mathbf{X}$ can lead to unintended consequences in correlated non-descendants $\mathbf{ND}$. Furthermore, the amount of change in descendant features $\mathbf{DE}$ change lacks a proper causal guarantee, e.g., counterfactual consistency (see Fig. 3(a) and Ex.7). Modern text-to-image editing methods typically pursue: (1) editing effectiveness: removing original features $\mathbf{x}$ and incorporating the target $\mathbf{x}'$, and (2) semantic invariance: preserving other content. These correspond to the first two terms in Eq.3. However, the third term, the descendant change, is often violated. As illustrated in Fig. 3(b), the $\mathbf{De}$ are forced to be unchanged, while it should follow counterfactual distribution $P^*(\mathbf{DE}_{\mathbf{x}'} \mid \mathbf{v}, \mathbf{l})$. See more discussion in App. E.4.

## 3.2 Backdoor Disentangled Causal Latent Space

In this section, we develop a class of generative models that ensures editing behavior consistent with both non-descendant and descendant requirements, as depicted by Thm. 1.

**Definition 4** (Backdoor Disentangled Causal Latent Space). Consider a true ASCM $\mathcal{M}^*$ with diagram $\mathcal{G}_{\mathbf{V},\mathbf{I}}$, and a target I-ctf distribution query $P(\mathbf{I}_{\mathbf{x}'} \mid \mathbf{i})$. Let $\mathbf{B} \subseteq \mathbf{V}$ be a backdoor set w.r.t. $\mathbf{X}$ to $\mathbf{I}$ in $\mathcal{G}_{\mathbf{V},\mathbf{I}}$. Denote the mapping from $\mathbf{I}$ to $\mathbf{ND}$ as $h_{\mathbf{ND}}^*$. A Backdoor Disentangled Causal Latent Space (BD-CLS) is an SCM $\widehat{\mathcal{M}}^{\text{BD-CLS}} = \langle \widehat{\mathbf{U}}, \mathbf{V} = \{\mathbf{X}, \mathbf{B}, \mathbf{Z}, \mathbf{I}\}, \widehat{\mathcal{F}}, P(\widehat{\mathbf{U}}) \rangle$, such that $\mathbf{I} \leftarrow \widehat{f}_{\mathbf{I}}(\mathbf{X}, \mathbf{B}, \mathbf{Z})$, and (1) (generation) $P^{\widehat{\mathcal{M}}^{\text{BD-CLS}}}(\mathbf{I} \mid \mathbf{X}, \mathbf{B}) = P^{\widehat{\mathcal{M}}^*}(\mathbf{I} \mid \mathbf{X}, \mathbf{B})$; (2) (disentanglement) $\partial \tau_{\mathbf{ND}}/\partial \mathbf{X} = 0$, where $\tau_{\mathbf{ND}} = h_{\mathbf{ND}}^* \circ \widehat{f}_{\mathbf{I}}$; (3) (structure) $\widehat{\mathcal{G}}$ is compatible as Fig. 4 [2]. ∎

In words, BD-CLS regarding to an editing task (I-ctf query $P(\mathbf{I}_{\mathbf{x}'} \mid \mathbf{i})$) is a proxy $\widehat{\mathcal{M}}^{\text{BD-CLS}}$ with endogenous variables $\{\mathbf{X}, \mathbf{B}, \mathbf{Z}\}$, where $\mathbf{B}$ is a backdoor set for $\mathbf{X}$ to image $\mathbf{I}$ in $\mathcal{G}_{\mathbf{V},\mathbf{I}}$. In addition, this proxy model should satisfy three requirements. First, $\widehat{\mathcal{M}}^{\text{BD-CLS}}$ induces the same $P(\mathbf{I} \mid \mathbf{X}, \mathbf{B})$ as ground truth $\mathcal{M}^*$. Second, BD-CLS $\widehat{\mathcal{M}}^{\text{BD-CLS}}$ requires $\tau_{\mathbf{ND}}$ be *disentangled* to $\mathbf{X}$, which means changing the value of $\mathbf{X}$ will not change the value of $\tau_{\mathbf{ND}}(\cdot, \mathbf{B}, \mathbf{Z})$, where

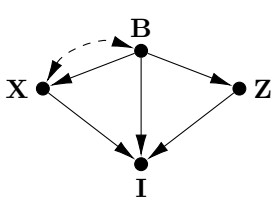

Figure 4: $\widehat{\mathcal{G}}$ in Def. 4.

$$\mathbf{ND} = \tau_{\mathbf{ND}}(\mathbf{X}, \mathbf{B}, \mathbf{Z}) = h_{\mathbf{ND}}^* \circ \widehat{f}_{\mathbf{I}}(\mathbf{X}, \mathbf{B}, \mathbf{Z}) \quad (4)$$

Third, $\widehat{\mathcal{G}}$ induced by $\widehat{\mathcal{M}}^{\text{BD-CLS}}$ should be compatible with Fig. 4. It is worth noting that $\mathbf{X}$ are independent of $\mathbf{Z}$ given $\mathbf{B}$, which implies that $\mathbf{B}$ inherits its backdoor property in $\mathcal{G}_{\mathbf{V},\mathbf{I}}$. See Ex. 8 for more details. The next result discusses the validity of generating samples from a BD-CLS.

**Theorem 2** (Causal validity of BD-CLS). *Consider an $\widehat{\mathcal{M}}^{BD\text{-}CLS}$ for $\mathcal{M}^*$ and the target query $P^*(\mathbf{i}'_{\mathbf{x}'} \mid \mathbf{i})$. Let $P^{\widehat{\mathcal{M}}^{BD\text{-}CLS}}(\mathbf{i}'_{\mathbf{x}'} \mid \mathbf{i})$ be an estimator for $P(\mathbf{i}'_{\mathbf{x}'} \mid \mathbf{i})$. Then, (a) (intervention) $P^{\widehat{\mathcal{M}}^{BD\text{-}CLS}}(\tilde{\mathbf{x}}_{\mathbf{x}'} \mid \mathbf{v}, \mathbf{l}) = \mathbf{1}[\tilde{\mathbf{x}} = \mathbf{x}']$, where $\tilde{\mathbf{x}} = h_{\mathbf{X}}(\mathbf{i}')$; (b) (non-descendants) $P^{\widehat{\mathcal{M}}^{BD\text{-}CLS}}(\mathbf{nd}'_{\mathbf{x}'} \mid \mathbf{v}, \mathbf{l}) = \mathbf{1}[\mathbf{nd}' = \mathbf{nd}]$, (c) (descendants) $P^{\widehat{\mathcal{M}}^{BD\text{-}CLS}}(\mathbf{de}'_{\mathbf{x}'} \mid \mathbf{v}, \mathbf{l})$ is ctf-consistent with $P^*(\mathbf{de}'_{\mathbf{x}'} \mid \mathbf{v}, \mathbf{l})$ under $\langle P(\mathbf{X}, \mathbf{nd}, \mathbf{De}), \mathcal{G}_{\mathbf{X},\mathbf{ND},\mathbf{I}} \rangle$, where $\mathbf{w} = h_{\mathbf{w}}(\mathbf{i}), \mathbf{w}' = h_{\mathbf{w}}(\mathbf{i}')$ for any $\mathbf{W} \subseteq \mathbf{V} \cup \mathbf{L}$.* ∎

In words, (a) implies that the query $P^{\widehat{\mathcal{M}}^{\text{BD-CLS}}}(\mathbf{i}'_{\mathbf{X}=\mathbf{x}'} \mid \mathbf{i})$ first achieves intervention consistency. The value $\tilde{\mathbf{x}}$ of feature $\mathbf{X}$ is exactly equal to the intervened value $\mathbf{x}'$. (b) implies that the estimation $P^{\widehat{\mathcal{M}}^{\text{BD-CLS}}}(\mathbf{i}'_{\mathbf{X}=\mathbf{x}'} \mid \mathbf{i})$ satisfies non-descendant invariance. To illustrate, the feature value $\mathbf{nd}'$ of the counterfactual image will be the same as the feature value $\mathbf{nd}$ of the initial image. (c) says that a BD-CLS can guarantee the descendant delta. To illustrate, $P^{\widehat{\mathcal{M}}^{\text{BD-CLS}}}(\mathbf{de}'_{\mathbf{x}'} \mid \mathbf{v}, \mathbf{l})$ is ensured to be within the bound of $P^*(\mathbf{de}'_{\mathbf{x}'} \mid \mathbf{v}, \mathbf{l})$. Thm. 2 is powerful since BD-CLS can achieve the causal editing principles of Thm. 1 - it precisely captures the first two principles and softly satisfies the third by providing an in-bound estimation. Thus, BD-CLS enables causal image editing, improving upon prior non-causal methods. Furthermore, BD-CLS necessitates supervision only for $\mathbf{X}$ and $\mathbf{B}$, whereas $\mathbf{ND}$ and $\mathbf{DE}$ can remain unlabeled. See Ex. 9 for illustration.

## 4 Learning Backdoor Disentangled Casual Latent Space

Now, we show how to obtain a BD-CLS from a pre-trained text-to-image diffusion model for sampling counterfactual images. Given a target distribution $P^*(\mathbf{I}_{\mathbf{x}'} \mid \mathbf{i})$ induced by the true model $\mathcal{M}^*$, the goal is to generate $\mathbf{i}' \sim P^{\widehat{\mathcal{M}}^{\text{BD-CLS}}}(\mathbf{I}_{\mathbf{x}'} \mid \mathbf{i})$ induced from a BD-CLS $\widehat{\mathcal{M}}^{\text{BD-CLS}}$ using initial image's label $\mathbf{v}$ (text prompt), the diagram $\mathcal{G}_{\mathbf{V},\mathbf{I}}$, and a pre-trained Stable Diffusion (SD) model.

SD models are capable of sampling images from $P(\mathbf{I} \mid \mathbf{c})$ with classifier-freence [21], where $\mathbf{c}$ is a given prompt/label [3]. The (reverse) generation process (Fig. S1(a)) starts from a noise vector $\mathbf{I}^{(T)} \sim \mathcal{N}(0, \mathbf{1})$ and iteratively denoises it to produce a clean image $\mathbf{I}$ with recursion $\mathbf{I}^{(t-1)} = \widehat{\mu}(\mathbf{I}^{(t)}, \mathbf{c}, t) + \sigma_t \mathbf{Z}^{(t)}$ where $\mathbf{I}^{(0)} = \mathbf{I}$, $\widehat{\mu}$ is the mean predictor; $\mathbf{Z}^{(t)}$ are gaussian random vectors, and $\sigma_t$ are variance terms. This generation process can be regarded as a proxy model $\mathcal{M}^{\text{SD}}$ over $\{\mathbf{C}, \mathbf{N} = \{\mathbf{I}^{(T)}, \mathbf{Z}^T, \dots, \mathbf{Z}^{(1)}\}\}$ and $\mathbf{I} \leftarrow f^{\text{SD}}(\mathbf{C}, \mathbf{N})$ (Fig. S1(b)). Setting the prompt $\mathbf{C}$ as the observed variable $\{\mathbf{X}, \mathbf{B}\}$, an SD model will directly satisfy condition (1) of a BD-CLS (Def. 4),

---

[2]The "compatible" here does not exactly means $\widehat{\mathcal{M}}$ induce the same graph in Fig. 4. There can be less edge in $\widehat{\mathcal{G}}$ than in Fig. 4 but there cannot be more edges. The definition of "compatible" is formally defined by Def. 9.

[3]More details about diffusion models can be found in App. A.

$P^{\widehat{\mathcal{M}}^{\text{SD}}}(\mathbf{I} \mid \mathbf{C}) = P^{\widehat{\mathcal{M}}^*}(\mathbf{I} \mid \mathbf{X}, \mathbf{B})$. Then an SD model can constitute a BD-CLS if condition (2-3), i.e., disentanglement and structure, are satisfied. The following proposition formally grounds this.

**Proposition 1** (**Sampling I-ctf instances through SD model**). *Consider a ground-truth ASCM $\mathcal{M}^*$ over $\{\mathbf{V}, \mathbf{I}\}$, the target $P^*(\mathbf{I}_{\mathbf{x}'} \mid \mathbf{i})$ and a pre-trained SD model $\widehat{\mathcal{M}}^{\text{SD}}$. Suppose there exists a pair of transformations satisfying $\mathbf{Z} = \psi_1(\mathbf{X}, \mathbf{B}, \mathbf{N})$ and $\mathbf{N} = \psi_2(\mathbf{X}, \mathbf{B}, \mathbf{Z})$. Denotes $\psi(\mathbf{x}, \mathbf{x}', \mathbf{b}, \mathbf{n}) = f^{\text{SD}}(\mathbf{x}', \mathbf{b}, \psi_2(\mathbf{x}', \mathbf{b}, \psi_1(\mathbf{x}, \mathbf{b}, \mathbf{n})))$. Then there exists a BD-CLS $\mathcal{M}^{\text{BD-CLS}}$ over $\{\mathbf{X}, \mathbf{B}, \mathbf{Z}\}$ such that the causal validity in Thm. 2 are also offered by $\sum_{\mathbf{n}} P^{\widehat{\mathcal{M}}^{\text{SD}}}(\mathbf{n} \mid \mathbf{i}, \mathbf{x}, \mathbf{b}) \mathbf{1}[\mathbf{i}' = \psi(\mathbf{x}, \mathbf{x}', \mathbf{b}, \mathbf{n})]$, if*

$$\psi(\mathbf{x}, \mathbf{x}', b, \mathbf{N}) \sim P(\mathbf{I} \mid \mathbf{x}', b), \quad and \quad h^*_{\mathbf{ND}}(\psi(\mathbf{x}, \mathbf{x}', b, \mathbf{n})) = h^*_{\mathbf{ND}}(\mathbf{i}). \qquad \blacksquare \qquad (5)$$

To illustrate, Prop. 1 says that sampling a counterfactual image $\mathbf{i}'$ from a BD-CLS involves two steps using the SD model after setting the prompt variable as $\{\mathbf{X}, \mathbf{B}\}$. First, invert $\mathbf{i}$ to sample noise $\mathbf{n} \sim P^{\widehat{\mathcal{M}}^{\text{BD-CLS}}}(\mathbf{n} \mid \mathbf{i}, \mathbf{x}, \mathbf{b})$ Then the sampled $\mathbf{n}$, the initial prompt $\{\mathbf{x}, \mathbf{b}\}$, and the target prompt $\{\mathbf{x}', \mathbf{b}\}$ are fed into the compose transformation $\psi$ to generate $\mathbf{i}'$. This $\psi$ should satisfy two constraints presented in Eq. 5. The first ensures that $\psi$ transforms Gaussian noise $\mathbf{N}$ into $\mathbf{I}$ following $P(\mathbf{I} \mid \mathbf{x}', b)$. This constraint implicitly encodes the structural relationship between $\mathbf{X}$ and $\mathbf{B}$, since $b$ remains unchanged in the counterfactual world, indicating that $\mathbf{X}$ has no effect on $\mathbf{B}$. Moreover, because $\mathbf{N}$ is independent of $\mathbf{C}$ in the SD model, $\mathbf{Z}$ is not bidirectionally connected to $\mathbf{X}, \mathbf{B}$. The second constraint enforces that the non-descendants $\mathbf{ND}$ remain unchanged, reflecting the disentanglement and ensuring that $\mathbf{Z}$ is not a descendant of $\mathbf{X}$.

Building on this, we develop **BD-CLS-Edit** (Alg. 1) to learn $\psi$ and generate $\mathbf{i}'$ simultaneously. The algorithm begins by identifying the largest backdoor set $\mathbf{B}$ from $\mathbf{X}$ to $\mathbf{I}$ in $\mathcal{G}_{\mathbf{V}, \mathbf{I}}$, then matches the prompt variables $\mathbf{C}$ with the observed values of $\mathbf{X} \cup \mathbf{B}$ from the label $\mathbf{v}$. Next, it samples noise $\mathbf{n}$ given the SD model. In this work, we follow the DDPM inversion [22] and details are provided in Appendix B. In the fourth step, we search the transformation $\psi_{\boldsymbol{\theta}}$ in a candidate space to ensure that counterfactual samples

---

**Algorithm 1**: BD-CLS-Edit

> **Input** : Initial image $\mathbf{i}$; Initial label/prompt $\mathbf{v}$; SD model; Causal diagram $\mathcal{G}_{\mathbf{V}, \mathbf{I}}$; Initialized transformation $\psi_{\boldsymbol{\theta}}$
> **Output** : Ctf-consistency counterfactual image $\mathbf{i}'$ for $P^*(\mathbf{I}_{\mathbf{x}'} \mid \mathbf{i})$

1 $\mathbf{B} \leftarrow \texttt{Backdoor}(\mathbf{X}, \mathcal{G}_{\mathbf{V}, \mathbf{I}})$     // Backdoor set
2 $\mathbf{c} = \{\mathbf{x}, \mathbf{b}\} \leftarrow \texttt{Prompt}(\mathbf{X}, \mathbf{B}, \mathbf{v})$    // Initial Prompt
3 $\mathbf{n} \leftarrow P^{\text{SD}}(\mathbf{N} \mid \mathbf{i}, \mathbf{x}, \mathbf{b})$
4 find any $\psi_{\boldsymbol{\theta}}$ s.t. $\psi_{\boldsymbol{\theta}}(\mathbf{x}, \mathbf{x}', b, \mathbf{N}) \sim P(\mathbf{I} \mid \mathbf{x}', b)$ and $h^*_{\mathbf{ND}}(\psi_{\boldsymbol{\theta}}(\mathbf{x}, \mathbf{x}', b, \mathbf{n})) = h^*_{\mathbf{ND}}(\mathbf{i})$
5 $\mathbf{i}' \leftarrow \psi_{\boldsymbol{\theta}}(\mathbf{x}, \mathbf{x}', b, \mathbf{n})$
6 **return** $\mathbf{i}'$

---

come from a BD-CLS. We detail this optimization procedure in the next section.

### 4.1 Implementation of searching transformation $\psi$

We now describe the implementation of the 4th-step in Alg. 1, first focusing on the candidate space of $\psi_{\boldsymbol{\theta}}$ and the optimized parameters $\boldsymbol{\theta}$. We extend the original denoising process. Formally, with a specific sample $\mathbf{n} = \{\mathbf{i}^{(T)}, \mathbf{z}^{(T)}, \dots, \mathbf{z}^{(1)}\}$, the new transformation $\psi_{\boldsymbol{\theta}}$ is the iterative process

$$\mathbf{i}'^{(t-1)} = \widehat{\mu}(\mathbf{i}'^{(t)}, \mathbf{c} + \theta_t(\mathbf{c}' - \mathbf{c}), t) + \sigma_t \mathbf{z}^{(t)} \qquad (6)$$

where $\mathbf{i}'^{(T)} = \mathbf{i}^{(T)}$. To illustrate, the new prompt is linearly mixed between $\mathbf{c}$ and $\mathbf{c}'$ at each time step $t$ by $\theta_t$. This prompt mixing technique leverages the coarse-to-fine nature of the denoising process and demonstrates an ability to disentangle features [56, 39, 2, 9].

Next, we illustrate how to search $\boldsymbol{\theta} = \{\theta_1, \theta_2, ..\theta_T\}$ to satisfy two constraints in Eq. 5. First, it is necessary for $\psi_{\boldsymbol{\theta}}(\mathbf{x}, \mathbf{x}', b, \mathbf{n})$ to exhibit prompts $\mathbf{c}' = \{\mathbf{x}', \mathbf{b}\}$ to guarantee the first constraint. It is demonstrated in [44] that updating the parameters with the direction

$$\nabla_{\theta_t} \mathcal{L}_{\text{SDS}}(\mathbf{i}'_t, \mathbf{c}, \epsilon, t) = (\widehat{\epsilon}(\mathbf{i}'_t, \mathbf{c}, t) - \epsilon) \frac{\partial \mathbf{i}'_t}{\partial \theta_t}, \qquad (7)$$

will motivate $\mathbf{i}'$ to exhibit features $\mathbf{c}'$, where $\epsilon$ is the noise added in the forward process and $\widehat{\epsilon}$ is the noise predictor in diffusion (App. A.2). This gradient update can be interpreted as asking counterfactuals: "Given that $\mu_{\theta_t}$ generates $\mathbf{i}'_t$, how should $\theta_t$ be updated had the resulting $\mathbf{i}'_t$ resembles feature $\mathbf{c}'$ than $\mathbf{c}$?". However, updating $\theta$ in this direction does not guarantee the second constraint. due to the entanglement of $\mu_{\theta_t}$ (See Ex. 10). The key challenge is that the true mapping $h^*_{\mathbf{ND}}$ is unknown, making it hard to keep non-descendants unchanged without influence other descendants. To address this, we propose an alternative optimization direction inspired by the following proposition.

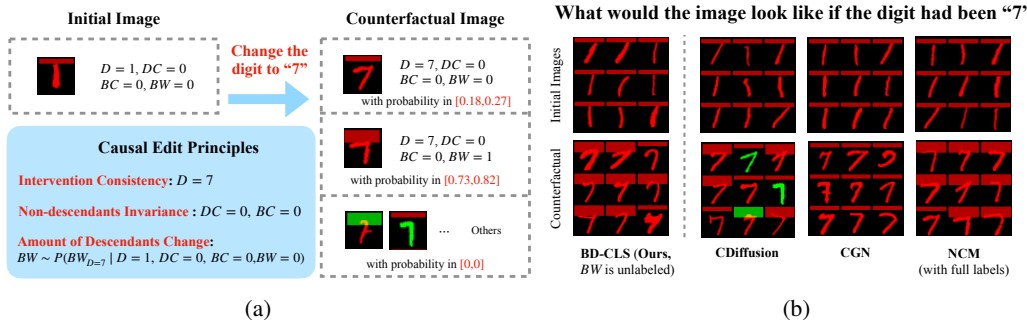

(a)                                                    (b)

Figure 5: Edit a red "1" with a thin red bar to digit "7". (a) Expectation of counterfactual consistent editing; (b) Edit results. Top - initial image. Bottom - counterfactual images.

**Proposition 2** (**Toy entanglement between binary** $X$, $Y$ **and** $R$). *Consider binary $X$, non-descendant $R$ and descendant $Y$. Suppose $P^*(y \mid \mathbf{pa}_Y) \neq P^*(y' \mid \mathbf{pa}_Y)$. Suppose $R$ and $Y$ are both entangled with $\{X, \mathbf{B}, \mathbf{N}\}$ in $\mathcal{M}^{\mathrm{SD}}$, and $R = \tau_R(X, \mathbf{B}, \mathbf{N}_1)$ and $Y = \tau_Y(X, \mathbf{B}, \mathbf{N}_2)$, where $\mathbf{N}_1 \cap \mathbf{N}_2 = \emptyset$ and $\mathbf{N}_1, \mathbf{N}_2 \subseteq \mathbf{N}$, then $P^{\widehat{\mathcal{M}}^{\mathrm{SD}}}(r'_{x'}|x, \mathbf{b}, r, y) = P^{\widehat{\mathcal{M}}^{\mathrm{SD}}}(r'_{x}|x', \mathbf{b}, r, y)$ and $P^{\widehat{\mathcal{M}}^{\mathrm{SD}}}(y'_{x'}|x, \mathbf{b}, r, y) \neq P^{\widehat{\mathcal{M}}^{\mathrm{SD}}}(y'_{x}|x', \mathbf{b}, r, y)$.* ∎

In words, this proposition suggests that even a descendant $Y$ and a non-descendant $R$, both entangled with $\mathbf{X}$, counterfactual behaviors can be leveraged to distinguish $Y$ and $R$. See Ex. 11. To use Prop. 2 for image editing tasks, let an $\mathbf{i}'$ and $\tilde{\mathbf{i}}$ have a different feature $X$ but the same other features. Then $P^{\widehat{\mathcal{M}}^{\mathrm{SD}}}(r'_{x'}|x, b, r, y)$ approximates the gradient of $R$ when $\mathbf{i}'$ moves to have feature $x'$ with Eq. 7. Similarly, $P^{\widehat{\mathcal{M}}^{\mathrm{SD}}}(r'_{x}|x', b, r, y)$ approximates the gradient when reverting $x'$ to $x$ in $\tilde{\mathbf{i}}$, and these two gradients are equal. In contrast, the gradients of $Y$ when toggling $X$ between $x$ and $x'$ in $\mathbf{i}$ and $\tilde{\mathbf{i}}$ under the same toggling are not equivalent. Thus, we can contrast two SDS losses to update gradients similar to DDS [18] [4] to encourage the new direction is orthogonal to $\mathbf{ND}$. Formally, this direction is

$$\nabla_{\theta_t}\mathcal{L}_{\mathrm{Ctf}}(\mathbf{i}'_t, \mathbf{c}', \tilde{\mathbf{i}}_t, \mathbf{c}, t) = (\widehat{\epsilon}(\mathbf{i}'_t, \mathbf{c}', t) - \widehat{\epsilon}(\tilde{\mathbf{i}}_t, \mathbf{c}, t))\frac{\partial \mathbf{i}'_t}{\partial \theta_t}, \tag{8}$$

and this idea is visualized in Fig. 6. Specifically, when guiding $\mathbf{i}'$ with the SDS direction and prompt $\mathbf{c}' = \{x', b\}$ (top-left panel in Fig. 6), the weather feature (e.g., the appearance of rain on the ground) changes from $x$ (sunny) to $x'$ (rainy). However, due to entanglement, the non-descendant feature, such as trees, also tends to change from $r$ to $r'$. Meanwhile, the descendant feature (e.g., the umbrella) correctly changes to $y'$. In contrast, when guiding $\tilde{\mathbf{i}}$ with the SDS direction and $\mathbf{c} = \{x, b\}$ (bottom left panel of Fig. 6), the weather changes from $x'$ (rainy) to $x$ (sunny). Yet, the non-descendant features again change in the same direction from $r$ to $r'$, and the umbrella no longer tends to change to $y'$. After combining these contrasting directions (right panel in Fig. 6), the weather reliably changes from $x$ to $x'$, the

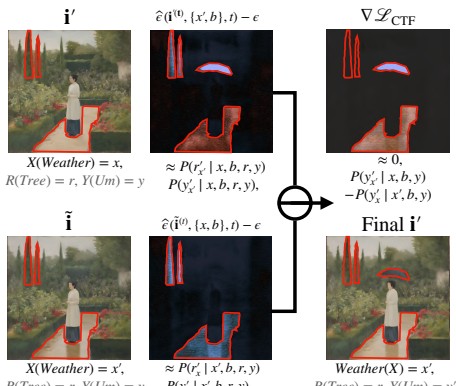

Figure 6: Optimization direction $\nabla_{\theta_t}\mathcal{L}_{\mathrm{Ctf}}$.

umbrella (a descendant) appropriately changes as well, and the non-descendant feature (e.g., the tree) remains invariant, correcting the entanglement artifacts through cancellation.

To obtain $\tilde{\mathbf{i}}_t$, we move $\mathbf{i}_t$ in the DDS direction, which only changes feature $\mathbf{X}$ but preserves others as the same. Specifically, a subset of time steps $\tilde{\mathbf{T}}$ is randomly selected, i.e., $\tilde{\mathbf{T}} \subseteq \{1, \dots, \mathbf{T}\}$. And for every $t \in \tilde{\mathbf{T}}$, we follow the DDS

$$\tilde{\mathbf{i}}^{(t)} = \mathbf{i}^{(t)} + \lambda_t(\widehat{\epsilon}(\mathbf{i}_t, \mathbf{c}', t) - \widehat{\epsilon}(\mathbf{i}_t, \mathbf{c}, t)) \tag{9}$$

where $\lambda_t$ is a hyperparamter controlling the intensity of the change of $\mathbf{X}$. More details on this optimization are given in App. B.

---

[4]See the details of DDS in App. A.2. Notice that the key improvement is that we leverage the Prop. 2.

# 5 Experiments

In this section, we empirically validate our theoretical results (Thm.2) and demonstrate the effectiveness of **BD-CLS-Edit** (Alg.1). Additional experimental details are in App. D.

## 5.1 Colored MNIST and Bars

We first evaluate the guarantees provided by BD-CLS (Thm.2) on a modified MNIST dataset [14, 36] featuring colored digits and bars. [5]. The ground truth ASCM includes factors: Digit (0-9 $D$), Digit Color (red $DC = 0$; green $DC = 1$), Bar Width (thin $BW = 0$; thick $BW = 1$), Bar Color (red $BC = 0$; green $BC = 1$), and other latent factors such as handwriting style. The causal relationships are shown in Fig. 7. To illustrate, the digit ($D$) and digit color ($DC$) are confounded: larger digits tend to be red, and smaller digits tend to be green, but they do not directly affect each other. The digit color ($DC$) has a positive effect on bar color ($BC$), as red digits often have red

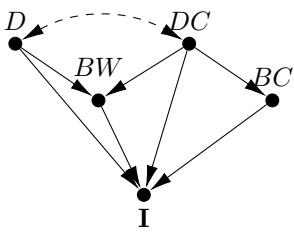

Figure 7: $\mathcal{G}$ in Sec. 5.1.

bars. The digit ($D$) has a positive effect on bar width ($BW$); larger digits are more likely to be with thick bars. However, this effect reverses when the digit color is green.

We first consider editing the digit. Suppose that we are editing a red "1" with a thin red bar and wonder what would happen had the digit "1" been a "7". According to the editing principles in Thm. 1, the edit should achieve (1) interventional consistency. The digit should be "7"; (2) non-descendants invariance. digit color ($DC$) and bar color ($BC$) remain red;. (3) Descendant Delta. The $BW$ as a descendant, may change thicker and the probability should be $Q = P(BW_{D=7} = 1 | D = 1, DC = 0, BC = 0, BW = 0)$. To guarantee counterfactual consistency, the estimation of $Q$ should be within the bound according to Def. 3. These edit expectations are shown in Fig. 5(a).

We evaluate both causal and non-causal methods on the digit editing task. According to Theorem 2, our proposed BD-CLS enables counterfactual consistent editing, even with unlabeled features. We obtain BD-CLS using a Neural Causal Model (NCM)[59, 60, 36] trained without labels for $BW$. we also train an NCM with full supervision. For comparison, we include two non-causal baselines: (1) Conditional Diffusion (CDiffusion), which relies on correlations, and (2) CGN[50], which preserves original semantics. The editing results (Fig. 5(b)) show that all models change the digits to "7". However, CDiffusion alters non-descendants (e.g., color), and CGN fails to change descendants (bar width). In contrast,

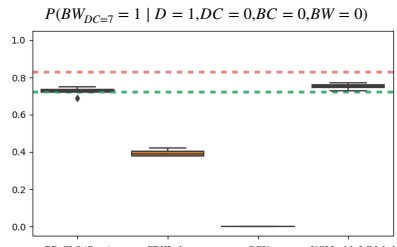

Figure 8: The estimated F-ctf query by our BD-CLS and baselines.

BD-CLS and fully supervised NCM preserve non-descendants and correctly update the $BW$, despite BD-CLS not using $BW$ labels. To quantify descendant changes, we estimate the query $Q$ (Fig.8) by measuring how often the bar becomes thicker after editing. Both BD-CLS and fully supervised NCM stay within theoretical bounds, while CDiffusion and CGN do not. Additional tasks are in App. D.1.

## 5.2 Real World Scenarios Counterfactual Editing

In this section, we validate **BD-CLS-Edit** for counterfactual image editing in real-world scenarios. We compare it against three non-causal SOTAs: (1) DDPM Inversion [22] and (2) SDEdit[33], representing LS inversion, and (3) DDS [22], which emphasizes semantic invariance. Qualitative evaluations are shown here. Quantitative evaluations and additional details are provided in App. D.3.

We begin with the setting from Example 1, where the goal is to change the weather from sunny to rainy in an image of a young (or old) lady in a garden (or street). The causal relationships between the generative factors are shown in Fig.9(a)). According to Thm. 1, non-descendants (e.g., scene, age, pose, person's height) should be preserved, while descendants (e.g., umbrella, shadows) should change accordingly regardless of whether they are prompted. For example, an umbrella may appear, and shadows should become fuzzier on wet ground due to the weather change. As shown in Fig. 9(d), all methods achieve interventional consistency. However, DDPM inversion and SDEdit alter non-

---

[5]A bar in an image refers to a complete row of pixels with the same color.

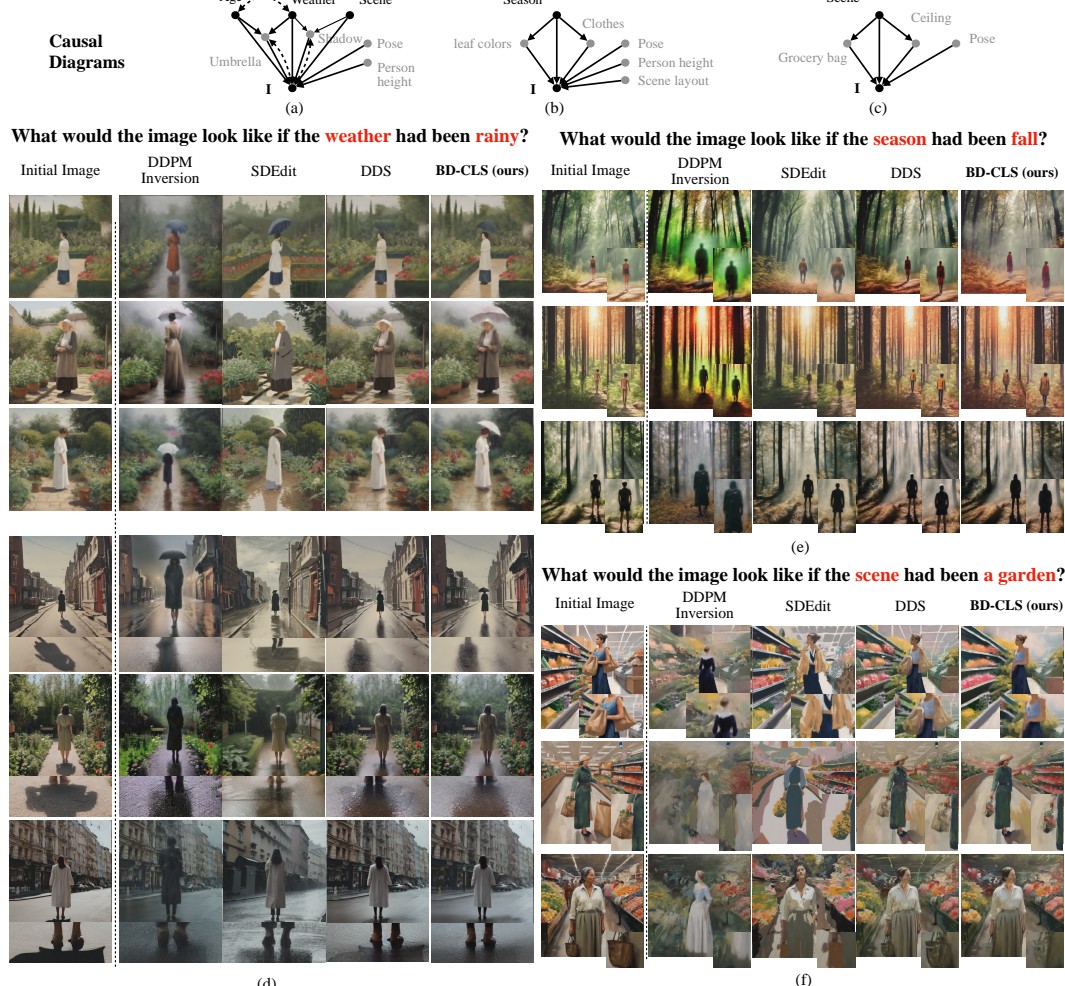

Figure 9: The causal diagrams and editing results for Sec. 5.2.

descendants, changing the lady pose and part of the scene. DDS maintains visual similarity to the original image, but does not reflect downstream effects. To illustrate, the umbrella does not appear and the shadows on sunny days are preserved. In contrast, BD-CLS preserves non-descendants and correctly reflects the causal effects on descendants, such as the umbrella and shadow.

Next, we edit an image of a person in a forest by changing the season from summer to fall (Fig.9(b)). Non-descendants (e.g., person's pose and heights, forest layout) should be preserved, while descendants, e.g., clothing and leaf color, should change since people wear warmer clothes in the fall. As shown in Fig. 9(e), DDPM inversion and SDEdit fail to preserve the person's identity. DDS preserves the person's features but produces unrealistic clothing by retaining too much from the original image. BD-CLS-Edit accurately reflects warmer clothing while preserving non-descendants. Third, we edit an image of a person in a grocery store by changing the scene to a garden (Fig. 9(c)). Non-descendants (e.g., a person's pose) should remain unchanged, while descendants, like a grocery bag, should be removed, as a person is unlikely to bring a grocery bag into a garden. In addition, the ceiling is also likely to be removed since a garden usually is not indoors. As shown in Fig. 9(f), DDPM and SDEdit inversion noticeably alter the person. DDS keeps the grocery bag. In contrast, BD-CLS-Edit preserves non-descendants and removes the grocery bag.

# 6 Conclusions

We develop a counterfactual image editing framework that works with pre-trained diffusion models under weak supervision, without retraining. We introduce a data structure called Backdoor Causal Latent Space (BD-CLS), which ensures counterfactual consistency (Thm. 1 and 2), and then develop an **BD-CLS-Edit** (Alg. 1) to extract it from a Stable Diffusion model. Our approach advances image editing in terms of causal realism, scalability, and weak supervision.

## Acknowledgments

This research is supported in part by the NSF, ONR, AFOSR, DoE, Amazon, JP Morgan, and The Alfred P. Sloan Foundation.

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

# Appendix

## Contents

## A  Background

### A.1  Causal Models

Our work relies on the basic semantical framework structural causal models (SCMs) [41, Ch. 7], and we follow the presentation in [4].

**Definition 5** (Structure Causal Model(SCM)). A Structure Causal Model (for short, SCM) is a 4-tuple $< \mathbf{U}, \mathbf{V}, \mathcal{F}, P(\mathbf{U}) >$, where (1) $\mathbf{U}$ is a set of background variables, also called exogenous variables, that are determined by factors outside the model; (2) $\mathbf{V} = \{V_1, V_2, \ldots, V_d\}$ is the set of endogenous variables that are determined by other variables in the model; (3) $\mathcal{F}$ is the set of functions $\{f_{V_1}, f_{V_2} \ldots, f_{V_d}\}$ mapping $\mathbf{U}_{V_j} \cup \mathbf{Pa}_{V_j}$ to $V_j$, where $\mathbf{U}_{V_j} \subseteq \mathbf{U}$ and $\mathbf{Pa}_{V_j} \subseteq \mathbf{V} \backslash V_j$; (4) $P(\mathbf{U})$ is a probability function over the domain of $\mathbf{U}$. ∎

We bring forth the longer and more formal definition of causal diagrams induced by the SCMs.

**Definition 6** (Causal Diagram [4, Def. 13]). Consider an SCM $\mathcal{M} = \langle \mathbf{U}, \mathbf{V}, \mathcal{F}, P(\mathbf{U}) \rangle$. We construct a graph $\mathcal{G}$ using $\mathcal{M}$ as follows:

(1) add a vertex for every variable in $\mathbf{V}$,

(2) add a directed edge $(V_j \to V_i)$ for every $V_i, V_j \in \mathbf{V}$ if $V_j$ appears as an argument of $f_{V_i} \in \mathcal{F}$,

(3) add a bidirected edge $(V_j \leftarrow\text{-}\rightarrow V_i)$ for every $V_i, V_j \in \mathbf{V}$ if the corresponding $\mathbf{U}_{V_i}, \mathbf{U}_{V_j} \subseteq \mathbf{U}$ are not independent or if $f_{V_i}$ and $f_{V_j}$ share some $U \in \mathbf{U}$ as an argument.

We refer to $\mathcal{G}$ as the causal diagram induced by $\mathcal{M}$ (or "causal diagram of $\mathcal{M}$" for short). ∎

Then a structure can be defined with the bi-drected edges in a causal diagram.

**Definition 7** (C-component [4, Def. 14]). Let $\mathcal{G}$ be a causal diagram. Let $\mathbf{C}_1, \mathbf{C}_2, \ldots, \mathbf{C}_k$ be a partition over the set of variables $\mathbf{V}$. where $\mathbf{C}_i$ is said to be a confounded component (C-component for short) of $\mathcal{G}$ if for every $V_a, V_b \in \mathbf{C}_i$ , there exists a path made entirely of bidirected edges between $V_a$ and $V_b$ in $\mathcal{G}$, and $\mathbf{C}_i$ is maximal. We denote $\mathbf{C}(V_a)$ as the C-component containing $V_a$. ∎

An intervention on a subset of $\mathbf{X} \subseteq \mathbf{V}$, denoted by $do(\mathbf{x})$, is an operation where $\mathbf{X}$ takes value $\mathbf{x}$, regardless how $\mathbf{X}$ are originally defined. For an SCM $\mathcal{M}$, let $\mathcal{M}_{\mathbf{x}}$ be the submodel of $\mathcal{M}$ induced by $do(\mathbf{x})$. For any subset $\mathbf{Y} \subseteq \mathbf{V}$, the potential outcome $\mathbf{Y}_{\mathbf{x}}(\mathbf{u})$ is defined as the solution of $\mathbf{Y}$ after feeding $\mathbf{U} = \mathbf{u}$ into the submodel $\mathcal{M}_{\mathbf{x}}$. Then $\mathbf{Y}_{\mathbf{x}}$ is called a counterfactual variable induced by $\mathcal{M}$. Specifically, the event $\mathbf{Y}_{\mathbf{x}} = \mathbf{y}$ represent "$\mathbf{Y}$ would be $\mathbf{y}$ had $\mathbf{X}$ been $\mathbf{x}$". The counterfactual quantities induced by an SCM $\mathcal{M}$ are defined as in Eq. 1:

$$P^{\mathcal{M}}(\mathbf{y}_{1[\mathbf{x}]}, \mathbf{y}_{2[\mathbf{x}_2]}, \ldots,) = \int_{\mathcal{X}_{\mathbf{U}}} \mathbb{1}[\mathbf{Y}_{\mathbf{x}_1}(\mathbf{u}) = \mathbf{y}_1, \mathbf{Y}_{\mathbf{x}_2}(\mathbf{u}) = \mathbf{y}_2, \ldots] dP(\mathbf{u}), \qquad (10)$$

Each $\mathbf{Y}_i$ corresponds to a set of variables in a world where the original mechanisms $f_X$ are replaced with constants $\mathbf{x}_i$ for each $X \in \mathbf{X}_i$; this is also known as the mutilation procedure. This procedure corresponds to interventions, and we use subscripts to denote the intervening variables (e.g. $\mathbf{Y}_{\mathbf{x}}$) or subscripts with brackets when the variables are indexed (e.g. $\mathbf{Y}_{1[\mathbf{x}_1]}$). For instance, $P(y_x, y'_{x'})$ is the probability of the joint counterfactual event $Y = y$ had $X$ been $x$ and $Y = y'$ had $X$ been $x'$.

After describing a causal model in SCM semantics, we can also define a graphical model independent of a particular generative process and instead based on a set of constraints. Counterfactual Bayesian Netwrok [3, 10], similarly to a Bayesian Network or a Causal Bayesian Network [4], which are graphical models that relate a graph and a (set of) counterfactual distribution(s) is defined as follows.

**Definition 8** (CTFBN Semi-Markovian [3, Def. 13.2.1]). Let $\mathbf{P}^{**}$ be the collection of all distributions of the form $P(W_{1[x1]}, W_{2[x2]}, \ldots)$, where $W_i \in \mathbf{V}$, $X_i \subseteq \mathbf{V}$. A directed acyclic graph $\mathcal{G}$ over $\mathbf{V}$ is a Counterfactual Bayesian Network for $\mathbf{P}^{***}$ if:

1. [Independence Restrictions] Let $\mathbf{W}_*$ be a set of counterfactuals of the form $W_{\mathbf{pa}_w}$, $\mathbf{C}_1, ..., \mathbf{C}_l$ the c-components of $G[\mathbf{V}(\mathbf{W}_*)]$, and $\mathbf{C}_{1*}, ..., \mathbf{C}_{l*}$ the corresponding partition over $\mathbf{W}_*$. Then $P(\mathbf{W}_*)$ factorizes as

$$P(\bigwedge_{W_{pa_w} \in \mathbf{W}_*} W_{pa_w}) = \prod_{j=1}^{l} P(\bigwedge_{W_{pa_w} \in \mathbf{C}_{j*}} W_{pa_w}) \qquad (11)$$

2. [Exclusion Restrictions] For every variable $Y \in \mathbf{V}$ with parents $\mathbf{Pa}_y$, for every set $\mathbf{Z} \subseteq \mathbf{V} \backslash (\mathbf{Pa}_y \cup \{Y\})$, and any counterfactual set $\mathbf{W}_*$, we have

$$P(Y_{\mathbf{pa}_y, \mathbf{z}}, \mathbf{W}_*) = P(Y_{\mathbf{pa}_y}, \mathbf{W}_*) \qquad (12)$$

3. [Local Consistency] For every variable $Y \in \mathbf{V}$ with parents $\mathbf{Pa}_y$, let $\mathbf{X} \subseteq \mathbf{Pa}_y$, then for every set $\mathbf{Z} \subseteq \mathbf{V} \backslash (\mathbf{X} \cup \{Y\})$, and any counterfactual set $\mathbf{W}_*$, we have

$$P(Y_{\mathbf{z}} = y, \mathbf{X}_{\mathbf{z}} = \mathbf{x}, \mathbf{W}_*) = P(Y_{\mathbf{xz}} = y, \mathbf{X}_{\mathbf{z}} = \mathbf{x}, \mathbf{W}_*) \qquad (13)$$

Then the compatibility of a graph and an SCM can be defined as follows.

**Definition 9** (Compatible with $\mathcal{G}$ on counterfactuals). Consider an SCM $\mathcal{M}$ over $\mathbf{V}$ and a DAG $\mathcal{G}$ over $\mathbf{V}$. The SCM $\mathcal{M}$ (or the graph $\mathcal{G}^{\mathcal{M}}$ induced by $\mathcal{M}$) is said to be compatible with $\mathcal{G}$ if $\mathcal{G}$ is a CTFBN for $\mathbf{P}^{**}$, where $\mathbf{P}^{**}$ be the collection of all distributions of the form $P(\mathbf{W}_{1[\mathbf{x}_1]}, \mathbf{W}_{2[\mathbf{x}_2]}, \ldots)$, where $\mathbf{W}_i \in \mathbf{V}$, $\mathbf{X}_i \subseteq \mathbf{V}$.

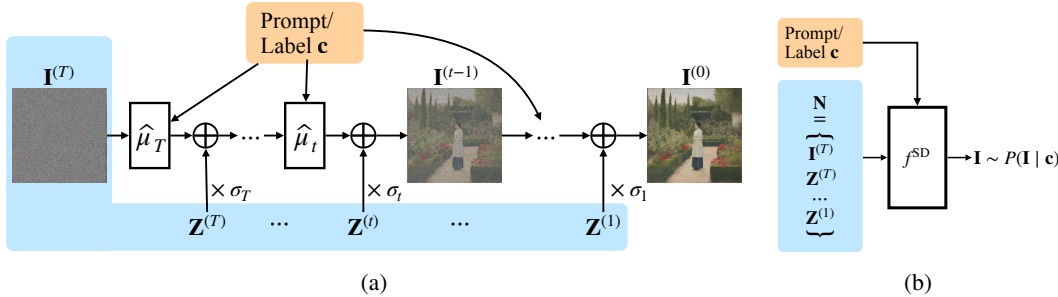

Figure S1: The generation process of a diffusion model. (a) recursion version; (b) proxy model version.

Informally speaking, we say that an SCM $\mathcal{M}$ (or the graph $\mathcal{G}^{\mathcal{M}}$ induced by $\mathcal{M}$) is compatible with a given graph $\mathcal{G}$ if $\mathcal{G}^{\mathcal{M}}$ imposes constraints that are at least as strong as those in $\mathcal{G}$. Since the absence of an edge in a causal diagram represents a constraint, this means that if $\mathcal{G}^{\mathcal{M}}$ has strictly fewer edges than $\mathcal{G}$, then $\mathcal{M}$ is compatible with $\mathcal{G}$. Thus, as we discussed in 4, an SCM is compatible with $\mathcal{G}$ does not mean that SCM induces $\mathcal{G}$ exactly.

Given the observed distribution $P(\mathbf{V})$ and the causal diagram $\mathcal{G}$, the optimal counterfactual bounds are closed intervals based on the optimization problem following [61].

**Definition 10** (Optimal Counterfactual Bounds). For a causal diagram $\mathcal{G}$ and observed distributions $P(\mathbf{V})$, the *optimal bound* $[l, r]$ over a counterfactual probability $P^{\mathcal{M}}(\mathbf{y_x}, \ldots, \mathbf{z_w})$ is defined as, respectively, the minimum and maximum of the following optimization problem:

$$\max / \min_{\mathcal{M} \in \Omega(\mathcal{G})} P^{\mathcal{M}}(\mathbf{y_x}, \ldots, \mathbf{z_w}) \ \text{s.t.} P^{\mathcal{M}}(\mathbf{V}) = P(\mathbf{V}) \tag{14}$$

where $\Omega(\mathcal{G})$ is the space of all SCMs that agree with the diagram $\mathcal{G}$, i.e., $\Omega(\mathcal{G}) = \{\forall \mathcal{M} | \mathcal{G}_{\mathcal{M}} = \mathcal{G}\}$. ∎

By the formulation of Eq. 14, all possible values of counterfactual query induced by SCMs that agree with the observational distributions and causal diagram are contained in the closed interval $[l, r]$.

## A.2 Denoising Diffusion Probabilistic Model and Score Distillation Sampling

A Denoising Diffusion Probabilistic Model (DDPM) model [54, 20] are deep generative models that consists of a forward process and reverse process with $T$ time-steps. The forward process gradually perturbs $\mathbf{I}^{(t-1)}$ (the image at step $t-1$) with gaussian noise to $\mathbf{I}^{(t)}$ (the image at step $t$), where $\mathbf{I}^{(0)}$ (image at step 0) is the original image. Formally,

$$\mathbf{I}^{(t)} = \sqrt{\bar{\alpha}_t}\mathbf{I}^{(0)} + \sqrt{1 - \bar{\alpha}_t}\boldsymbol{\mathcal{E}} \tag{15}$$

where $\bar{\alpha}_t$ is the noise scheduler and $\boldsymbol{\mathcal{E}}$ is the standard gaussian noise. In the reverse process, diffusion model predict noise $\boldsymbol{\mathcal{E}}$ at each time step using a neural network $\widehat{\epsilon}$ taking $\mathbf{I}^{(t)}$ and a text prompt or label $\mathbf{c}$ as input. Specifically, the reverse starts from a random Gaussian noise vector $\mathbf{I}^{(T)} \sim \mathcal{N}(0, \mathbf{1})$ and iteratively predicts noise with a using recursion

$$\mathbf{I}^{(t-1)} = \widehat{\mu}(\mathbf{I}^{(t)}, \mathbf{c}, t) + \sigma_t \mathbf{Z}^{(t)} \tag{16}$$

where $\mathbf{c}$ is the text prompt/label and $\mathbf{Z}^{(t)}$ are gaussian random vectors; and $\sigma_t$ are pre-specified variance terms.

$$\widehat{\mu}(\mathbf{I}^{(t)}, \mathbf{c}, t) = \sqrt{\frac{\bar{\alpha}_{t-1}}{\bar{\alpha}_t}}(\mathbf{I}^{(t)} - \frac{\bar{\alpha}_t}{\bar{\alpha}_{t-1}\sqrt{1 - \bar{\alpha}_t}}\widehat{\epsilon}(\mathbf{I}^{(t)}, \mathbf{c}, t)) \tag{17}$$

This process is illustrated in Fig. S1(a).

Text conditioned diffusion models use classifier-free guidance [21] to sample images from conditional distribution $P(\mathbf{I} \mid \mathbf{c})$. Specifically, the reverse process does not only involve noise precitor $\widehat{\epsilon}(\mathbf{i}^{(t)}, \mathbf{c}, t)$ with prompt $\mathbf{c}$ but also a non-conditional term. Formally, the denoise term is

$$(1 + \omega)\widehat{\epsilon}(\mathbf{i}^{(t)}, \mathbf{c}, t) - \widehat{\epsilon}(\mathbf{i}^{(t)}, \emptyset, t) \tag{18}$$

where $\omega$ is a hyperparameter and $\emptyset$ is the null text. In this work, we fix the parameter $\omega$ and simply denote the denoise term **in the generation process** as $\widehat{\epsilon}(\mathbf{i}^{(t)}, \mathbf{c}, t)$. Similarly, the corresponding mean predictor Then, as we discussed in Sec. 4, this recursion process can be seen as a function that takes input $\mathbf{N} = \{\mathbf{I}^{(T)}, \mathbf{Z}^{(T)}, \ldots, \mathbf{Z}^{(1)}\}$ and generates the image $\mathbf{I}$ as illustrated in Fig. S1(b). In Sec. 4, we will demonstrate how to transform $\{\mathbf{C}, \mathbf{N}\}$ to our proposed BD-CLS and use it for image editing.

In the DDPM training process, the network is trained to predict the noise $\mathcal{E}$ scheduled in the forward process. The training objective can be expressed as:

$$\mathcal{L}_{\text{Diff}} = \|\widehat{\epsilon}(\mathbf{i}^{(t)}, \mathbf{c}, t) - \boldsymbol{\epsilon}\|_2^2 \tag{19}$$

Recently, [44] proposed Score Distillation Sampling (SDS) and showed that given an arbitrary differentiable generator $g_{\boldsymbol{\theta}}$ that can generate $\mathbf{i}'^{(t)}$ (the noise image $\mathbf{i}'$ at the time step $t$), updating the parameters of the $g_{\boldsymbol{\theta}}$ in the following direction can render features $\mathbf{c}$ in the image:

$$\nabla_{\boldsymbol{\theta}} \mathcal{L}_{\text{SDS}}(\mathbf{i}'_t, \mathbf{c}', \epsilon, t) = (\widehat{\epsilon}(\mathbf{i}'_t, \mathbf{c}', t) - \epsilon) \frac{\partial \mathbf{i}'_t}{\partial \boldsymbol{\theta}}, \tag{20}$$

Later, [18] proposes the Delta Denosing Score (DDS) and shows that by updating the parameters in the following direction, the generator produces the image $\mathbf{i}'$ that is the closest image to $\mathbf{i}$, where $\mathbf{i}'$ matches the text $\mathbf{c}'$ and $\mathbf{i}'$ matches the text $\mathbf{c}$,

$$\nabla_{\theta_t} \mathcal{L}_{\text{DDS}}(\mathbf{i}'_t, \mathbf{c}, \epsilon, t) = (\widehat{\epsilon}_\omega(\mathbf{i}'_t, \mathbf{c}', t) - \widehat{\epsilon}_\omega(\mathbf{i}_t, \mathbf{c}, t)) \frac{\partial \mathbf{i}'_t}{\partial \boldsymbol{\theta}}, \tag{21}$$

In other words, DDS is one of the semantic invariance image editing approaches that preserves the features except $\mathbf{c}$ as close as possible to the features in the initial image.

## B  Algorithm Details

Here we illustrate more details of our proposed **BD-CLS-Edit**.

First, we justify the necessity of the second step in **BD-CLS-Edit**, constructing the prompt using $\mathbf{X}$, $\mathbf{B}$. The key idea is that the prompt must not include descendants of $\mathbf{X}$. If the target prompt involves a descendant of $\mathbf{X}$, then after the intervention, that descendant will have a fixed value that is aligned with the prompt. This contradicts one of the editing principles, descendant delta, demonstrated by Thm.1.

In the third step of **BD-CLS-Edit**, $\mathbf{N}$ is sampled from the observational distribution $P^{\text{SD}}(\mathbf{N} \mid \mathbf{i}, \mathbf{x}, \mathbf{b})$. This sampling process is related to inversion process that aim to find a noise sample $\mathbf{n}$ given the prompt $\mathbf{c} = \mathbf{x}, \mathbf{b}$, such that the diffusion model $\mathcal{M}^{\text{SD}}$ reproduces the source real image $\mathbf{i}$ [11, 19, 34, 55, 38]. In other words, this is related to the inference process corresponding to $P(\mathbf{n} \mid \mathbf{i}, \mathbf{c})$. However, exiting methods often focus on finding a single valid noise sample $\mathbf{n}$ that can reproduce the initial image, rather than sampling from the distribution $P^{\text{SD}}(\mathbf{N} \mid \mathbf{i}, \mathbf{c})$. This is incorrect according to Thm.1. For instance, in an extreme case, some methods deterministically compute a specific $\mathbf{n}$ given the initial image. If such a deterministic $\mathbf{n}$ is used

---

**Algorithm 2**: DDPM Sampling [22]

**Input** : Initial image $\mathbf{i}$; Prompt $\mathbf{c}$; SD model;
**Output** : $\mathbf{n}$ from

1   $\mathbf{n} \leftarrow \{\}$
2   **for** $t \leftarrow T$ **to** 1 **do**
3     $\epsilon \sim \mathcal{N}(0, 1)$
4     $i^{(t)} \sim \sqrt{\bar{\alpha}_t} i^{(t)} + \sqrt{1 - \bar{\alpha}_t}\epsilon$
5   **for** $t \leftarrow 1$ **to** $T$ **do**
6     $z^{(t)} \sim (i^{(t-1)} - \widehat{\mu}(i^{(t)}, \mathbf{c}, t))/\sigma_t$
7     $i^{(t-1)} \leftarrow \widehat{\mu}(i^{(t)}, \mathbf{c}, t) + \sigma_t z^{(t)}$
8   **return** $\mathbf{n} = \{i^{(T)}, z^{(T)}, \ldots, z^{(1)}\}$

---

during generation after an intervention, the descendants become fixed, then there is no randomness left in the process. However, Thm.1 implies that the descendants should follow a counterfactual distribution $P^*(\mathbf{DE} \mid \mathbf{v}, \mathbf{l})$, and therefore should vary accordingly (see Ex. 7). In this work, we leverage the DDPM inversion [22] to sample $\mathbf{n}$ from $P^{\text{SD}}(\mathbf{N} \mid \mathbf{i}, \mathbf{c})$ and this full sampling algorithm is shown in Alg. 2.

For the forth step of **BD-CLS-Edit**, we first elaborate more on $\widehat{\mu}_{\boldsymbol{\theta}}$. The denoising process is modified to take as input a mixing prompt of $\mathbf{c} = \{\mathbf{x}, \mathbf{b}\}$ and $\mathbf{c}' = \{\mathbf{x}', \mathbf{b}\}$. Formally, with a specific sample $\mathbf{n} = \{\mathbf{i}^{(T)}, \mathbf{z}^{(T)}, \ldots, \mathbf{z}^{(1)}\}$, the transformation $\psi_{\boldsymbol{\theta}}$ is the iterative process

$$\mathbf{i}'^{(t-1)} = \widehat{\mu}(\mathbf{i}'^{(t)}, \mathbf{c} + \theta_t(\mathbf{c}' - \mathbf{c}), t) + \sigma_t \mathbf{z}^{(t)} \tag{22}$$

**Algorithm 3**: Search $\psi$ in the forth step of Alg. 1

---

**Input** : $\mathbf{n} = \{\mathbf{i}^{(T)}, \mathbf{z}^{(T)}, \ldots, \mathbf{z}^{(1)}\}$; Initial $\boldsymbol{\theta} = \{\theta_1, \ldots, \theta_T\}$; Selected $\tilde{\mathbf{T}}$; Adjustment parameters $\lambda\theta = \{\lambda_1, \ldots, \lambda_T\}$; Target prompt embedding $\mathbf{c}' = \{\mathbf{x}', \mathbf{b}\}$; Initial prompt embedding $\mathbf{c} = \{\mathbf{x}, \mathbf{b}\}$; Optimization iteration number $n_{\max}$; Noise predictor $\widehat{\epsilon}$ and mean predictor $\widehat{\mu}$ in SD model; Variance scheduler $\{\sigma_t\}_{t=1}^T$ in SD model; learning rate $\gamma$; Clip value $\theta_{\max}$

**Output**: $\boldsymbol{\theta} = \{\theta_1, \ldots, \theta_T\}$

---

**1** $\mathbf{i}'^{(t)} \leftarrow \mathbf{i}^{(T)}$

**2** **for** $t \leftarrow T$ **to** $2$ **do**

    // get $\tilde{\mathbf{i}}^{(t-1)}$

**3**    $\tilde{\mathbf{i}}^{(t-1)} \leftarrow \widehat{\mu}(\mathbf{i}'^{(t)}, \mathbf{c}, t) + \sigma_t \mathbf{z}^{(t)}$

**4**    **if** $t \in \tilde{\mathbf{T}}$ **then**

**5**       $\tilde{\mathbf{i}}^{(t)} \leftarrow \tilde{\mathbf{i}}^{(t)} + \lambda_t(\widehat{\epsilon}(\mathbf{i}_t, \mathbf{c}', t) - \widehat{\epsilon}(\mathbf{i}_t, \mathbf{c}, t))$

**6**    **for** $i \leftarrow 1$ **to** $n_{\max}$ **do**

        // get $\mathbf{i}'^{(t-1)}$

**7**       $\mathbf{c}_{\mathrm{mix}} \leftarrow \mathbf{c} + \theta_t(\mathbf{c}' - \mathbf{c})$

**8**       $\mathbf{i}'^{(t-1)} \leftarrow \widehat{\mu}(\mathbf{i}'^{(t)}, \mathbf{c}_{\mathrm{mix}}, t) + \sigma_t \mathbf{z}^{(t)}$

        // Update $\boldsymbol{\theta}$

**9**       $\boldsymbol{\theta} \leftarrow \boldsymbol{\theta} + \gamma \nabla_{\theta_t} \mathcal{L}_{\mathrm{Ctf}}(\mathbf{i}'_t, \mathbf{c}', \tilde{\mathbf{i}}_t, \mathbf{c}, t)$

**10**    $\boldsymbol{\theta} \leftarrow \mathrm{clip}(\boldsymbol{\theta}, \theta_{\max})$

**11**    $\mathbf{i}'^{(t-1)} \leftarrow \widehat{\mu}(\mathbf{i}'^{(t)}, \mathbf{c}_{\mathrm{mix}}, t) + \sigma_t \mathbf{z}^{(t)}$

**12** **return** $\boldsymbol{\theta} = \{\theta_1, \ldots, \theta_T\}$

---

where $\mathbf{i}'^{(T)} = \mathbf{i}^{(T)}$. To illustrate, the new prompt is linearly mixed between $\mathbf{c}$ and $\mathbf{c}'$ at each time step $t$ by a different parameter $\theta_t$. When $\theta_t = 0$, the input prompt at step $t$ is the initial labels $\mathbf{c} = \{\mathbf{x}, \mathbf{b}\}$, which encourages the image output to have features $\mathbf{X} = \mathbf{x}$ and $\mathbf{B} = \mathbf{b}$; when $\theta_t = 1$, the input prompt at step $t$ is the target labels $\mathbf{c}' = \{\mathbf{x}, \mathbf{b}\}$, which encourages the image output to have features $\mathbf{X} = \mathbf{x}'$ and $\mathbf{B} = \mathbf{b}$. Formally,

$$\begin{aligned}
\psi_{\boldsymbol{\theta}}(\mathbf{c}, \mathbf{N}) &\sim P(\mathbf{I} \mid \mathbf{c}) \quad \text{If } \theta_1 = \theta_2 = \cdots = \theta_T = 0 \\
\psi_{\boldsymbol{\theta}}(\mathbf{c}', \mathbf{N}) &\sim P(\mathbf{I} \mid \mathbf{c}') \quad \text{If } \theta_1 = \theta_2 = \cdots = \theta_T = 1
\end{aligned} \tag{23}$$

Then, we illustrate the new counterfactual updating direction designed based on Prop. 2. Here we explain more about this process leveraging the visualization. Specifically, when guiding $\mathbf{i}'$ with the SDS direction and prompt $\mathbf{c}' = \{x', b\}$ (top-left panel in Fig. S2), the weather feature (e.g., the appearance of rain on the ground) changes from $x$ (sunny) to $x'$ (rainy). However, due to entanglement, the non-descendant feature, such as trees, also tends to change from $r$ to $r'$. Meanwhile, the descendant feature (e.g., the umbrella) correctly changes to $y'$. In contrast, when guiding $\tilde{\mathbf{i}}$ with the SDS direction and $\mathbf{c} = \{x, b\}$ (bottom left panel of Fig. S2), the weather changes from $x'$ (rainy) to $x$ (sunny). Yet, the non-descendant features again change in the same direction from $r$ to $r'$, and the umbrella no longer tends to change to $y'$. After combining these contrasting directions (right panel in Fig. S2), the weather reliably changes from $x$ to $x'$, the umbrella (a descendant) appropriately changes as well, and the non-descendant feature (e.g., the tree) remains invariant, correcting the entanglement artifacts through cancellation.

To obtain $\tilde{\mathbf{i}}_t$, we move $\mathbf{i}_t$ in the DDS direction, which only changes feature $\mathbf{X}$ but preserves others as the same. Specifically, a subset of time steps $\tilde{\mathbf{T}}$ is selected, i.e., $\tilde{\mathbf{T}} \subseteq \{1, \ldots, \mathbf{T}\}$. And for every $t \in \tilde{\mathbf{T}}$, we follow the DDS

$$\tilde{\mathbf{i}}^{(t)} = \mathbf{i}^{(t)} + \lambda_t(\widehat{\epsilon}(\mathbf{i}_t, \mathbf{c}', t) - \widehat{\epsilon}(\mathbf{i}_t, \mathbf{c}, t)) \tag{24}$$

where $\lambda_t$ is a hyperparamter controlling the intensity of the change of $\mathbf{X}$. Finally, to prevent $\theta_t$ from being too large, $\theta_t$ is cut to a fixed maximum value $\theta_{\max}$. This step is needed since a valid solution to satisfy the first constraint of Eq. 5 is $\theta_1 = \theta_2 = \cdots = \theta_T = 1$ as illustrated in Eq. 23. Thus, $\theta_t$ should be encouraged to be around 1. The complete procedure for searching $\psi$ is shown in Alg.3.

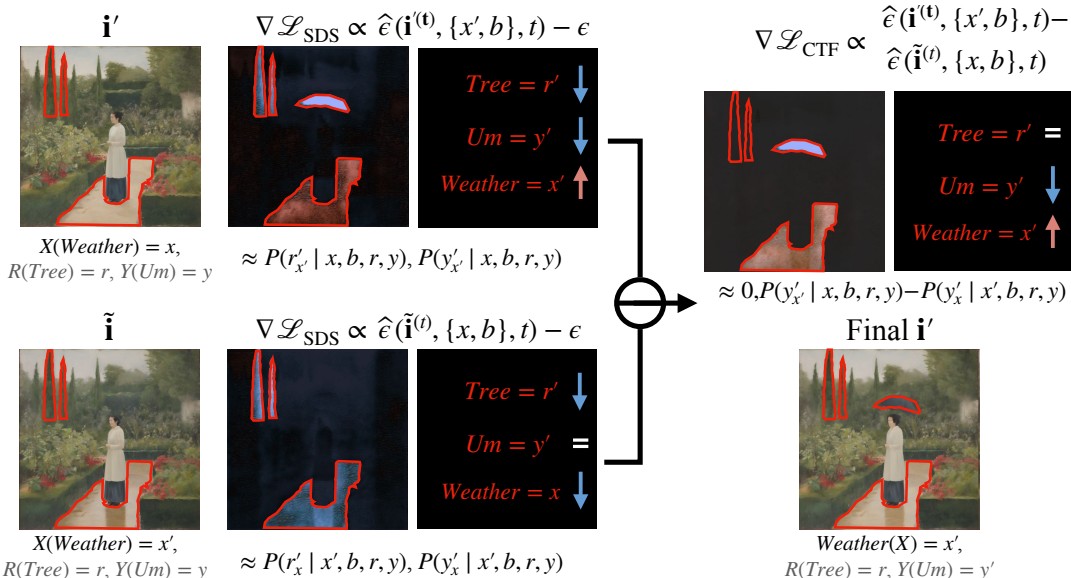

Figure S2: The contracting updating direction $\nabla\mathcal{L}_{\text{CTF}}$. The entanglement of non-descendants are canceled by contrasting while the intervention and effect on descendants are reflected in results.

## C   Omitted Proofs

### C.1   Proof of Thm. 1

**Theorem 1.** *Consider the true underlying ASCM $\mathcal{M}^*$ over $\{\mathbf{V}, \mathbf{I}\}$. Let $\mathbf{ND}$ denote $\cap_{X_i}\mathbf{ND}(X_i)\backslash\mathbf{X}$ (non-descendants of $\mathbf{X}$) in $\mathcal{G}_{\mathbf{V},\mathbf{L}}$ and $\mathbf{DE}$ denote $\cup_{X_i}\mathbf{DE}(X_i)\backslash\mathbf{X}$ (descendants of $\mathbf{X}$) in $\mathcal{G}_{\mathbf{V},\mathbf{L}}$. The target I-ctf query $P^*(\mathbf{I}_{\mathbf{x}'} = \mathbf{i}' \mid \mathbf{I} = \mathbf{i})$ can be factorized as*

$$P^*(\mathbf{I}_{\mathbf{x}'} = \mathbf{i}' \mid \mathbf{I} = \mathbf{i}) = \underbrace{\mathbf{1}[h_{\mathbf{X}}^*(\mathbf{i}') = \mathbf{x}']}_{\textit{Intervention Consistency}} \cdot \underbrace{\mathbf{1}[\mathbf{nd}' = \mathbf{nd}]}_{\textit{Non-descendants Invariance}} \cdot \underbrace{P^*(\mathbf{De}_{\mathbf{x}'} = \mathbf{de}' \mid \mathbf{v}, \mathbf{l})}_{\textit{Amount of Descendant Changing}} \tag{3}$$

*where $\mathbf{nd} = h_{\mathbf{ND}}^*(\mathbf{i})$, $\mathbf{nd}' = h_{\mathbf{ND}}^*(\mathbf{i}')$, $\mathbf{de} = h_{\mathbf{DE}}^*(\mathbf{i})$, and $\{\mathbf{v}, \mathbf{l}\} = h_{\mathbf{V},\mathbf{L}}^*(\mathbf{i})$.* ∎

*Proof.* We will use ctf-calculus [10, Thm 3.1] for solving this counterfactual. Specifically, we will use rule 3, the exclusion rule

$$P(y_{\mathbf{x}\mathbf{z}}, \mathbf{w}*) = P(y_{\mathbf{z}}, \mathbf{w}*) \tag{25}$$

if $\mathbf{X} \cap \mathbf{Anc}(y) = \emptyset$ in $\mathcal{G}_{\overline{\mathbf{X}}}$, here $\mathbf{Anc}(y) = \cup_{Y_i \in y}Anc(Y_i)$. Also, Recall $h_{\mathbf{W}}$ is the inverse mapping from $\mathbf{I}$ to $\mathbf{W}$, for any $\mathbf{W} \subseteq \mathbf{V} \cup \mathbf{L}$.

$$P^*(\mathbf{I}_{\mathbf{x}'} = \mathbf{i}' \mid \mathbf{I} = \mathbf{i}) \tag{26}$$

$$= \sum_{\mathbf{v}'',\mathbf{l}''} P(\mathbf{i}'_{\mathbf{x}'} \mid \mathbf{i}, \mathbf{v}'', \mathbf{l}'')P(\mathbf{v}'', \mathbf{l}'' \mid \mathbf{i}) \qquad\qquad \text{sum over } \mathbf{v}'', \mathbf{l}'' \tag{27}$$

$$= \sum_{\mathbf{v}'',\mathbf{l}''} P(\mathbf{i}'_{\mathbf{x}'} \mid \mathbf{i}, \mathbf{v}'', \mathbf{l}'')\mathbf{1}[\mathbf{v}'', \mathbf{l}'' = h_{\mathbf{V},\mathbf{L}}(\mathbf{i})] \qquad\qquad \text{invertibility} \tag{28}$$

$$= \sum_{\mathbf{x}'',\mathbf{nd}'',\mathbf{de}''} P(\mathbf{i}'_{\mathbf{x}'} \mid \mathbf{x}''_{\mathbf{x}'}, \mathbf{nd}''_{\mathbf{x}'}, \mathbf{de}''_{\mathbf{x}'}\mathbf{v}, \mathbf{l})P(\mathbf{x}''_{\mathbf{x}'}, \mathbf{nd}''_{\mathbf{x}'}, \mathbf{de}''_{\mathbf{x}'} \mid \mathbf{v}, \mathbf{l}) \tag{29}$$

$$= P(\mathbf{X}_{\mathbf{x}'} = \mathbf{x}'', \mathbf{nd}'_{\mathbf{x}'}, \mathbf{de}'_{\mathbf{x}'} \mid \mathbf{v}, \mathbf{l}) \qquad\qquad \text{invertibility} \tag{30}$$

$$= \mathbf{1}[h_{\mathbf{X}}(\mathbf{i}') = \mathbf{x}']P(\mathbf{nd}'_{\mathbf{x}'}, \mathbf{de}'_{\mathbf{x}'} \mid \mathbf{v}, \mathbf{l}) \qquad\qquad \text{intervention definition} \tag{31}$$

$$= \mathbf{1}[h_{\mathbf{X}}(\mathbf{i}') = \mathbf{x}']P(\mathbf{nd}', \mathbf{de}'_{\mathbf{x}'} \mid \mathbf{v}, \mathbf{l}) \qquad \text{Rule 3, } \mathbf{X} \cap \mathbf{Anc}(\mathbf{ND}) = \emptyset \text{ in } \mathcal{G}_{\overline{\mathbf{X}}} \tag{32}$$

$$= \mathbf{1}[h_{\mathbf{X}}(\mathbf{i}') = \mathbf{x}']\mathbf{1}[\mathbf{nd}' = \mathbf{nd}]P(\mathbf{de}'_{\mathbf{x}'} \mid \mathbf{v}, \mathbf{l}) \qquad\qquad \mathbf{ND} \subseteq \mathbf{V} \cup \mathbf{L} \tag{33}$$

□

## C.2 Proof of Thm. 2

We first introduce the following lemma to map a F-ctf query to the generative level.

**Lemma 1.** *Consider a ground truth $\mathcal{M}^*$. For every BD-CLS $\widehat{\mathcal{M}}^{\text{BD-CLS}}$, there exists another $\mathcal{M}'$ over $\{\mathbf{X}, \mathbf{B}, \mathbf{ND}, Y, \mathbf{I}\}$ that induces the same I-ctf distribution $P(\mathbf{I}_{\mathbf{x}'} \mid \mathbf{i})$ and is compatible with $\mathcal{G}'$ shown in Fig. 1, where $Y \in \mathbf{De}$.*

Figure S3: The structure in Lemma 1.

*Proof.* Consider the Def. 4, for any $\widehat{\mathcal{M}}^{\text{BD-CLS}}$, the mixing mechanism is expressed as:

$$\mathbf{I} \leftarrow \widehat{f}_{\mathbf{I}}(\mathbf{X}, \mathbf{B}, \mathbf{Z}) \tag{34}$$

Since $f_{\mathbf{I}}^*$ is invertible $\mathbf{DE} = h_{\mathbf{DE}} \circ \widehat{f}_{\mathbf{I}}(\mathbf{X}, \mathbf{B}, \mathbf{Z})$, where $h_{\mathbf{DE}}$ is the mapping from $\mathbf{I}$ to $\mathbf{DE}$. According to structure condition Def. 4, $\mathbf{Z} \leftarrow \widehat{f}_{\mathbf{Z}}(\mathbf{B}, \mathbf{U}_{\mathbf{Z}})$. According to the disentanglement conditions in Def. 4 state that function $\tau_{\mathbf{ND}} = h_{\mathbf{ND}}^* \circ \widehat{f}_{\mathbf{I}}$ such that

$$\mathbf{ND} = \tau_{\mathbf{ND}}(\mathbf{Z}, \mathbf{B}) \tag{35}$$

Notice that

$$\mathbf{I} = \widehat{f}_{\mathbf{I}}(\mathbf{X}, \mathbf{B}, \mathbf{Z}) = f_{\mathbf{I}}^* \circ h_{\mathbf{V}, \mathbf{L}} \circ f(\mathbf{X}, \mathbf{B}, \mathbf{Z}) \tag{36}$$

Thus, we construct an $\mathcal{M}'$ exact the same as $\widehat{\mathcal{M}}^{\text{BD-CLS}}$ except

$$\mathbf{ND} \leftarrow \tau_{\mathbf{ND}}(f_{\mathbf{Z}}(\mathbf{B}, \mathbf{U}_{\mathbf{Z}}), \mathbf{B}) \tag{37}$$

$$Y \leftarrow h_Y \circ f(f_{\mathbf{Z}}(\mathbf{B}, \mathbf{U}_{\mathbf{Z}}), \mathbf{B}, \mathbf{X}) \tag{38}$$

$$\mathbf{I} \leftarrow \widehat{f}_{\mathbf{I}}(\mathbf{X}, \mathbf{B}, \mathbf{Z}) = f_{\mathbf{I}}^*(\mathbf{X}, \mathbf{B}, \mathbf{ND} \backslash \mathbf{B}, Y, h_{\mathbf{DE} \backslash Y} \circ f(\mathbf{X}, \mathbf{B}, f_{\mathbf{Z}}(\mathbf{B}, \mathbf{U}_{\mathbf{Z}}))) \tag{39}$$

This will lead the same $P(\mathbf{I}_{\mathbf{x}'} \mid \mathbf{i})$ since the mechanism is the same over $\{\mathbf{X}, \mathbf{B}, \mathbf{Z}\}$ and the diagram is shown in Fig. S3 according to the mechanisms. $\square$

Before formally proving Thm. 2. We first state our important assumption and clarify some notation here. We assume that the domains of $\mathbf{X}, Y, \mathbf{ND}$ are discrete and finite. $P(\mathbf{X}, \mathbf{nd}, \mathbf{De})$ denotes all observational quantities that $\mathbf{ND}$ takes $\mathbf{nd}$.

**Theorem 2** (Causal validity of BD-CLS). *Consider an $\widehat{\mathcal{M}}^{\text{BD-CLS}}$ for $\mathcal{M}^*$ and the target query $P^*(\mathbf{i}'_{\mathbf{x}'} \mid \mathbf{i})$. Let $P^{\widehat{\mathcal{M}}^{\text{BD-CLS}}}(\mathbf{i}'_{\mathbf{x}'} \mid \mathbf{i})$ be an estimator for $P(\mathbf{i}'_{\mathbf{x}'} \mid \mathbf{i})$. Then, (a) (intervention) $P^{\widehat{\mathcal{M}}^{\text{BD-CLS}}}(\tilde{\mathbf{x}}_{\mathbf{x}'} \mid \mathbf{v}, \mathbf{l}) = \mathbf{1}[\tilde{\mathbf{x}} = \mathbf{x}']$, where $\tilde{\mathbf{x}} = h_{\mathbf{X}}(\mathbf{i}')$; (b) (non-descendants) $P^{\widehat{\mathcal{M}}^{\text{BD-CLS}}}(\mathbf{nd}'_{\mathbf{x}'} \mid \mathbf{v}, \mathbf{l}) = \mathbf{1}[\mathbf{nd}' = \mathbf{nd}]$, (c) (descendants) $P^{\widehat{\mathcal{M}}^{\text{BD-CLS}}}(\mathbf{de}'_{\mathbf{x}'} \mid \mathbf{v}, \mathbf{l})$ is ctf-consistent with $P^*(\mathbf{de}'_{\mathbf{x}'} \mid \mathbf{v}, \mathbf{l})$ under $\langle P(\mathbf{X}, \mathbf{nd}, \mathbf{De}), \mathcal{G}_{\mathbf{X}, \mathbf{ND}, \mathbf{I}} \rangle$, where $\mathbf{w} = h_{\mathbf{w}}(\mathbf{i}), \mathbf{w}' = h_{\mathbf{w}}(\mathbf{i}')$ for any $\mathbf{W} \subseteq \mathbf{V} \cup \mathbf{L}$.* $\blacksquare$

*Proof.* **(a) intervention.** According to Def. 4 condition (1),

$$P^{\widehat{\mathcal{M}}^{\text{bd-cls}}}(\mathbf{I} \mid \mathbf{x}) = P^*(\mathbf{I} \mid \mathbf{x}) \tag{40}$$

Due to invertibility from $\mathbf{I}$ to $\mathbf{X}$ in $\mathcal{M}^*$ and the conditional distribution match, the invertibility also exists from $\mathbf{I}$ to $\mathbf{X}$ in $\widehat{\mathcal{M}}^{\text{bd-cls}}$. Then $P^{\widehat{\mathcal{M}}^{\text{BD-CLS}}}(\tilde{\mathbf{x}}_{\mathbf{x}'} \mid \mathbf{v}, \mathbf{l}) = \mathbf{1}[\tilde{\mathbf{x}} = \mathbf{x}']$, where $\tilde{\mathbf{x}} = h_{\mathbf{X}}(\mathbf{i}')$.

**(b) non-descendant.** Then we use Lem. 1 to map a F-ctf query to the generative level. According to Def. 2 and $\mathbf{ND} = h_{\mathbf{ND}}(\mathbf{I})$,

$$P^{\widehat{\mathcal{M}}^{\text{bd-cls}}}(\mathbf{nd}'_{\mathbf{x}'} \mid \mathbf{v}, \mathbf{l}) \tag{41}$$

$$= P^{\widehat{\mathcal{M}}^{\text{bd-cls}}}(\mathbf{nd}'_{\mathbf{x}'} \mid \mathbf{i}) \tag{42}$$

$$= \int_{\mathbf{i}'' \in \mathcal{X}_{\mathbf{I}}} \mathbf{1}\left[h_{\mathbf{ND}}^*(\mathbf{i}'') = \mathbf{nd}'\right] dP^{\widehat{\mathcal{M}}^{\text{bd-cls}}}(\mathbf{i}) \tag{43}$$

$$= P^{\mathcal{M}'}(\mathbf{nd}'_{\mathbf{x}'} \mid \mathbf{i}) \qquad\qquad P^{\widehat{\mathcal{M}}^{\text{bd-cls}}}(\mathbf{I}_{\mathbf{x}'} \mid \mathbf{i}) = P^{\mathcal{M}'}(\mathbf{I}_{\mathbf{x}'} \mid \mathbf{i}), \text{ Lem. 1} \tag{44}$$

$$\tag{45}$$

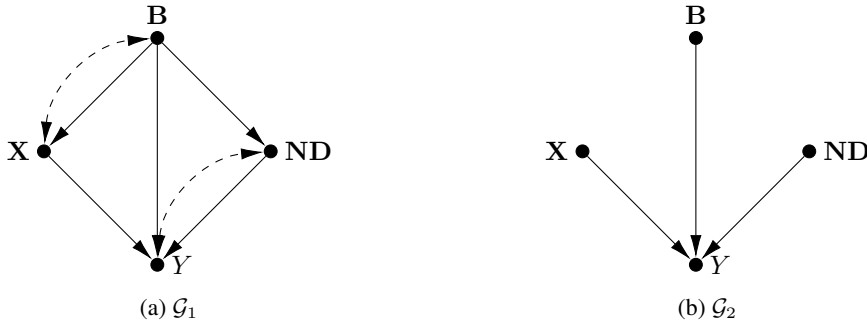

Figure S4: Diagrams for two proxy SCMs in Proof for Thm. 2.

Since $\mathbf{ND}$ in $\mathcal{M}'$ is also non-descendant of $\mathbf{X}$, according to proof of Thm. 1,

$$P^{\mathcal{M}'}(\mathbf{nd}'_{\mathbf{x}'} \mid \mathbf{i}) = \mathbf{1}[\mathbf{nd}' = \mathbf{nd}] \tag{46}$$

$$\tag{47}$$

**(c) descendant.** According to Def. 3, now the proof goal is to show the bound of F-ctf query $P^{\widehat{\mathcal{M}}^{\text{bd-cls}}}(\mathbf{de}'_{\mathbf{x}'} \mid \mathbf{x}, \mathbf{nd}, \mathbf{de})$ (given $\mathcal{G}^{\widehat{\mathcal{M}}^{\text{bd-cls}}}$ and $P(\mathbf{V}, \mathbf{I})$) is in the bound of query $P^*(\mathbf{de}'_{\mathbf{x}'} \mid \mathbf{x}, \mathbf{nd}, \mathbf{de})$ (given $\mathcal{G}_{\mathbf{X}, \mathbf{ND}, \mathbf{I}}$ and $P(\mathbf{X}, \mathbf{nd}, \mathbf{DE})$). Leveraging Lem.1, we can map a F-ctf query to the generative level.

Following the same procedure of the mapping of an F-ctf query above using the fact $Y = h_Y(\mathbf{I})$:

$$P^{\widehat{\mathcal{M}}^{\text{bd-cls}}}(\mathbf{de}'_{\mathbf{x}'} \mid \mathbf{x}, \mathbf{nd}, \mathbf{de}) \tag{48}$$

$$= P^{\widehat{\mathcal{M}}^{\text{bd-cls}}}(\mathbf{de}'_{\mathbf{x}'} \mid \mathbf{i}) \tag{49}$$

$$= \frac{\int_{\mathbf{i}, \mathbf{i}' \in \mathcal{X}_{\mathbf{I}}} \mathbf{1}\left[h_Y^*(\mathbf{i}') = y', h_{\mathbf{ND}}^*(\mathbf{i}) = \mathbf{nd}\right] dP^{\widehat{\mathcal{M}}^{\text{bd-cls}}}(\mathbf{i}, \mathbf{i}'_{\mathbf{x}'})}{\int_{\mathbf{i} \in \mathcal{X}_{\mathbf{I}}} \mathbf{1}\left[h_{\mathbf{ND}}^*(\mathbf{i}) = \mathbf{nd}\right] dP^{\widehat{\mathcal{M}}^{\text{bd-cls}}}(\mathbf{i})} \tag{50}$$

$$= P^{\mathcal{M}'}(\mathbf{de}'_{\mathbf{x}'} \mid \mathbf{x}, \mathbf{nd}, \mathbf{de}) \qquad \text{Lem. 1 and Eq. 1} \tag{51}$$

$$\tag{52}$$

Now the goal is to proof $P^{\mathcal{M}'}(\mathbf{de}'_{\mathbf{x}'} \mid \mathbf{x}, \mathbf{nd}, \mathbf{de})$ (given $\mathcal{G}^{\mathcal{M}'}$ and $P(\mathbf{X}, nd, \mathbf{I})$) is in the bound of query $P^*(\mathbf{de}'_{\mathbf{x}'} \mid \mathbf{x}, \mathbf{nd}, \mathbf{de})$ (given $\mathcal{G}_{\mathbf{X}, \mathbf{ND}, \mathbf{I}}$ and $P(\mathbf{X}, nd, \mathbf{I})$). Since $P^{\mathcal{M}^*}(\mathbf{I} \mid \mathbf{x}, \mathbf{b}) = P^{\widehat{\mathcal{M}}^{\text{bd-cls}}}(\mathbf{I} \mid \mathbf{x}, \mathbf{b}) = P^{\mathcal{M}'}(\mathbf{I} \mid \mathbf{x}, \mathbf{b})$,

$$P^{\mathcal{M}^*}(\mathbf{DE}, \mathbf{ND} \mid \mathbf{X}, \mathbf{B}) = P^{\mathcal{M}'}(\mathbf{DE}, \mathbf{ND} \mid \mathbf{X}, \mathbf{B}) \tag{53}$$

To illustrate, this means that the observational distributions over $\{\mathbf{DE}, \mathbf{ND}, \mathbf{X}, \mathbf{B}\}$ are equivalent between $\mathcal{M}$ and $\mathcal{M}'$. However the graph between them are different, i.e., $\mathcal{G}^* \neq \mathcal{G}'$. Thus, the proof goal now is show the bound of the same query given same observational distribution but different graph. We will prove that the bound is the same as given $\mathcal{G}_1$ and the diagram $\mathcal{G}_2$ shown in Fig. S4 if the observational distribution can be matched between $\mathcal{M}(\mathcal{G}_2)$ and $\mathcal{M}(\mathcal{G}_1)$. To simplify the notation, we simplify $\mathbf{DE}$ as $Y$ since $\mathcal{G}_{\mathbf{X}, \mathbf{ND}, \mathbf{I}}$ does not restrict any structure among $\mathbf{DE}$.

Denote the two bounds as $\mathcal{B}_{\mathcal{G}_1}$ and $\mathcal{B}_{\mathcal{G}_2}$. $\mathcal{B}_{\mathcal{G}_2} \subseteq \mathcal{B}_{\mathcal{G}_1}$ since $\mathcal{G}_2$ is a subgraph of $\mathcal{G}_1$ thus $\mathcal{M}(\mathcal{G}_2)$ satisfy all constraints induced in $\mathcal{M}(\mathcal{G}_1)$.

We prove $\mathcal{B}_{\mathcal{G}_1} \subseteq \mathcal{B}_{\mathcal{G}_2}$ by proving that for every $\mathcal{M}_1$ that induces $\mathcal{G}_1$, we can find another $\mathcal{M}_2$ that induces $\mathcal{G}_2$ such that $P^{\mathcal{M}_1}(y'_{\mathbf{x}'} \mid \mathbf{x}, \mathbf{b}, \mathbf{nd}, y) = P^{\mathcal{M}_2}(y'_{\mathbf{x}'} \mid \mathbf{x}, \mathbf{b}, \mathbf{nd}, y)$. Formally, We will use the spirit of canonical SCM [61, Def. 2.2] for expressing the $P^{\mathcal{M}_1}(y'_{\mathbf{x}'} \mid y, \mathbf{nd}, \mathbf{x}, \mathbf{b})$ and $P^{\mathcal{M}_2}(y'_{\mathbf{x}'} \mid y, \mathbf{nd}, \mathbf{x}, \mathbf{b})$. Denote the domain of $\mathbf{X}$ as $\{\mathbf{x}_1, \mathbf{x}_2, \ldots, \mathbf{x}_{d_{\mathbf{x}}}\}$, the domain of $\mathbf{B}$ as $\{\mathbf{b}_1, \mathbf{b}_2, \ldots, \mathbf{b}_{d_{\mathbf{b}}}\}$, the domain of $\mathbf{ND}$ as $\{\mathbf{nd}_1, \mathbf{nd}_2, \ldots, \mathbf{nd}_{d_{\mathbf{nd}}}\}$, the domain of $Y$ as $\{y_1, y_2, \ldots, y_{d_y}\}$.

Consider the function class $\mathcal{F}_1^Y$

$$
\begin{aligned}
\mathcal{F}^Y = \{ & \{y_1 \leftarrow \{\mathbf{x}_1, \mathbf{b}_1, \mathbf{nd}_1\}, y_1 \leftarrow \{\mathbf{x}_1, \mathbf{b}_1, \mathbf{nd}_2\}, \ldots, y_1 \leftarrow \{\mathbf{x}_{d_\mathbf{x}}, \mathbf{b}_{d_\mathbf{b}}, \mathbf{nd}_{d_\mathbf{nd}}\} \\
& \{y_1 \leftarrow \{\mathbf{x}_1, \mathbf{b}_1, \mathbf{nd}_1\}, y_1 \leftarrow \{\mathbf{x}_1, \mathbf{b}_1, \mathbf{nd}_2\}, \ldots, y_2 \leftarrow \{\mathbf{x}_{d_\mathbf{x}}, \mathbf{b}_{d_\mathbf{b}}, \mathbf{nd}_{d_\mathbf{nd}}\} \\
& \cdots , \\
& \{y_{d_y} \leftarrow \{\mathbf{x}_1, \mathbf{b}_1, \mathbf{nd}_1\}, y_{d_y} \leftarrow \{\mathbf{x}_1, \mathbf{b}_1, \mathbf{nd}_2\}, \ldots, y_{d_y} \leftarrow \{\mathbf{x}_{d_\mathbf{x}}, \mathbf{b}_{d_\mathbf{b}}, \mathbf{nd}_{d_\mathbf{nd}}\} \\
& \}
\end{aligned}
\tag{54}
$$

and the function class $\mathcal{F}_1^{\mathbf{ND}}$:

$$
\begin{aligned}
\mathcal{F}^Y = \{ & \{\mathbf{nd}_1 \leftarrow \{\mathbf{b}_1, \}, \mathbf{nd}_1 \leftarrow \{\mathbf{b}_2\}, \ldots, \mathbf{nd}_1 \leftarrow \{\mathbf{b}_{d_\mathbf{b}}\} \\
& \{\mathbf{nd}_1 \leftarrow \{\mathbf{b}_1, \}, \mathbf{nd}_1 \leftarrow \{\mathbf{b}_2\}, \ldots, \mathbf{nd}_2 \leftarrow \{\mathbf{b}_{d_\mathbf{b}}\} \\
& \cdots , \\
& \{\mathbf{nd}_{d_\mathbf{nd}} \leftarrow \{\mathbf{b}_1, \}, \mathbf{nd}_{d_\mathbf{nd}} \leftarrow \{\mathbf{b}_2\}, \ldots, \mathbf{nd}_{d_\mathbf{nd}} \leftarrow \{\mathbf{b}_{d_\mathbf{b}}\} \\
& \}
\end{aligned}
\tag{55}
$$

Consider $f_Y^{\mathcal{M}_1}$ and $f_{\mathbf{ND}}^{\mathcal{M}_1}$ in the canonical type for $\mathcal{M}_1$

$$
Y \leftarrow f_Y^{\mathcal{M}_1} = f_Y^{\text{canonical}}(\mathbf{X}, \mathbf{B}, R) \tag{56}
$$

$$
\mathbf{ND} \leftarrow f_{\mathbf{ND}}^{\mathcal{M}_1} = f_{\mathbf{ND}}^{\text{canonical}}(\mathbf{B}, R) \tag{57}
$$

where the domain of $R$ are discrete values $\{r_{f^{\mathbf{ND}}, f^Y}\}_{f^{\mathbf{ND}} \in \mathcal{F}^{\mathbf{ND}}, f^Y \in \mathcal{F}^Y}$. Let, $f_Y^{\text{canonical}}(\mathbf{x}, \mathbf{b}, r_{f^{\mathbf{ND}}, f^Y} = 1) = f^Y(\mathbf{X}, \mathbf{B})$ and $f_{\mathbf{ND}}^{\text{canonical}}(\mathbf{x}, \mathbf{b}, r_{f^{\mathbf{ND}}, f^Y} = 1) = f^{\mathbf{ND}}(\mathbf{B})$. For every $\mathcal{M}'$, the functions $f_Y'$ and $f_{\mathbf{ND}}'$ can be expressed in the above way. Then:

$$
Q_1^1 = P^{\mathcal{M}_1}(y_{\mathbf{x}'}' \mid y, \mathbf{nd}, \mathbf{x}, \mathbf{b}) = \frac{\sum_{y' = f^Y(\mathbf{x}', \mathbf{b}, \mathbf{nd}), y = f^Y(\mathbf{x}, \mathbf{b}, \mathbf{nd}), \mathbf{nd} = f^{\mathbf{ND}}(\mathbf{b})} P(r_{f^Y, f^{\mathbf{ND}}} = 1)}{P(\mathbf{nd}, y \mid \mathbf{x}, \mathbf{b})}
\tag{58}
$$

and the conditional observational distribution can be expressed as:

$$
Q_2^1 = P^{\mathcal{M}_1}(y, \mathbf{nd} \mid \mathbf{x}, \mathbf{b}) = \sum_{y = f^Y(\mathbf{x}, \mathbf{b}, \mathbf{nd}), \mathbf{nd} = f^{\mathbf{ND}}(\mathbf{b})} P(r_{f^Y, f^{\mathbf{ND}}}) \tag{59}
$$

and

$$
Q_3^1 = P^{\mathcal{M}_1}(y', \mathbf{nd} \mid \mathbf{x}', \mathbf{b}) = \sum_{y' = f^Y(\mathbf{x}', \mathbf{b}, \mathbf{nd}), \mathbf{nd} = f^{\mathbf{ND}}(\mathbf{b})} P(r_{f^Y, f^{\mathbf{ND}}}) \tag{60}
$$

For $\mathcal{M}_2$, consider the same function class

$$
\begin{aligned}
\mathcal{F}^Y = \{ & \{y_1 \leftarrow \{\mathbf{x}_1, \mathbf{b}_1, \mathbf{nd}_1\}, y_1 \leftarrow \{\mathbf{x}_1, \mathbf{b}_1, \mathbf{nd}_2\}, \ldots, y_1 \leftarrow \{\mathbf{x}_{d_\mathbf{x}}, \mathbf{b}_{d_\mathbf{b}}, \mathbf{nd}_{d_\mathbf{nd}}\} \\
& \{y_1 \leftarrow \{\mathbf{x}_1, \mathbf{b}_1, \mathbf{nd}_1\}, y_1 \leftarrow \{\mathbf{x}_1, \mathbf{b}_1, \mathbf{nd}_2\}, \ldots, y_2 \leftarrow \{\mathbf{x}_{d_\mathbf{x}}, \mathbf{b}_{d_\mathbf{b}}, \mathbf{nd}_{d_\mathbf{nd}}\} \\
& \cdots , \\
& \{y_{d_y} \leftarrow \{\mathbf{x}_1, \mathbf{b}_1, \mathbf{nd}_1\}, y_{d_y} \leftarrow \{\mathbf{x}_1, \mathbf{b}_1, \mathbf{nd}_2\}, \ldots, y_{d_y} \leftarrow \{\mathbf{x}_{d_\mathbf{x}}, \mathbf{b}_{d_\mathbf{b}}, \mathbf{nd}_{d_\mathbf{nd}}\} \\
& \}
\end{aligned}
\tag{61}
$$

But with a different canonical model

$$
Y \leftarrow f_Y^{\mathcal{M}_2} = f_Y^{\text{canonical}}(\mathbf{X}, \mathbf{B}, S) \tag{62}
$$

$$
\tag{63}
$$

where the domain of $R$ are discrete values $\{S_{f^Y}\}_{f^Y \in \mathcal{F}^Y}$. For a given canonical $\mathcal{M}_1$, the counterfactual quantity is

$$
Q_1^2 = P^{\mathcal{M}_2}(y_{\mathbf{x}'}' \mid y, \mathbf{nd}, \mathbf{x}, \mathbf{b}) = \frac{\sum_{y' = f^Y(\mathbf{x}', \mathbf{b}, \mathbf{nd}), y = f^Y(\mathbf{x}, \mathbf{b}, \mathbf{nd})} P(s_{f^Y} = 1)}{P(y \mid \mathbf{x}, \mathbf{b}, \mathbf{nd})}
\tag{64}
$$

and the condition observational distribution:

$$
Q_2^2 = P^{\mathcal{M}_2}(y \mid \mathbf{nd}, \mathbf{x}, \mathbf{b}) = \sum_{y = f^Y(\mathbf{x}, \mathbf{b}, \mathbf{nd})} P(s_{f^Y} = 1) \tag{65}
$$

and

$$Q_3^2 = P^{\mathcal{M}_2}(y' \mid \mathbf{nd}, \mathbf{x}', \mathbf{b}) = \sum_{y' = f^Y(\mathbf{x}', \mathbf{b}, \mathbf{nd})} P(s_{f^Y} = 1) \tag{66}$$

We set $Q_1^2$ to be equivalent to $Q_1^1$, namely,

$$\sum_{y' = f^Y(\mathbf{x}', \mathbf{b}, \mathbf{nd}), y = f^Y(\mathbf{x}, \mathbf{b}, \mathbf{nd})} P(s_{f^Y} = 1) \tag{67}$$

$$= \sum_{y' = f^Y(\mathbf{x}', \mathbf{b}, \mathbf{nd}), y = f^Y(\mathbf{x}, \mathbf{b}, \mathbf{nd}), \mathbf{nd} = f^{\mathbf{ND}}(\mathbf{b})} P(r_{f^Y, f^{\mathbf{ND}}} = 1) \frac{P(y \mid \mathbf{x}, \mathbf{b}, \mathbf{nd})}{P(\mathbf{nd}, y \mid \mathbf{x}, \mathbf{b})} \tag{68}$$

$$= \frac{\sum_{y' = f^Y(\mathbf{x}', \mathbf{b}, \mathbf{nd}), y = f^Y(\mathbf{x}, \mathbf{b}, \mathbf{nd}), \mathbf{nd} = f^{\mathbf{ND}}(\mathbf{b})} P(r_{f^Y, f^{\mathbf{ND}}} = 1)}{P(\mathbf{nd} \mid \mathbf{x}, \mathbf{b})} \tag{69}$$

$$\tag{70}$$

This set is feasible due to the following reason. First, the observation constrain $P(y, \mid \mathbf{x}, \mathbf{b}, \mathbf{nd})$ $(Q_2^1, Q_2^2)$and $P(y', \mid \mathbf{x}', \mathbf{b}, \mathbf{nd})$ $(Q_3^1, Q_3^2)$ are satisfied. The reason is that all summed term $P(s)$ in $Q_1^2$ are strictly subsets of $P(s)$ in $Q_2^2$ and $P(s)$ in $Q_3^2$; all summed term $P(r)$ in $Q_1^1$ are strictly subsets of $P(r)$ in $Q_2^1$ and $P(r)$ in $Q_3^1$; $Q_2^1 = Q_2^2$; $Q_3^1 = Q_3^2$. Setting these sub-terms will not violate the sum. Second, this will not violate the observational constraints for any $P(y'', \mid \mathbf{x}'', \mathbf{b}, \mathbf{nd})$, where $\{y'' \neq y, \mathbf{x}'' \neq \mathbf{x}\}$ or $\{y'' \neq y', \mathbf{x}'' \neq \mathbf{x}'\}$. For all other observation quantity $P(y'', \mid \mathbf{x}'', \mathbf{b}, \mathbf{nd})$, there is no $P(S)$ in $Q_1^2$ and $P(R)$ in $Q_1^1$ belongs to them. Third, this will not violate the observational constraints for any $P(y'', \mid \mathbf{x}'', \mathbf{b}'', \mathbf{nd}'')$ for $\mathbf{b}'' \neq \mathbf{b}$ and $\mathbf{nd}'' \neq \mathbf{nd}$. Any terms in $Q_1^1$ and $Q_1^2$ are partially summed into these quantities. To construct all terms in $Q_1^1$, $P(y'', \mid \mathbf{x}'', \mathbf{b}'', \mathbf{nd}'')$ must be sum for all $y'' \in \text{Domain}(Y)$. Then $Q_1^1$ satisfies

$$Q_1^1 \leq \sum_{y''} P(y'', \mid \mathbf{x}'', \mathbf{b}'', \mathbf{nd}'') = 1 \tag{71}$$

From this construction, we know the bound is the same as given $\mathcal{G}_1$ and the diagram $\mathcal{G}_2$ shown in Fig. S4. And similarly, if any edge from $\mathbf{B}$ and $\mathbf{N}D$ into $Y$ is missing in $\mathcal{G}_2$ compared to the true diagram $\mathcal{G}^*$, but $\mathcal{M}(\mathcal{G}^*)$ and $\mathcal{M}(\mathcal{G}_2)$ are capable of inducing the same observational distribution, the bound will be the same. Then we conclude $P^{\mathcal{M}'}(y'_{\mathbf{x}'} \mid y, \mathbf{nd})$ (given $\mathcal{G}^{\mathcal{M}'}$ and $P(\mathbf{X}, \mathbf{nd}, Y)$) is in the bound of query $P^*(y'_{\mathbf{x}'} \mid y, \mathbf{nd}, \mathbf{x})$ since the bound of $P^{\mathcal{M}'}(y'_{\mathbf{x}'} \mid y, \mathbf{nd}, \mathbf{x})$ (given $\mathcal{G}^{\mathcal{M}'}$ is the same with the bound of $P^*(y'_{\mathbf{x}'} \mid y, \mathbf{nd}, \mathbf{x})$. □

### C.3 Proof of Prop. 1

We first list the important assumption about the pretrained model. We assume the pretrained model $\mathcal{M}^{\text{SD}}$ matches perfectly the conditional distribution $P^*(\mathbf{I} \mid \mathbf{X}, \mathbf{B})$, i.e.,

$$P^{\mathcal{M}^{\text{SD}}}(\mathbf{I} \mid \mathbf{x}'', \mathbf{b}'') = P^*(\mathbf{I} \mid \mathbf{x}'', \mathbf{b}'') \tag{72}$$

for every $\mathbf{x}''$ and $\mathbf{b}''$.

**Proposition 1 (Sampling I-ctf instances through SD model).** *Consider a ground-truth ASCM $\mathcal{M}^*$ over $\{\mathbf{V}, \mathbf{I}\}$, the target $P^*(\mathbf{I}_{\mathbf{x}'} \mid \mathbf{i})$ and a pre-trained SD model $\widehat{\mathcal{M}}^{\text{SD}}$. Suppose there exists a pair of transformations satisfying $\mathbf{Z} = \psi_1(\mathbf{X}, \mathbf{B}, \mathbf{N})$ and $\mathbf{N} = \psi_2(\mathbf{X}, \mathbf{B}, \mathbf{Z})$. Denotes $\psi(\mathbf{x}, \mathbf{x}', \mathbf{b}, \mathbf{n}) = f^{\text{SD}}(\mathbf{x}', \mathbf{b}, \psi_2(\mathbf{x}', \mathbf{b}, \psi_1(\mathbf{x}, \mathbf{b}, \mathbf{n})))$. Then there exists a BD-CLS $\mathcal{M}^{\text{BD-CLS}}$ over $\{\mathbf{X}, \mathbf{B}, \mathbf{Z}\}$ such that the causal validity in Thm. 2 are also offered by $\sum_{\mathbf{n}} P^{\widehat{\mathcal{M}}^{\text{SD}}}(\mathbf{n} \mid \mathbf{i}, \mathbf{x}, \mathbf{b}) \mathbf{1}[\mathbf{i}' = \psi(\mathbf{x}, \mathbf{x}', \mathbf{b}, \mathbf{n})]$, if*

$$\psi(\mathbf{x}, \mathbf{x}', b, \mathbf{N}) \sim P(\mathbf{I} \mid \mathbf{x}', b), \quad and \quad h^*_{\mathbf{ND}}(\psi(\mathbf{x}, \mathbf{x}', b, \mathbf{n})) = h^*_{\mathbf{ND}}(\mathbf{i}). \qquad \blacksquare \tag{5}$$

*Proof.* let $\mathbf{x} = h_{\mathbf{X}}(\mathbf{i})$ and $\mathbf{b} = h_{\mathbf{B}}(\mathbf{i})$. First, all samples $\mathbf{i}'$ from $\sum_{\mathbf{n}} P^{\widehat{\mathcal{M}}^{\text{SD}}}(\mathbf{n} \mid \mathbf{i}, \mathbf{x}, \mathbf{b}) \mathbf{1}[\mathbf{i}' = \psi(\mathbf{x}, \mathbf{x}', \mathbf{b}, \mathbf{n})]$ satisfied the interventional consistency validity, which means $\mathbf{x}' = h^*_{\mathbf{X}}(\mathbf{i})$. This is because $P^{\mathcal{M}^{\text{SD}}}(\mathbf{I} \mid \mathbf{x}'', \mathbf{b}'') = P^*(\mathbf{I} \mid \mathbf{x}'', \mathbf{b}'')$ and

$$\begin{cases} P^*(\mathbf{i}' \mid \mathbf{x}'', \mathbf{b}'') = 0, & \text{if } \mathbf{x}'' \neq h^*_{\mathbf{X}}(\mathbf{i}'), \\ P^*(\mathbf{i}' \mid \mathbf{x}'', \mathbf{b}'') = 1, & \mathbf{x}'' = h^*_{\mathbf{X}}(\mathbf{i}') \end{cases} \tag{73}$$

Second, all samples $\mathbf{i}'$ from $\sum_{\mathbf{n}} P^{\widehat{\mathcal{M}}^{\text{SD}}}(\mathbf{n} \mid \mathbf{i}, \mathbf{x}, \mathbf{b})\mathbf{1}[\mathbf{i}' = \psi(\mathbf{x}, \mathbf{x}', b, \mathbf{n})]$ satisfied the non-descendant validity since

$$h^*_{\mathbf{ND}}(\mathbf{i}') = h^*_{\mathbf{ND}}(\psi(\mathbf{x}, \mathbf{x}', b, \mathbf{n})) = h^*_{\mathbf{ND}}(\mathbf{i}) \tag{74}$$

Then we will prove that all samples $\mathbf{i}'$ from $\sum_{\mathbf{n}} P^{\widehat{\mathcal{M}}^{\text{SD}}}(\mathbf{n} \mid \mathbf{i}, \mathbf{x}, \mathbf{b})\mathbf{1}[\mathbf{i}' = \psi(\mathbf{x}, \mathbf{x}', \mathbf{b}, \mathbf{n})]$ also satisfies descendant validity. since $\mathcal{M}^{\text{SD}}$ matches the observational distribution $P(\mathbf{I} \mid \mathbf{x}, \mathbf{b})$, $P^{\mathcal{M}^{\text{SD}}}(\mathbf{x}, \mathbf{b} \mid \mathbf{I}) = 1$ due to the invertibility. Then,

$$P^{\mathcal{M}^{\text{SD}}}(\mathbf{n}, \mathbf{x}, \mathbf{b} \mid \mathbf{I}) = P^{\mathcal{M}^{\text{SD}}}(\mathbf{n} \mid \mathbf{I}, \mathbf{x}, \mathbf{b})P(\mathbf{x}, \mathbf{b} \mid \mathbf{I}) \tag{75}$$

$$= P^{\mathcal{M}^{\text{SD}}}(\mathbf{n} \mid \mathbf{I}, \mathbf{x}, \mathbf{b}) \tag{76}$$

Let $\widehat{\mathcal{M}}$ be an SCM over $\{\mathbf{X}, \mathbf{B}, \mathbf{Z}\}$ and compatible with $\mathcal{G}$ in Fig. 4.

$$\sum_{\mathbf{n}} P^{\widehat{\mathcal{M}}^{\text{SD}}}(\mathbf{n} \mid \mathbf{i}, \mathbf{x}, \mathbf{b})\mathbf{1}[\mathbf{i}' = \psi(\mathbf{x}, \mathbf{x}', \mathbf{b}, \mathbf{n})] \tag{77}$$

$$= \sum_{\mathbf{n}} P^{\widehat{\mathcal{M}}^{\text{SD}}}(\mathbf{n} \mid \mathbf{i})P^{\widehat{\mathcal{M}}^{\text{SD}}}(f^{\text{SD}}(\mathbf{x}', \mathbf{b}, \psi_2(\psi_1(\mathbf{x}, \mathbf{b}, \mathbf{n}), \mathbf{x}', \mathbf{b})) = \mathbf{i}') \tag{78}$$

$$= \sum_{\mathbf{n}} P^{\widehat{\mathcal{M}}^{\text{SD}}}(\mathbf{n} \mid \mathbf{i})P^{\widehat{\mathcal{M}}^{\text{SD}}}(f^{\text{SD}}(\mathbf{x}', \mathbf{b}, \psi_2(\psi_1(\mathbf{x}, \mathbf{b}, \mathbf{n}), \mathbf{x}', \mathbf{b})) = \mathbf{i}') \tag{79}$$

$$= \sum_{\mathbf{n}, \mathbf{x}'', \mathbf{b}''} P^{\widehat{\mathcal{M}}^{\text{SD}}}(\mathbf{n}, \mathbf{x}'', \mathbf{b}'' \mid \mathbf{i})P^{\widehat{\mathcal{M}}^{\text{SD}}}(f^{\text{SD}}(\mathbf{x}', \mathbf{b}, \psi_2(\psi_1(\mathbf{x}, \mathbf{b}, \mathbf{n}), \mathbf{x}', \mathbf{b})) = \mathbf{i}') \tag{80}$$

$$= \sum_{\mathbf{z}} P^{\widehat{\mathcal{M}}}(\mathbf{z} \mid \mathbf{i})P^{\widehat{\mathcal{M}}}(f^{\text{SD}}(\mathbf{x}', \mathbf{b}, \psi_2(\mathbf{z}, \mathbf{x}', \mathbf{b})) = \mathbf{i}' \mid \mathbf{x}, \mathbf{b}) \tag{81}$$

$$= \sum_{\mathbf{z}} P^{\widehat{\mathcal{M}}}(\mathbf{z}, \mathbf{x}, \mathbf{b} \mid \mathbf{i})P^{\widehat{\mathcal{M}}^{\text{BD-CLS}}}(\mathbf{i}_{\mathbf{x}'} \mid \mathbf{x}, \mathbf{b}, \mathbf{z}) \tag{82}$$

$$= P^{\widehat{\mathcal{M}}}(\mathbf{I}_{\mathbf{X}=\mathbf{x}'} = \mathbf{i}' \mid \mathbf{i}) \tag{83}$$

Since $\psi(\mathbf{x}, \mathbf{x}', b, \mathbf{N}) \sim P^*(\mathbf{I} \mid \mathbf{x}', b)$ and $f^{\text{SD}}(\mathbf{x}, \mathbf{b}, \psi_2(\psi_1(\mathbf{x}, \mathbf{b}, \mathbf{N}), \mathbf{x}, \mathbf{b})) = f^{\text{SD}}(\mathbf{x}, \mathbf{b}, \mathbf{N})$, which implies $P^{\widehat{\mathcal{M}}}(\mathbf{I} \mid \mathbf{X}, \mathbf{b})$. Then $\widehat{\mathcal{M}}$ is a BD-CLS offering escendant validity. $\square$

### C.4 Proof of Prop. 2

**Proposition 2** (**Toy entanglement between binary** $X$**,** $Y$ **and** $R$). *Consider binary $X$, non-descendant $R$ and descendant $Y$. Suppose $P^*(y \mid \mathbf{pa}_Y) \neq P^*(y' \mid \mathbf{pa}_Y)$. Suppose $R$ and $Y$ are both entangled with $\{X, \mathbf{B}, \mathbf{N}\}$ in $\mathcal{M}^{\text{SD}}$, and $R = \tau_R(X, \mathbf{B}, \mathbf{N}_1)$ and $Y = \tau_Y(X, \mathbf{B}, \mathbf{N}_2)$, where $\mathbf{N}_1 \cap \mathbf{N}_2 = \emptyset$ and $\mathbf{N}_1, \mathbf{N}_2 \subseteq \mathbf{N}$, then $P^{\widehat{\mathcal{M}}^{\text{SD}}}(r'_{x'} \mid x, \mathbf{b}, r, y) = P^{\widehat{\mathcal{M}}^{\text{SD}}}(r'_x \mid x', \mathbf{b}, r, y)$ and $P^{\widehat{\mathcal{M}}^{\text{SD}}}(y'_{x'} \mid x, \mathbf{b}, r, y) \neq P^{\widehat{\mathcal{M}}^{\text{SD}}}(y'_x \mid x', \mathbf{b}, r, y)$.* $\blacksquare$

*Proof.* Construct an SCM $\widehat{\mathcal{M}}$ over $\{X, \mathbf{B}, R, Y\}$ with $f_R = \tau_R$, $f_Y = \tau_Y$ and $\mathbf{U}_R = \mathbf{N}_1$, $\mathbf{U}_R = \mathbf{N}_2$.
Then

$$P^{\widehat{\mathcal{M}}^{SD}}(w'_{x'} \mid x, \mathbf{b}, r, y) = P^{\widehat{\mathcal{M}}}(w'_{x'} \mid x, \mathbf{b}, r, y) \tag{84}$$

for any $W \in \{R, Y\}$. With fact $\mathbf{N}_1 \perp \mathbf{N}_2$,

$$P^{\widehat{\mathcal{M}}}(r'_{x'} \mid x, \mathbf{b}, r, y) = P^{\widehat{\mathcal{M}}}(r'_{x'} \mid x, \mathbf{b}, r) \tag{85}$$

and

$$P^{\widehat{\mathcal{M}}}(y'_{x'} \mid x, \mathbf{b}, y) = P^{\widehat{\mathcal{M}}}(y'_{x'} \mid x, \mathbf{b}, y) \tag{86}$$

Since $W_X \perp X$ for any $W \in \{R, Y\}$ in $\widehat{\mathcal{M}}$

$$P^{\widehat{\mathcal{M}}}(w_{x'}, w_x \mid \mathbf{b}) = P^{\widehat{\mathcal{M}}}(w_{x'}, w \mid x, \mathbf{b}) = P^{\widehat{\mathcal{M}}}(w, w_x \mid x', \mathbf{b}) \tag{87}$$

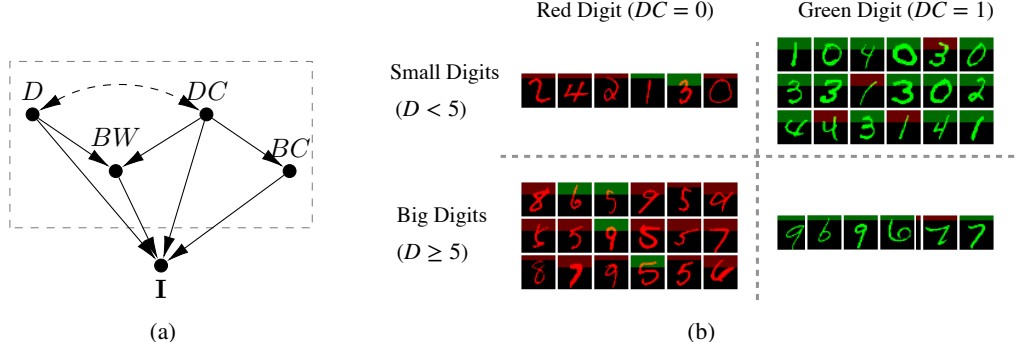

Figure S5: The causal diagram and samples from the ground truth generation process in Colored MNIST and Bars experiments.

When $P(w \mid x, b) = P(w \mid x', b)$,

$$\frac{P^{\widehat{\mathcal{M}}}(w_{x'}, w \mid x, \mathbf{b})}{P(w \mid x, b)} = \frac{P^{\widehat{\mathcal{M}}}(w, w_x \mid x', \mathbf{b})}{P(w \mid x', b)} \tag{88}$$

$$P^{\widehat{\mathcal{M}}}(w_{x'} \mid x, b, w) = P^{\widehat{\mathcal{M}}}(w_x \mid x', b, w) \tag{89}$$

$$1 - P^{\widehat{\mathcal{M}}}(w_{x'} \mid x, b, w) = 1 - P^{\widehat{\mathcal{M}}}(w_x \mid x', b, w) \tag{90}$$

$$P^{\widehat{\mathcal{M}}}(w'_{x'} \mid x, b, w) = P^{\widehat{\mathcal{M}}}(w'_x \mid x', b, w) \tag{91}$$

Since $\mathbf{B}$ is the backdoor set that $X \perp R \mid \mathbf{B}$ in $\mathcal{M}^*$, we have $P(r \mid x, b) = P(r \mid x', b)$, then $P^{\widehat{\mathcal{M}}}(r'_{x'} \mid x, b, r) = P^{\widehat{\mathcal{M}}}(r'_x \mid x', b, r)$.

On the other hand, since $P(y \mid x, \mathbf{b}) \neq P(y \mid x', \mathbf{b})$,

$$P^{\widehat{\mathcal{M}}}(y'_{x'} \mid x, b, y) \neq P^{\widehat{\mathcal{M}}}(y'_x \mid x', b, y) \tag{92}$$

$\square$

## D Experiments

### D.1 Colored MNIST and Bars

We first evaluate the guarantees provided by BD-CLS (Thm.2) on a modified MNIST dataset [14, 36] featuring colored digits and bars. [6]. The ground truth ASCM includes generative factors: Digit (0-9 $D$), Digit Color (red: $DC = 0$; green: $DC = 1$), Bar Width (thin: $BW = 0$; thick: $BW = 1$), Bar Color (red: $BC = 0$; green: $BC = 1$), and other latent factors such as handwriting style. The causal relationships are shown in Fig. S5(a) Other factors (e.g., writing style $\mathbf{S}$) are considered independent factors and are ignored in the diagram.

To illustrate, digit ($D$) and digit color ($DC$) are confounded, showing a negative correlation: the larger digits ($\geq 5$) tend to be red, and the smaller digits ($< 5$) tend to be green, but do not directly affect each other. The digit color ($DC$) has a positive effect on bar color ($BC$); for example, red digits are more likely to have red bars. The digit ($D$) has a positive effect on bar width ($BW$); larger digits are more likely to be with thick bars. However, when the digit color is green, this causal relationship is flipped, and the digit negatively affects the bar width. Formally, the ground truth generation process $\mathcal{M}^*$ is given by

$$\begin{cases} D \leftarrow U_D \\ DC \leftarrow \mathbf{1}[U_D \geq 5] \oplus U_{DC} \\ BC \leftarrow DC \oplus U_{BC} \\ BW \leftarrow ((\mathbf{1}[D \geq 5] \wedge U_1) \oplus (\mathbf{1}[D < 5] \wedge U_2)) \oplus DC \\ \mathbf{S} \leftarrow f_{\mathbf{S}}(\mathbf{U}_S) \\ \mathbf{I} \leftarrow f_{\mathbf{I}}(D, DC, BC, \mathbf{S}), \end{cases} \tag{93}$$

---

[6]A bar in an image refers to a complete row of pixels with the same color.

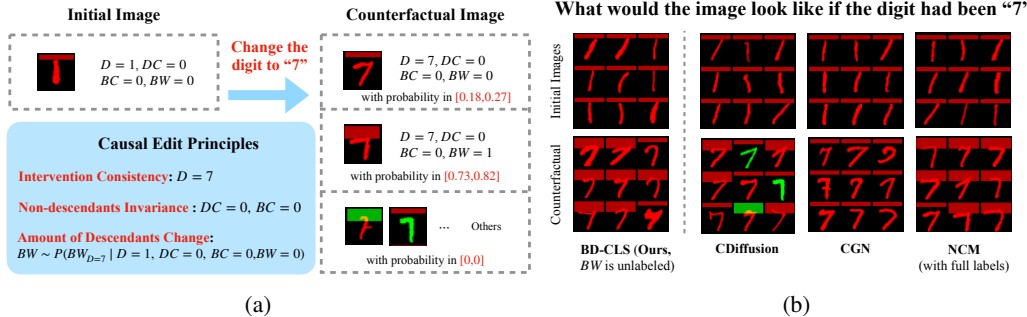

(a)                                                                    (b)

Figure S6: Replot of Fig. 5. Edit a red "1" with a thin red bar to digit "7". (a) Expectation of counterfactual consistent editing; (b) Edit results. Top - initial image. Bottom - counterfactual images.

where the exogenous variable distributions are:

$$U_D \sim \text{Uniform}[0, 9]$$
$$U_{DC} \sim \text{Bernoulli}(0.75)$$
$$U_{BC} \sim \text{Bernoulli}(0.4) \qquad (94)$$
$$U_1 \sim \text{Bernoulli}(0.75)$$
$$U_2 \sim \text{Bernoulli}(0.1)$$

Fig. S5(b) shows 50 random samples in the data set.

**Task 1: Counterfactually editing digits**

We first consider editing the digit $D$ in an image. Suppose that we are editing a red "1" with a thin red bar ($D = 1, DC = 0, BC = 0, BW = 0$) and wonder what would happen had the digit "1" been a "7". According to the data generation model $\mathcal{M}^*$ and the counterfactual behavior delivered by Thm. 1, the digit should be a "7", which implies interventional consistency is achieved; (2) the non-descendants should be invariant. So digit color ($DC$) and bar color ($BC$) remain red;. (3) The descendant $BW$ (bar width) should change, and the probability of being thicker is

$$Q_1 = P(BW_{D=7} = 1 \mid D = 1, DC = 0, BC = 0, BW = 0) \qquad (95)$$

To guarantee counterfactual consistency (Def. 3), the estimation of $Q_1$ should be within the bound $[0.73, 0.82]$ according to Def. 3. These edit expectations are summarized in Fig. S6(a).

The editing results are shown in Fig. S6(b). All models achieve interventional consistency, that is, all edited images depict the digit '7'. However, CDiffusion fails to preserve non-descendant invariance: both the digit color and bar color sometimes change to green. CGN, on the other hand, fails to reflect descendant delta: the bar width remains unchanged, even when the digit changes. In contrast, both the BD-CLS (without $BW$ labels) and the NCM (with full labels) achieve counterfactual consistency. They preserve the color of both the digit and the bar and successfully induce an increase width in bar width when editing the digit. Notably, while the fully supervised NCM requires labeled data for $BW$, BD-CLS achieves the same behavior without requiring those labels, demonstrating its ability to provide counterfactual consistency for unlabeled features.

To quantify descendant delta, we report the results of estimating the query $Q$ in Fig. S7(a). Specifically, we repeat each method four times and measure the probability that the bar becomes thicker after changing the digit to "7". The numerical results show that both BD-CLS and the NCM with full labels maintain the estimate within the theoretical bounds, whereas the CDiffusion and CGN do not.

**Task 2: Counterfactually Edit Digit Color**

We next consider editing a digit's color. Suppose that we are editing a green "0" with a thick green bar and wonder what would happen had the digit color been red. According to the data generation model $\mathcal{M}^*$ and the counterfactual behavior established by Thm. 1, the digit should be green, which implies interventional consistency is achieved; (2) the non-descendants should be invariant. So the digit ($D$) remain a "0"; (3) The descendant $BC$ (bar color) should change, and the probability of

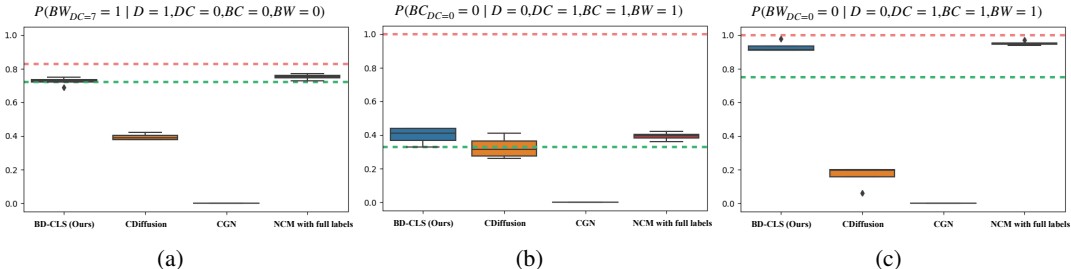

Figure S7: Numerical evaluations of F-ctf queries in Colored MNIST and Bars experiments. (a) counterfactually edit digits; (b, c) counterfactually edit digit color. To achieve counterfactual consistency, the estimation should fall within the bounds (between the green line and the red line).

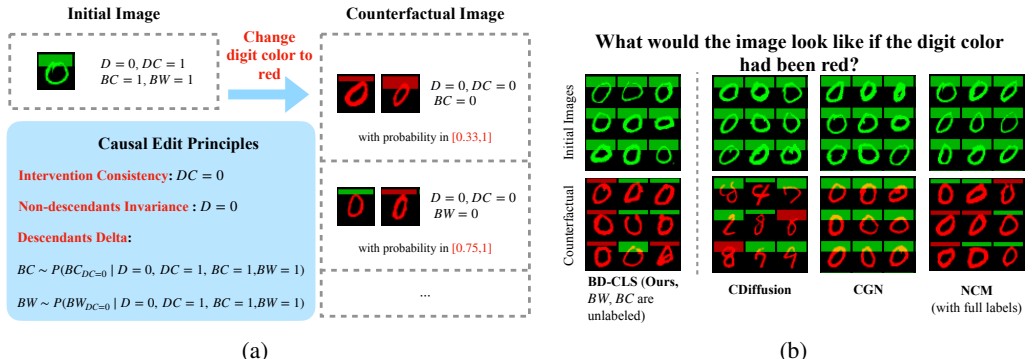

Figure S8: Edit a green "0" with a green thick bar to red digit. (a) Expectation of counterfactual consistent editing; (b) Edit results. Top - initial image. Bottom - counterfactual images.

being red is

$$Q_2 = P(BC_{DC=0} = 0 \mid D = 0, DC = 1, BC = 1, BW = 1) \tag{96}$$

To guarantee counterfactual consistency, the estimation of $Q_1$ should be within the bound $[0.33, 1]$. Another descendant $BW$ (bar width) should also change and the probability of being thin is

$$Q_3 = P(BW_{DC=0} = 0 \mid D = 0, DC = 1, BC = 1, BW = 1) \tag{97}$$

To guarantee counterfactual consistency, the estimation of $Q_3$ should be within the bound $[0.89, 1]$ according to Def. 3. These edit expectations are shown in Fig. S6(a). Unlike editing digits (Task 1), BD-CLS are obtained in this task with only labels of $D$ and $DC$.

The editing results are shown in Fig. S6(b). All models achieve interventional consistency, that is, all edited images depict the red digit. However, CDiffusion fails to preserve non-descendant invariance: the digit almost always changes. CGN, on the other hand, does not reflect the change in descendant: the bar color and width remain unchanged, even when the digit changes. In contrast, both the BD-CLS and the NCM with full labels achieve counterfactually consistent results. They preserve the digit, and successfully change the bar color to red and reduce the bar width. Notably, while the fully supervised NCM requires labeled data for $BC$ and $BW$, BD-CLS achieves the same behavior without requiring those labels, demonstrating its ability to provide counterfactual consistency for unlabeled features.

To quantify descendant delta, we report the results of estimating the query $Q$ in Fig. S7(b, c). Specifically, we repeat each method four times and measure (1) the probability that the bar becomes red after changing the digit to red (Fig. S7(b)); (2) the probability that the bar becomes thicker after changing the digit to red (Fig. S7(c)). The numerical results show that both BD-CLS and the NCM with full labels maintain the estimate within the theoretical bounds, whereas the CDiffusion and CGN do not.

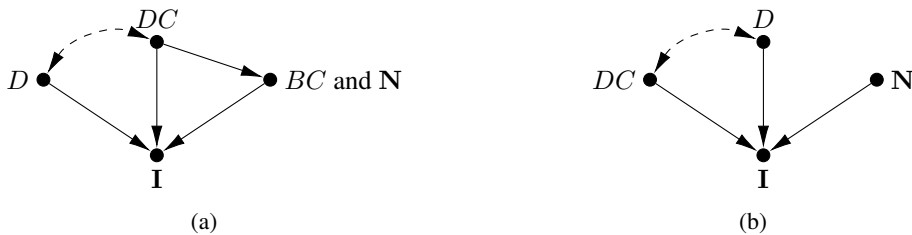

Figure S9: (a) BD-CLS for Task 1 ; (b) BD-CLS for Task 2.

## D.2  Implementation Details for Colored MNIST and Bars Experiments

We first provide more details on the architectures of the BD-CLS and other baselines: conditional diffusion, CGN, and NCM with supervision. We first present the formal definition of NCM [59, 60].

**Definition 11** ($\mathcal{G}$-Constrained Neural Causal Model ($\mathcal{G}$-NCM)). Given a causal diagram $\mathcal{G}$, a $\mathcal{G}$-constrained Neural Causal Model (for short, $\mathcal{G}$-NCM) $\widehat{\mathcal{M}}(\boldsymbol{\theta})$ over variables $\mathbf{V}$ with parameters $\boldsymbol{\theta} = \{\theta_{V_i} : V_i \in \mathbf{V}\}$ is an SCM $\langle \widehat{\mathbf{U}}, \mathbf{V}, \widehat{\mathcal{F}}, \widehat{P}(\widehat{\mathbf{U}}) \rangle$ such that $\widehat{\mathbf{U}} = \{\widehat{U}_\mathbf{C} : \mathbf{C} \subseteq \mathbf{V}\}$, where
(1) each $\widehat{U}$ is associated with some subset of variables $\mathbf{C} \subseteq \mathbf{V}$, and $\mathcal{X}_{\widehat{U}} = [0,1]$ for all $\widehat{U} \in \widehat{\mathbf{U}}$;
(2) $\widehat{\mathcal{F}} = \{\hat{f}_{V_i} : V_i \in \mathbf{V}\}$, where each $\hat{f}_{V_i}$ is a feed forward neural network parameterized by $\theta_{V_i} \in \boldsymbol{\theta}$ mapping values of $\mathbf{U}_{V_i} \cup \mathbf{Pa}_{V_i}$ to values of $V_i$ for $\mathbf{U}_{V_i} = \{\widehat{U}_\mathbf{C} : \widehat{U}_\mathbf{C} \in \widehat{\mathbf{U}} \text{ s.t. } V_i \in \mathbf{C}\}$ and $\mathbf{Pa}_{V_i} = Pa_{\mathcal{G}}(V_i)$;
(3) $\widehat{P}(\widehat{\mathbf{U}})$ is defined s.t. $\widehat{U} \sim \text{Unif}(0,1)$ for each $\widehat{U} \in \widehat{\mathbf{U}}$. ∎

**BD-CLS**. As illustrated in Sec. 5.1, the implementation is based on NCM. The architecture designed has two stages, which mimics the ASCM generation process. In the first stage, we train a GAN-NCM [60] on observed generative factors at the generative level. Specifically, the observed generative factors are $\{D, DC, BC\}$ and $BW$ does not belong to $\mathbf{V}$ in the NCM for task 1 (editing digits). In the second stage, we train a conditional diffusion model $\widehat{f}_\mathbf{I}$ taking conditions $\{D, DC, BC\}$ and noise $\mathbf{N}$ as input to generate image $\mathbf{I}$.

NCMs ensure that the resulting model satisfies the definition of a BD-CLS. For example, in our setting, Digit Color ($DC$) serves as a backdoor set for Digit ($D$) based on the ground-truth causal graph $\mathcal{G}$ shown in Fig.7. According to Def.4, this augmented NCM model satisfies the generation condition, as the conditional diffusion model is trained to approximate $P(\mathbf{I} \mid D, DC)$. Second, taking $\mathbf{Z} = \{D, \mathbf{N}\}$, the non-descendants $\{DC, BC\}$ are directly modeled in the NCM and remain disentangled from the intervention variable $D$. The structure of this augmented NCM, shown in Fig. S9(a), aligns with the structural condition in Def.4, confirming its compatibility with the BD-CLS framework. For task 2 (editing digit's color), the observed generative factors are $\{D, DC\}$ and $\{BC, BW\}$ do not belong to $\mathbf{V}$ in the NCM. The corresponding NCM structure is shown in Fig. S9(b).

For detailed implementation, at the generative level, each function $\widehat{f}_V$ $\widehat{\mathcal{F}}$ in $\widehat{\mathcal{M}}$ is a feedforward neural network with 2 hidden layers of width 64 with layer normalization applied [1]. Each exogenous variable $\widehat{U} \in \widehat{\mathbf{U}}$ is a standard normal four-dimensional distribution. The generator and discriminator are trained with a learning rate of $10^{-4}$, and are optimized with Adam optimizer [27]. All training processes are performed with a batch size of 100. The model architecture of conditional diffusion follows the implementation in [20]. Specifically, we use four feature map resolutions ($32 \times 32$ to $4 \times 4$). Two residual blocks per feature map and self-attention blocks at $16 \times 16$ are implemented. The total step size $T$ is set as 1000. We train the model on a single NVIDIA H100 GPU epoch for 100 epoch. In addition, we generate a pair of initial image and counterfactual image from the model in this experiment. In other words, we do not take a real image as input, but we generate the initial image and edit it at the same time.

**Conditional Diffusion**. The first non-causal baseline is chosen as the conditional diffusion model that approximates $P(\mathbf{I} \mid \mathbf{X})$. To have a better comparison, we use the exact same architecture of $\widehat{f}_\mathbf{I}$ for BD-CLS.

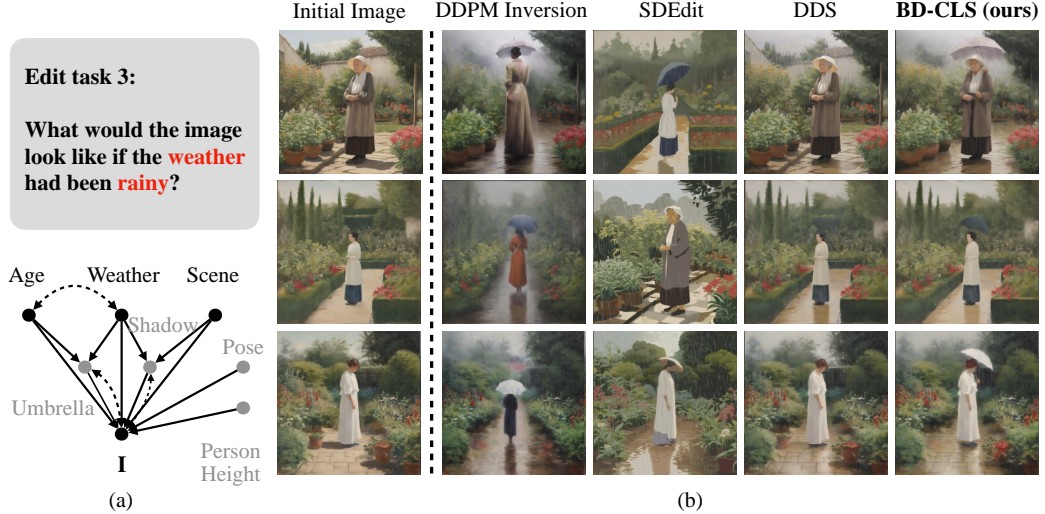

Figure S10: The causal diagrams and image editing results for Task 3 - Edit the weather (umbrella).

**CGN [50]**. The second baseline is CGN. We follow the implementation in [36]. CGN proposes to encode an SCM over variables $Shape, Texture, Background$, and $Label$ into the proxy generative model. Given the label of the image, $Shape, Texture, Background$ are independent. Formally, the mechanism of this SCM is designed as follows:

$$
\begin{cases}
Label \leftarrow f_l(U_l) \\
Shape \leftarrow \widehat{f}_s(Label, U_d) \\
Texture \leftarrow \widehat{f}_t(Label, U_s) \\
Background \leftarrow \widehat{f}_b(Label, U_b) \\
\mathbf{I} \leftarrow \widehat{f}_{\mathbf{I}}(Shape, Texture, Background),
\end{cases}
\tag{98}
$$

where mechanism $f_s, f_t, f_b$ is designed to learn the conditional distribution $P(V \mid Label)$ with prior knowledge, where $V \in \{Shape, Texture, Background\}$. The composition mechanism $\widehat{f}_{\mathbf{I}}$ is not learned but is defined analytically. After fitting the given observational distribution $P(Label, \mathbf{I})$, the intervention can be performed by changing $Label$. In task 1, the digit and the writing style are regarded as $Shape$; the color is regarded as $Texture$ and the colored bar is regarded as $Background$. In task 2, the color of the digit and the writing style are considered as $Shape$; the digit is considered as $background$ and the colored bar is regarded as $Texture$ and the colored bar is regarded as $Background$.

We use the same conditional diffusion model learn mechanism $f_s$. $f_t, f_b$ are directly hand designed in task 1 while $f_t$ is learned through conditional diffusion in task 2. Theoretically, CGN learns the independent mechanism from $Shape, Texture, Background$ to the image. After performing interventions on one variable, others should be preserved in the image.

**Full supervised NCM**. The third baseline is chosen as fully supervised NCMs. The implementation of this casual basline is exactly the same as BD-CLS but with all the labels over $\{D, DC, BC, BW\}$.

### D.3    Real World Scenarios Editing

In this section, we validate **BD-CLS-Edit** for sampling counterfactual images in more open scenarios. We compare it against three non-causal SOTAs: (1) DDPM Inversion[22] and (2) SDEdit[33], representing LS inversion, and (3) DDS[22], which emphasizes semantic invariance.

**Task 3: Counterfactually editing the weather**

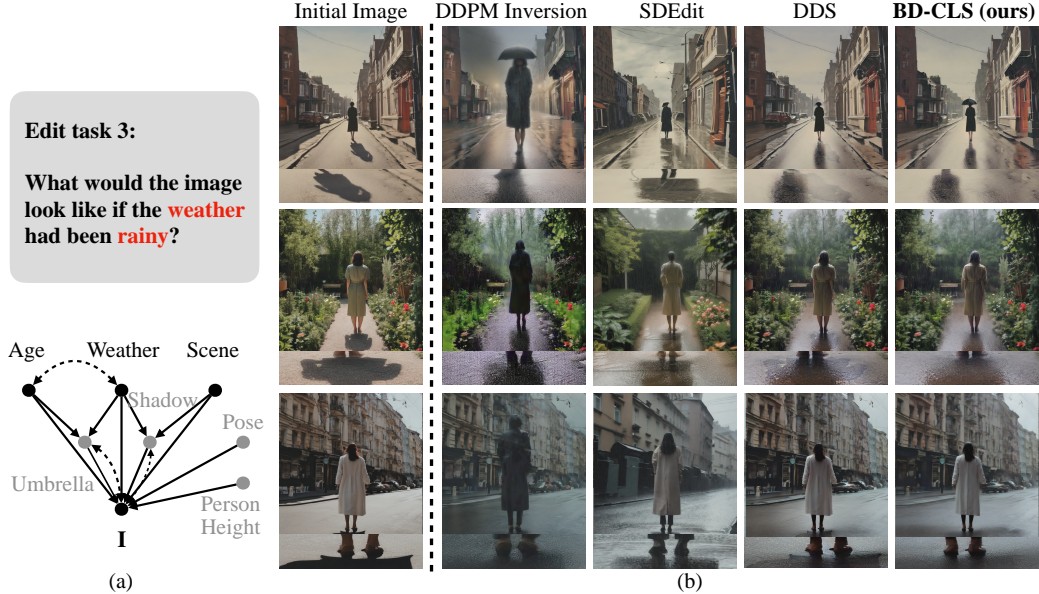

Figure S11: The causal diagrams and image editing results for Task 3 - Edit the weather (shadow).

We begin with the setting from Example 1, where the goal is to change the weather from sunny to rainy in an image of a young (or old) lady in a garden (or street). The causal relationships between the generative factors are shown in Fig.S10(a))[7].

According to Theorem1, non-descendants (e.g., scene layout, age, person's height) should be preserved, while descendants (e.g., umbrella, shadows) should change accordingly regardless of whether they are prompted. For example, an umbrella may appear and shadows should become fuzzier on wet ground due to the weather change. As shown in Fig. S10 and Fig. S11, all methods achieve interventional consistency. However, DDPM inversion alters non-descendants, changing the scene and person's height. SDEdit also changes the layout and person's pose. DDS maintains visual similarity to the original image but fails to reflect downstream effects. To illustrate, the umbrella does not appear, and the shadows on a sunny day are preserved. In contrast, BD-CLS preserves non-descendants and correctly reflects the causal effects on descendants like the umbrella and shadow.

### Task 4: Counterfactually editing season

Next, we consider editing an image described as 'a person in a forest' by changing the season from summer to fall. The corresponding causal diagram is shown in Fig. S12(a). According to Theorem1, non-descendants, for example, the person's gender and height, forest layout, should be preserved, even if not prompted, while descendants, such as clothing and leaf color, should change according to the causal effect of season. To illustrate, a person in the fall intends to wear more clothes. Fig. S12(b) shows the editing results of our BD-CLS method compared to the baselines. DDPM inversion fails to generate details for the person and the person's height changes. SDEdit also changes the person's identity. DDS preserves personal details, but the resulting clothing appears unrealistic since it keeps too many original details in the clothes. In contrast, BD-CLS produces appropriate generate the season and realistic warmer clothing while preserving non-descendant features.

### Task 5: Counterfactually editing the scene

Third, we consider editing an image described as "a person in a grocery store". Specifically, we intervene on the scene and aim to change the background to a garden. The causal diagram is shown in Fig.S13(a). According to the causal structure, non-descendants, for instance, background layout, person's pose, should remain unchanged, while descendants, such as a grocery bag, should be

---

[7]It is worth noting that even though the diagram shows fewer than 10 variables in the demonstrated graph, many additional variables are implicitly present in the causal graph, and BD-CLS-Edit also provides counterfactual guarantees for these implicit variables. For example, other factors, such as the hair color, and scene lightness, are also part of generative factors in images but are not drawn explicitly for the clean presentation.

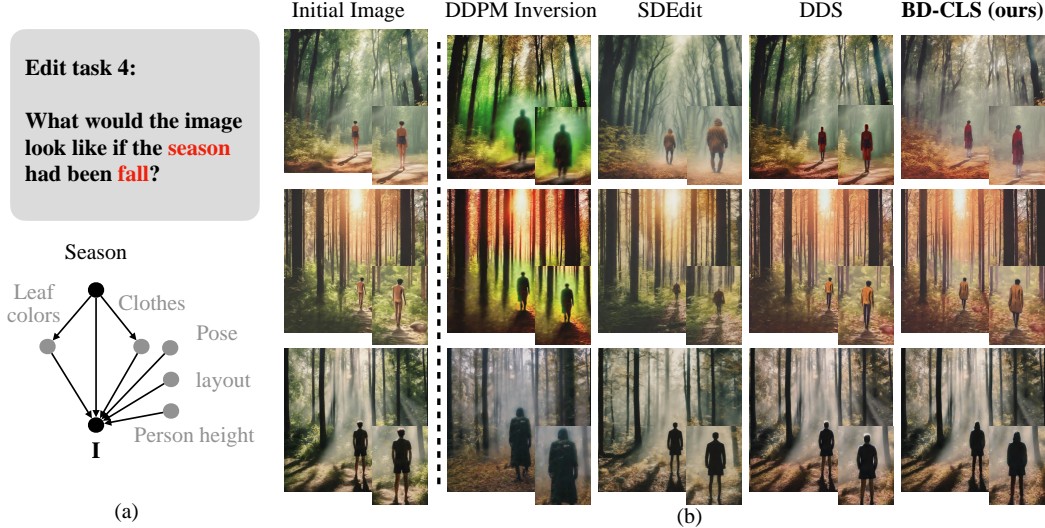

Figure S12: The causal diagrams and image editing results for Task 4 - Edit the season.

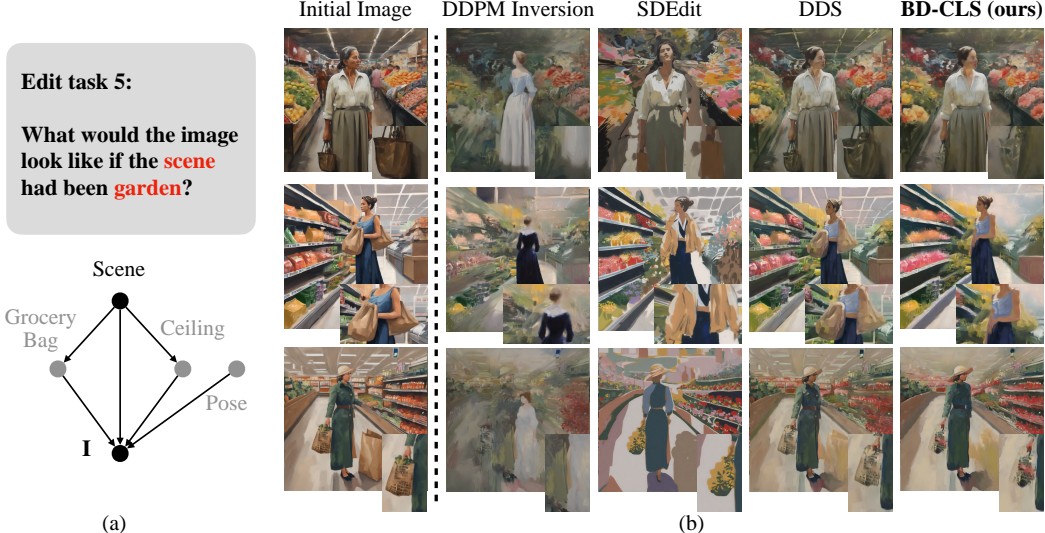

Figure S13: The causal diagrams and image editing results for Task 5 - Edit the scene.

removed, as it is unlikely to appear in a garden setting. Figure S13(b) shows the editing results. DDPM inversion and SDEdit alters the person significantly, failing to preserve non-descendants. DDS retains most personal details, but incorrectly preserves the grocery bag. In contrast, BD-CLS maintains interventional consistency while correctly removing the grocery bag, reflecting the expected causal effect.

**Task 6: Counterfactually editing the place and the sport** Fourth, we consider editing an image described as 'a person is skiing in the snow' by intervening in the place and the sport. The corresponding causal diagram is shown in Fig.S14(a). According to Thm. 1, non-descendants, such as the position of the person in the image, should remain unchanged, while descendants, including surrounding details and sports equipment, should change accordingly. Figure S14(b) shows the editing results. DDPM inversion alters the person's gesture and location, failing to preserve non-descendants. DDS retains most visual details from the original image, but this leads to unrealistic edits; for instance, snowy mountains and trees remain, and skiing gear (e.g., ski poles and clothing) are preserved, which would not usually appear in a surfing scene. In contrast, SDEdit and BD-CLS preserves the person's gesture and location while transforming the background into ocean-like waves and replacing skiing

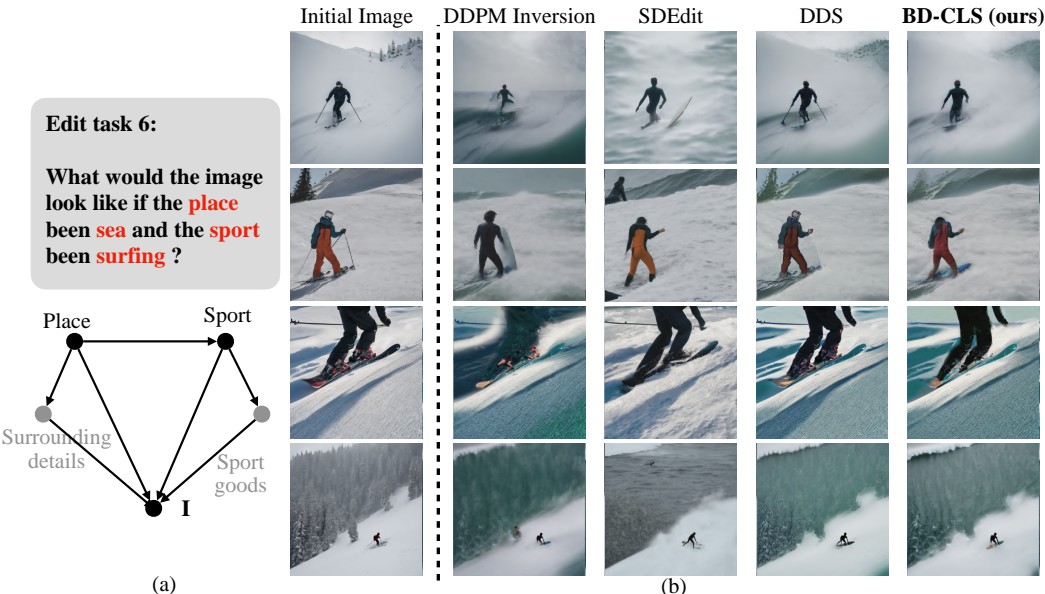

Figure S14: The causal diagrams and image editing results for Task 6 - Edit the place and the sport.

| Task # | DDPM Inversion | SDEdit | DDS | BD-CLS-Edit |
|--------|---------------|--------|------|-------------|
| 3 | 0.25 | 0.36 | 0.13 | 0.16 |
| 4 | 0.33 | 0.15 | 0.14 | 0.14 |
| 5 | 0.71 | 0.69 | 0.66 | 0.61 |
| 6 | 0.58 | 0.55 | 0.57 | 0.47 |

Figure S15: LPIPS comparison across editing methods. Lower values indicate stronger structure preservation. BD-CLS-Edit maintains low LPIPS across diverse tasks while preserving causal consistency.

gear with appropriate surfing equipment. The reason that SDEdit performs well in this task is that there are not too many non-descendants needed to be preserved.

To qualitatively assess the generation results, we edit 120 images for each real-world task and report LPIPS [62] to quantify the degree of structure preservation. A lower LPIPS value indicates that the edited image is more similar to the original one. According to Thm. 1, a lower LPIPS suggests better preservation of non-descendant features. However, a low LPIPS does not necessarily imply better editing quality, as it may conflict with the third principle - descendant delta. The LPIPS results are shown in Fig. S15. As illustrated, BD-CLS-Edit achieves lower LPIPS values than DDPM Inversion and SDEdit, but higher LPIPS than DDS. In addition, when tasks primarily require preserving non-descendants rather than changing descendants (Tasks 1 and 2), the LPIPS of BD-CLS-Edit is close to that of LS inversion methods. Conversely, when tasks demand more modification of descendants than preservation of non-descendants (Tasks 3 and 4), the LPIPS of BD-CLS-Edit again aligns more closely with those methods. This observation aligns with the expected behavior of counterfactual editing: latent-space inversion methods may inadvertently alter non-descendants, while semantic-invariance approaches can mistakenly preserve descendants. In contrast, the proposed BD-CLS-Edit generates more causally consistent and realistic results.

### D.4 Implementation Details for Real World Scenarios Experiments

We first illustrate the dataset construction process for the real-world scenario. For each scenario, the underlying ASCM can be illustrated by a specific text. To illustrate,

- Task 3. *An image of a* $(Age)$ *woman standing in a* $(Scene)$ *during a* $(Weather)$ *day*.
- Task 4. *An image of a person standing in a forest during a* $(Season)$.
- Task 5. *An image of a lady standing in a* $(Scene)$.

- Task 6. *An image of a person* $(Sport)$ *in* $(Place)$.

We construct data from the observed distribution $P(\mathbf{V}, \mathbf{I})$ and initial images $\mathbf{i}$ for editing tasks in three steps:

1. **Design the generative SCM.**
   Define a structural causal model (SCM) over the labeled generative factors $\mathbf{V}$ that is compatible with the causal diagram $\mathcal{G}_{\mathbf{V}}$, and sample concept data $\mathbf{v} \sim P(\mathbf{V})$.[8]

2. **Generate images.**
   For each sampled concept datapoint $\mathbf{v}$, generate an image $\mathbf{i} \sim P^{\text{Gen}}(\mathbf{I} \mid \mathbf{v})$, where $P^{\text{Gen}}(\mathbf{I} \mid \mathbf{v})$ denotes a text-to-image generator (DiffusionXL[43] is employed in this work).

3. **Select editing inputs.**
   Collect a set of images that all contain a specific feature $\mathbf{x}$ to serve as initial inputs.

Following this procedure, the resulting observed distribution $P(\mathbf{V}, \mathbf{I})$ is guaranteed to be compatible with $\mathcal{G}_{\mathbf{V}, \mathbf{I}}$. In addition, all initial images share the same original feature values, which differ from the intervened ones ($\mathbf{x} \neq \mathbf{x}'$ assumed in this work). Specifically, a generative SCM over $\{Scene, Age, Weather\}$ for Task 3 is designed as follows:

$$\begin{cases} Scene \leftarrow U_S \\ Age \leftarrow U_{A_1} \wedge (U_W \oplus U_{A_2}) \\ Weather \leftarrow U_W \end{cases} \tag{99}$$

where the exogenous variable distributions are:

$$\begin{aligned} U_S &\sim \text{Bernoulli}(0.5) \\ U_{A_1} &\sim \text{Bernoulli}(0.5) \\ U_{A_2} &\sim \text{Bernoulli}(0.1) \\ U_W &\sim \text{Bernoulli}(0.4) \end{aligned} \tag{100}$$

where $Scene = 0$ means the scene is a garden; $Age = 0$ means the age is young; $Weather = 1$ means the weather is rainy. Then, data for Scene, Age, Weather can be sampled from this generative SCM. In Step 2, for each sampled datapoint—such as Scene $= 0$, Age $= 1$, Weather $= 0$—an image $\mathbf{i}'$ is generated by providing the prompt "A woman standing in a garden on a sunny day" to the text-to-image generator. Finally, in Step 3, all images corresponding to the sunny condition are collected as the initial set for editing.

We use Stable Diffusion XL[43], and all editing is performed in the latent space, after encoding the input image. In other words, $\mathbf{i}^{(0)}$ in Alg. 3 refers to the latent representation obtained via the pre-trained SDXL autoencoder. The input image size is $1024 \times 1024 \times 3$, and the image after encoding is $128 \times 128 \times 4$. For classifier-free guidance, we fix the parameter $\omega$ (Eq. 18) is fixed as 7.5. Other hyperparameters in Alg. 3 are given as follows. The total inference steps are set to 200. $\overline{\mathbf{T}}$ of length 40 is randomly sampled from $\{1, ..200\}$. We manualLy tune the hyperparamters for BD-CLS-Edit, including learning rate $\gamma$, optimization iteration number $n_{max}$, and clipping value $\theta_{\max}$. Specifically, we compare the combination of $\gamma \in \{1e-1, 1e-2, 1e-3, 1e-4\}$, $n_{\max} \in [2, 10, 20]$, and $\theta_{max} \in \{1.2, 1.5, 2.0\}$. We found that $\gamma = 1e-2$ and $n_{\max} = 10$ lead to the best BD-CLS-Edit's performance considering both effectiveness and optimization time. The BD-CLS-Edit is relatively robust to $\theta_{m}ax$. The initial $\boldsymbol{\theta}$ is set to 0 for $\theta_T$ through $\theta_{T-50}$ and the others are initialized as 1. $\boldsymbol{\theta}$ are optimized individually for each input image. The adjusted parameters follow the coefficients in DDS [18]. The learning rate $\mu$ is set as 0.1 and the optimization is performed with SGD. The experiments are also conducted on a single NVIDIA H100 GPU.

### D.5 Sensitivity and Robustness for BD-CLS

In this section, we examine the sensitivity and robustness of BD-CLS to the input causal graph. Recall that Thm. 2 requires the labeled or prompted feature set $\mathbf{B}$ to serve as a valid backdoor set in the true causal diagram $\mathcal{G}$. Following this requirement, BD-CLS-Edit set the prompt to include a description

---

[8]We exclude unlabeled generative factors from the design, as there can be hundreds or thousands of them in an image, many of which cannot be easily described in text, such as subtle lighting details or shadow textures.

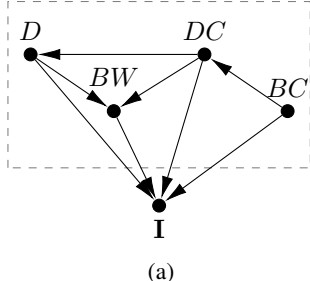 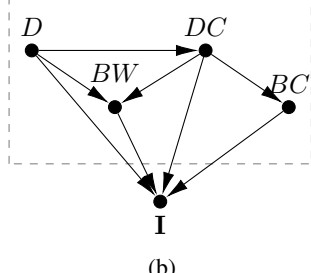

(a)                                        (b)

Figure S16: Non-complete or misspecified causal graphs used for sensitivity analysis experiments. (a) $\mathcal{G}'$; (b) $\mathcal{G}''$

involving the feature **b** at step 2. Here, we theoretically and empirically analyze scenarios where the provided causal graph is mis-specified.

Let $\mathcal{G}'$ denote a non-complete or mis-specified causal graph. If the backdoor set $\mathbf{B}'$ derived from $\mathcal{G}'$ still satisfies the backdoor criterion in the true graph $\mathcal{G}$, then BD-CLS continues to provide causal guarantees. For example, consider the Colored MNIST and Bars experiments (App.D.1). In the ground-truth graph $\mathcal{G}$ (Fig.S5(a)), the digit ($D$) is confounded with the digit color ($DC$), and the bar color ($BC$) is a descendant of $DC$. Even if one mistakenly assumes that both $DC$ and $BC$ are ancestors of $D$ ($\mathcal{G}'$ in Fig. S16), the set $DC, BC$ still satisfies the backdoor criterion from $D$ to $\mathbf{I}$ in both the incorrect and true graphs. Thus, BD-CLS-Edit remains valid under this mis-specification, and counterfactual editing performance is unaffected. To ground this, we practically evaluate F-ctf query $P(BW_{D=7} = 1 | D = 1, DC = 0, BC = 0, BW = 0)$, the same query as in Fig. S7(a), using the misspecified graph $\mathcal{G}'$. This query measures the probability that the bar becomes thicker after editing, capturing the causal effect from $D$ to $BW$. The results (Fig. S17, third column) show that the F-ctf estimate remains within the theoretical bounds, indicating that BD-CLS provides robust and reliable counterfactual edits even with a misspecified input graph.

Conversely, if the backdoor set derived from the input graph is not valid in the true causal graph, counterfactual editing may become unreliable. For instance, suppose the misspecified graph $\mathcal{G}''$ (Fig. S16(b)) incorrectly models $DC$ as a

| $l$ | $r$ | BD-CLS ($\mathcal{G}'$) | BD-CLS ($\mathcal{G}''$)) |
|------|------|------|------|
| 0.73 | 0.82 | $0.73 \pm 0.03$ | $0.45 \pm 0.06$ |

Figure S17: Sensitivity analysis results.

descendant of $D$, leading to its exclusion from the backdoor set $\mathbf{B}$. In this case, interventions on $D$ can inadvertently affect $DC$, violating the non-descendant invariance editing principle in Thm. 1. In addition, such incorrect adjustment results in biased estimation of the causal effect from $D$ to $BW$. Evaluating the same F-ctf query under this $\mathcal{G}''$ (column 4 in Fig. S17) confirms this: the estimated effect from $D$ to $BW$ is significantly underestimated.

## D.6 Editing with BD-CLS for Other Datasets

The experiments for other datasets are provided in the full technical report [37].

## E Further Discussions and Examples

### E.1 Augmented Structural Causal Models (Def. 1)

We begin with several remarks regarding this ASCM generative process.

**Remark 1** (Unlabeled factors **L**). *The unlabeled factors* **L** *are the key difference compared to the ASCM in [36]. An image often contains rich concepts that cannot be fully captured by humans. Thus, the labeled information cannot be given to all of them. For example, annotations of an image are only given to several user care features; a text description of an image usually focuses on main concepts and ignores details.*

**Remark 2** (Unobserved endogenous variable **L** and unobserved exogenous variable **U** in ASCM). *There can be two standard confusions related to the difference between* **U** *and* **L** *as they are not all unobserved/labeled. First, generative factors* $\mathbf{L} \in \mathbf{Pa}(\mathbf{I})$ *are directly reflected in the image, while*

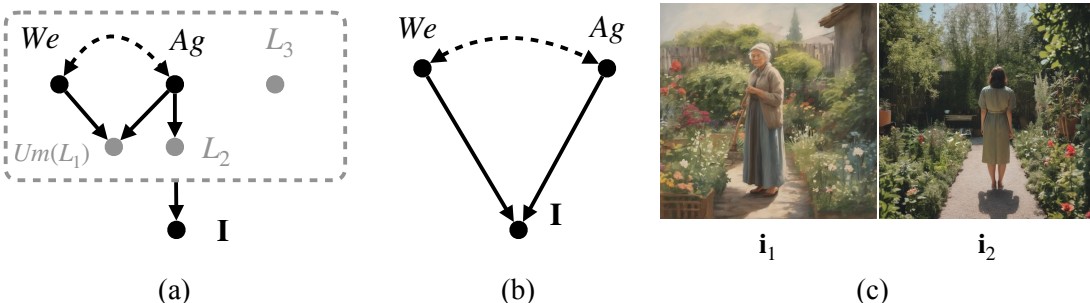

*A young/old lady is standing in the garden during a rainy/sunny day*

(a)  (b)  (c)

Figure S19: (a) The causal diagram over $\mathbf{V}$ and $\mathbf{L}$ at generative level; (b) The causal diagram over $\mathbf{V}$ and $\mathbf{I}$; (c) observational image samples in Example. 2.

$\mathbf{U}$ *is not. Specifically, even if the unsupervised concept is not described in the annotation or text, it exists in the image and can be mapped by $h$ from $\mathbf{I}$. See example 2 for more details.*

**Remark 3** (No exogenous variable $\mathbf{U_I}$ for image $\mathbf{I}$). $\mathbf{L}$ *are unobserved parents of image variable $\mathbf{I}$. While one might surmise that $\mathbf{L}$ can be treated as the exogenous variable $\mathbf{U_I}$ associated with $\mathbf{I}$—that is, denote $\mathbf{L}$ as $\mathbf{U_I}$—this is not the case. In the SCM, the variables in $\mathbf{L}$ are endogenous and may be descendants of $\mathbf{V}$, whereas $\mathbf{U_I}$, by definition, must not be descendants of any observed variables.*

**Remark 4** (The invertibility of $f_{\mathbf{I}}$). *The $f_{\mathbf{I}}$ is assumed invertible in the generative process since these generative factors are present directly in a given image, regardless of features being labeled or not. This assumption is standardly used in non-linear ICA and representation learning literature [32, 29, 24, 25, 31].*

The ASCM induces a causal diagram $\mathcal{G}_{\mathbf{V},\mathbf{L},\mathbf{I}}$ over all generative factors $\mathbf{V}, \mathbf{L}$, and image $\mathbf{I}$. This full diagram can be projected onto a causal diagram involving only the observed variables, denoted $\mathcal{G}_{\mathbf{V},\mathbf{I}}$. In this work, we assume that prior knowledge of $\mathcal{G}_{\mathbf{V},\mathbf{I}}$ - sometimes abbreviated as $\mathcal{G}$ for simplicity - is available, from the human common sense or from experts in the domain and is used as an inductive bias. However, it is not assumed that the complete generative graph $\mathcal{G}_{\mathbf{V},\mathbf{L},\mathbf{I}}$ is known.

| $We$ | $Ag$ | $Um$ | $P(We, Ag, Um)$ |
|---|---|---|---|
| 0 | 0 | 0 | 0.4416 |
| 0 | 0 | 1 | 0.0384 |
| 0 | 1 | 0 | 0.3136 |
| 0 | 1 | 1 | 0.0064 |
| 1 | 0 | 0 | 0.0224 |
| 1 | 0 | 1 | 0.0576 |
| 1 | 1 | 0 | 0.0984 |
| 1 | 1 | 1 | 0.0216 |

Figure S18: $P(\mathbf{V})$ induced by the ASCM in Ex.. 2.

**Example 2** (continued Example. 1). Consider an image describing "a young/old lady is standing in the garden during a rainy/non-rainy day". We consider the augmented generative process, ASCM

$$\mathcal{M}^* = \langle \mathbf{U} = \{U_1, U_2, U_3, U_4, \mathbf{U_L}\}, \{\mathbf{V} = \{We, Ag\}, \mathbf{L} = \{Um(L_1), L_2, L_3, ...\}, \mathbf{I}\}, \mathcal{F}^*, P^*(\mathbf{U}) \rangle \tag{101}$$

where the mechanisms

$$\mathcal{F}^* = \begin{cases} We \leftarrow U_1 \\ Ag \leftarrow U_1 \oplus U_2 \\ Um \leftarrow ((\neg Ag) \oplus U_3) \wedge (We \oplus U_4) \\ L_2 \leftarrow f_{L_2}^*(Ag, U_{L_1}), L_3 \leftarrow f_{L_3}^*(U_{L_2}) \\ ... \\ \mathbf{I} \leftarrow f_{\mathbf{I}}^*(We, Ag, Um, L_2, L_3, ...) \end{cases} \tag{102}$$

and exogenous variables $U_1, U_2, U_3, U_4$ are independent binary variables, and $P(U_1 = 1) = 0.2, P(U_2 = 1) = 0.4, P(U_3 = 1) = 0.2, P(U_4 = 1) = 0.1$. $\mathbf{U_L} = \{U_{L_2}, U_{L_3}, ...\}$ are also independent of $\{U_1, U_2, U_3, U_4\}$.

At the generative level, the labeled variables $\mathbf{V}$ contain two variables $\{We, Ag\}$; $We$ represents if the weather is rainy (rainy $We = 1$; non-rainy $We = 0$); $Ag$ represents the age of the lady (Young $Ag = 1$; Old $Ag = 0$;). $\mathbf{L}$ represents unlabeled factors that do not appear in the text description,

including whether the person has an umbrella $Um(L_1)$ (with umbrella $Um = 1$; without umbrella $Um = 0$), the hair color of the person ($L_2$), pose ($L_3$), etc. As discussed in Remark 2, although these factors are not labeled, they are parents of image variable $\mathbf{I}$, and play different roles with $U_1, U_2, U_3, U_4, \mathbf{U_L}$ in the generative process. The causal diagram $\mathcal{G}$ over $\mathbf{V}, \mathbf{L}$ induced by $\mathcal{M}_0^*$ at the generative level is shown in Fig. S19(a). The distribution $P(We, Ag, Um, L_2, L_3, ...)$ induced by $\mathcal{M}^*$ is displayed in Fig. S18 (only one unlabeled factor $L_1(Um)$ is shown explicitly for simplicity). This distribution suggests that there is a positive correlation between rainy ($We = 1$) and young age ($Ag = 1$); a negative correlation between the umbrella ($Um = 1$) and young age ($Ag = 1$); a positive correlation between rainy $We = 1$ and umbrella $Um = 1$.

In the second stage of the generative process, all $\mathbf{V}$ and $\mathbf{L}$ are mixed by function $f_{\mathbf{I}}^*$ and generate the corresponding pixels. Some image samples $\{\mathbf{i}_1, \mathbf{i}_2\}$ from the observational distribution are shown in Fig. S19(c). The causal diagram $\mathcal{G}$ over the observed variables $\mathbf{V}, \mathbf{I}$ (projected from the whole diagram) is shown in Fig. S19(b). ∎

Equipped with ASCMs (Def. 1), our task to edit the concepts $\mathbf{X}$ in an original image $\mathbf{i}$ from $\mathbf{X} = \mathbf{x}$ to $\mathbf{X} = \mathbf{x}'$ can be formalized as querying an *Image counterfactual distribution (I-ctf)* $P^*(\mathbf{I}_{x'} = \mathbf{i}' \mid \mathbf{I} = \mathbf{i})$ induced by the true underlying model $\mathcal{M}^*$. In addition, ASCMs can formalize the counterfactual effect in editing between generative factors. Formally, given factual factors $\mathbf{W}_1 = \mathbf{w}_1$ ($\mathbf{W}_1 \subseteq \mathbf{V}$), the probability that factors $\mathbf{W}_2$ will be $\mathbf{w}_2$ after the edit $do(\mathbf{X} = \mathbf{x}')$ is formalized by a counterfactual quantity $P^*(\mathbf{W}_{2[\mathbf{X}=\mathbf{x}']} = \mathbf{w}_2 \mid \mathbf{w}_1)$ at the generative level $\mathcal{M}_0$. For example, the task "edit the original image $\mathbf{i}_1$ (shown in Fig. S19(c)) to a rainy day" can be written as a counterfactual distribution "what would the image be had the weather been rainy?", corresponding to the I-ctf distribution $P^*(\mathbf{I}_{We=1} \mid \mathbf{I} = \mathbf{i}_1)$. One may be interested in counterfactual probability over features "given an old lady without an umbrella on a sunny day, would the age of the person still be old and the umbrella be added?". This probability corresponds to $P^*(Ag_{We=1} = 1, Um_{We=1} = 1 \mid We = 0, Ag = 0, Um = 0)$, where $\mathbf{W}_1$ as $\{We, Ag, Um\}$ (weather, age, umbrella) and the counterfactual set $\mathbf{W}_2$ as $\{Ag, Um\}$.

The I-ctf query can be mapped back to the generative level by the following result.

**Lemma 2.** *Consider a true generative process described by ASCM $\mathcal{M}^*$. Then,*

$$\underbrace{P^{\mathcal{M}^*}(\mathbf{i}_{\mathbf{x}'}' \mid \mathbf{i})}_{\text{images}} = \underbrace{P^{\mathcal{M}^*}(\mathbf{v}_{\mathbf{x}'}', \mathbf{l}_{\mathbf{x}'}' \mid \mathbf{v}, \mathbf{l})}_{\text{generative factors}} \tag{103}$$

*where $\mathbf{v}, \mathbf{l} = f_{\mathbf{I}}^{-1}(\mathbf{i})$.* ∎

To illustrate, Lemma. 2 states that an I-ctf query is equivalent to asking "What would all generative factors be had a concept change to $\mathbf{x}'$?". For example, $P^*(\mathbf{I}_{We=1} \mid \mathbf{I} = \mathbf{i}_2)$ in Ex. 2 is equivalent to asking what age, umbrella, shadow, the pose of the lady, and other unlabeled factors would be had the weather changed to a raining day. However, it is reasonable that users may only care about counterfactual reasoning about a subset of the generative factors. For example, a user may specifically care what age and umbrella would be had the weather changed to rainy ($do(We = 1)$) given an old lady without an umbrella on a sunny day, namely a counterfactual distribution $P^{\mathcal{M}^*}(Ag_{We=1}, Um_{We=1} \mid We = 0, Ag = 0, Um = 0)$.

### E.2 Proxy Models and Latent Space

As illustrated in Sec. 2, editing a given image by alternating latent vectors ($do(\mathbf{T} = \mathbf{t}'), \mathbf{T} \subseteq \mathbf{Z}$) in proxy models can be modeled as a counterfactual query $P^{\widehat{\mathcal{M}}}(\mathbf{I}_{\mathbf{T}=\mathbf{t}'} \mid \mathbf{i})$.

**Example 3** (continued Ex. 2). Consider the ASCM image generation process $\mathcal{M}^*$ illustrated in Ex. 2. Consider an SCM,

$$\widehat{\mathcal{M}} = \langle \widehat{\mathbf{U}} = \{\widehat{U}_1, \widehat{U}_2, \widehat{U}_3, \widehat{U}_4, \widehat{\mathbf{U}}_{\mathbf{L}}\}, \mathbf{Z} = \{Z_1, Z_2, Z_3, Z_4, Z_5, Z_6, ...\}, \widehat{\mathcal{F}}, P^*(\widehat{\mathbf{U}})\rangle, \tag{104}$$

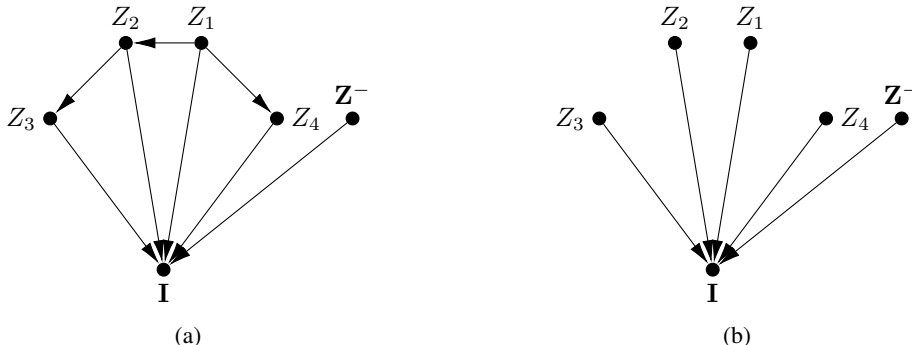

Figure S20: (a) Causal proxy model. ; (b) A standard LS.

where the mechanism,

$$\widehat{\mathcal{F}} = \begin{cases} Z_1 \leftarrow \widehat{U}_1 \\ Z_2 \leftarrow \widehat{U}_2 \oplus Z_1 \\ Z_3 \leftarrow (\neg Z_2) \oplus \widehat{U}_3, Z_4 \leftarrow Z_1 \oplus \widehat{U}_4 \\ Z_5 \leftarrow \widehat{U}_{L_1}, Z_6 \leftarrow \widehat{U}_{L_2} \\ \dots \\ \mathbf{I} \leftarrow \widehat{f}_{\mathbf{I}}(\mathbf{Z}) = f_{\mathbf{I}}^*(Z_1, Z_2, Z_3 \wedge Z_4, f_{L_2}^*(Z_2, Z_5), f_{L_3}^*(Z_6), \dots) \end{cases} \tag{105}$$

and $\widehat{\mathbf{U}}$ follows the same distribution as $\mathbf{U}$ of $\mathcal{M}^*$, namely, $P(\widehat{\mathbf{U}}) = P^*(\mathbf{U})$. It is verifiable that $\widehat{\mathcal{M}}$ defines the same mapping from $\mathbf{U}$ to $\mathbf{I}$ as the true $\mathcal{M}^*$, thus $\widehat{\mathcal{M}}$ induces the same observational image distribution, i.e., $P^{\mathcal{M}^*}(\mathbf{I}) = P^{\widehat{\mathcal{M}}}(\mathbf{I})$. Obtaining synthetic generative models is a process to get such an SCM by training models on collected data. Since $P^{\mathcal{M}^*}(\mathbf{I})$ approximates the true $P^{\widehat{\mathcal{M}}}(\mathbf{I})$, one can sample from $\mathbf{U}$ to generate latent factors $\mathbf{Z}$, such as latent representations or intermediate variables (e.g., hidden units in a neural network), and subsequently generate the corresponding image samples.

Then, editing an image $\mathbf{i}$ through $\mathbf{Z}$ can be interpreted as evaluating a counterfactual distribution, $P^{\widehat{\mathcal{M}}}(\mathbf{I}_{\mathbf{T}=\mathbf{t}'} \mid \mathbf{i})$, where $\mathbf{T} \subseteq \mathbf{Z}$. For example, one might intervene the latent representation $Z_1$ as value $z_1'$ to change the weather to a rainy day, and the counterfactual distribution is $P^{\widehat{\mathcal{M}}}(\mathbf{I}_{Z_1=z_1'} \mid \mathbf{i})$. ∎

### E.3 Feature Counterfactual Query (Def. 2)

The following example gives illustration about the counterfactuals over features when editing images through a proxy model $\widehat{\mathcal{M}}$

**Example 4** (continued Example. 3)**.** Consider the proxy model $\widehat{\mathcal{M}}$ in Example 3. Suppose that the F-ctf query interested is the probability that "given an old lady without an umbrella on a sunny day, the age of the person would still be old and the umbrella would be added if $Z_1 = 1$". According to Def. 2, the factual set $\mathbf{W}_1$ is chosen as $\{We, Ag, Um\}$ (weather, age, umbrella) and the counterfactual set $\mathbf{W}_2$ is chosen as $\{Ag, Um\}$ (age, umbrella). Then the F-ctf query is $P^{\widehat{\mathcal{M}}}(Ag_{Z_1=1}, Um_{Z_1=1} \mid We = 0, Ag = 0, Um = 0)$. Since $\{Ag, Um, We\}$ are not endogenous variables in $\widehat{\mathcal{M}}$, the F-ctf query cannot be calculated directly through $\widehat{\mathcal{M}}$ and should be computed from Def. 2. To illustrate, the denominator of Eq. 2 evaluates the factual part: the probability of generated images describing "an old lady without an umbrella in a sunny day", which is

$$\int_{\mathbf{i}_1 \in \mathcal{X}_{\mathbf{I}}} \mathbf{1}\left[h_{We}^*(\mathbf{i}) = 0, h_{Ag}^*(\mathbf{i}) = 0, h_{Um}^*(\mathbf{i}) = 0\right] dP(\mathbf{i}_1) = P(Z_1 = 0, Z_2 = 0, (\neg Z_3) \wedge Z_4 = 0). \tag{106}$$

The numerator evaluates the counterfactual part, integrating over counterfactual worlds $P(\mathbf{i}, \mathbf{i}'_{[\mathbf{T}=\mathbf{t}']})$ such that $\mathbf{i}$ describing "an old lady without an umbrella in a sunny day" and $\mathbf{i}'$ describing "an old lady

with an umbrella in a rainy day".

$$\int_{\mathbf{i},\mathbf{i}'\in\mathcal{X}_\mathbf{I}} \mathbf{1}\left[h^*_{We}(\mathbf{i})=0, h^*_{Ag}(\mathbf{i})=0, h^*_{Um}(\mathbf{i})=0, h^*_{Ag}(\mathbf{i}')=0, h^*_{We}(\mathbf{i}')=1, h^*_{Um}(\mathbf{i}')=1\right] dP(\mathbf{i}, \mathbf{i}'_{[Z_1=1]}).$$
$$= P(Z_1=0, Z_2=0, (\neg Z_3)\wedge Z_4=0, Z_{2[Z_1=1]}=0, (Z_3\wedge Z_4)_{[Z_1=1]}=1)$$
(107)

Then we have

$$P^{\widehat{\mathcal{M}}}(Ag_{Z_1=1}, Um_{Z_1=1} \mid We=0, Ag=0, Um=0)$$
$$=P(Z_{2[Z_1=1]}=0, (Z_3\wedge Z_4)_{[Z_1=1]}=1 \mid Z_1=0, Z_2=0, (\neg Z_3)\wedge Z_4=0)=0$$
(108)

This quantity implies that applying intervention $do(Z_1=1)$ on proxy model $\widehat{\mathcal{M}}$ for editing would never generate an old woman with an umbrella. ∎

Next, we present an example to illustrate how to evaluate an F-ctf query is a valid estimation for the ground truth using ctf-consistency Def. 3, even when the query is not identifiable.

**Example 5** (continued Example 4). Consider the true ASCM introduced in Example 2 in $\mathcal{M}^*$, the proxy model $\widehat{\mathcal{M}}$ with the F-ctf query $P^{\widehat{\mathcal{M}}}(Ag_{Z_1=1}, Um_{Z_1=1} \mid We=0, Ag=0, Um=0)$ in Example 4. Ctf-consistency provides a way to evaluate whether the F-ctf query is a ctf-consistent query for the query $P^*(Ag_{We=1}=1, Um_{We=1}=1 \mid We=0, Ag=0, Um=0)$. According to mechanism $\mathcal{F}^*$ (Eq. 102) and $P^*(\mathbf{U})$,

$$P^*(Ag_{We=1}=1, Um_{We=1}=1 \mid We=0, Ag=0, Um=0) = \frac{P((\neg U_3)\wedge(\neg U_4)=0)}{P((\neg U_3)\wedge U_4=0)} = 0.78$$
(109)

This is the ground truth and not immediately obtainable. On the other hand, the bound $[l,r]$ of this query given $P(\mathbf{V})$ and $\mathcal{G}_{\mathbf{V},\mathbf{L}}$ can be derived as (see [41, Thm. 9.2.12]):

$$l = \max\{0, 1 - \frac{P(Um=0 \mid We=1, Ag=0)}{P(Um=0=0 \mid We=0, Ag=0)}\} = 0.70$$
$$r = \min\{1, \frac{P(Um=1 \mid We=1, Ag=0)}{P(Um=0 \mid We=0, Ag=0)}\} = 0.78$$
(110)

Def. 3 is saying that any value within the bound $[0.70, 0.78]$ is regarded as a counterfactual consistent estimation for the ground truth $P^*(Ag_{We=1}=1, Um_{We=1}=1 \mid We=0, Ag=0, Um=0)$ and any value out of this bound will be regarded as invalid from a causal stand point. Specifically, since the F-ctf query $P^{\widehat{\mathcal{M}}}(Ag_{Z_1=1}, Um_{Z_1=1} \mid We=0, Ag=0, Um=0)$ induced by the proxy model is always 0 (Eq. 108, $\widehat{\mathcal{M}}$ is not a considered as a counterfactual consistent estimator for this query. ∎

### E.4 Counterfactually Editing principles - Thm. 1

We first apply Thm. 1 to the raining and umbrella setting introduce in Ex. 2.

**Example 6** (continued Ex. 2). Consider the ASCM introduced in Ex. 2 and the task of editing the weather in image $\mathbf{i}_1$ (describing an old lady standing in a sunny day without an umbrella, shown in Fig. S19(c)) to rainy. The target query is written as $P(\mathbf{i}'_{We=0} \mid \mathbf{i}_1)$, where $\mathbf{i}' \in \mathcal{X}_\mathbf{I}$. Following Thm. 1, we can have

$$P^*(\mathbf{i}_{\mathbf{x}'}=\mathbf{i}' \mid \mathbf{i}=\mathbf{i}) = \underbrace{\mathbf{1}[h^*_{We}(\mathbf{i}')=1]}_{\text{Intervention Consistency}} \cdot \underbrace{\mathbf{1}[h^*_{Ag}(\mathbf{i}')=0, L_{1[We=0]}=h^*_{L_1}(\mathbf{i}'), h^*_{L_2}(\mathbf{i}')=h^*_{L_2}(\mathbf{i}), ...]}_{\text{Non-descendants Invariance}}$$
$$\cdot \underbrace{P^*(Um_{We=0}=h^*_{Um}(\mathbf{i}'))}_{\text{Descendant Delta}}$$
(111)

The first term ascertains that the weather in the result $\mathbf{i}'$ must be indeed rainy. The second term says that non-descendants, such as age, should be invariant after the edit. The third says that the weather's descendants, such as the umbrella, should change following $P^*(Um_{We=0} \mid \mathbf{v}, \mathbf{l})$. ∎

We then present a concrete example to clarify the notion of the "amount of change"—illustrating why a factor may change during editing, yet still fail to reflect a valid counterfactual.

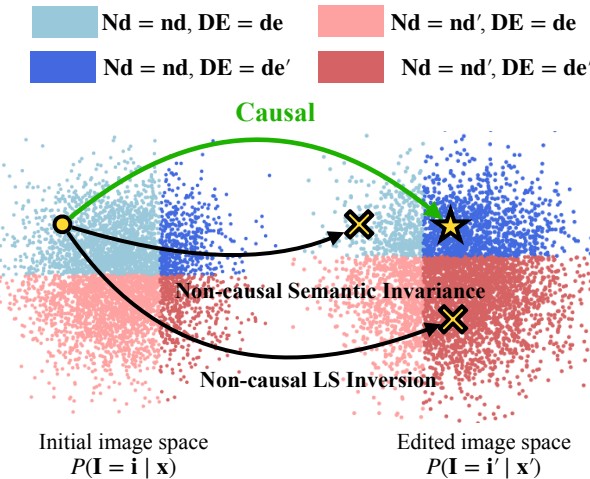

Figure S21: The comparison of non-causal editing methods, and the causal editing methods.

**Example 7** (continued Ex. 4 and 5). Consider the true ASCM $\mathcal{M}^*$ introduced in Ex. 2 and the image editing task "change the weather to rainy", which corresponds to I-ctf query $P^*(\mathbf{I}_{We=1} \mid \mathbf{i}_1)$, where $\mathbf{i}_1$ is shown in Fig. S19(c). Since $Z_1$ exactly represents $We$, one may use $P^{\mathbf{Z}^{\mathrm{LS}}}(\mathbf{I}_{Z_1=1} \mid \mathbf{i}_1)$ to estimate $P(\mathbf{I}_{We=1} \mid \mathbf{i}_1)$.

Consider an interesting probability that "given an old lady without an umbrella on a sunny day, an umbrella would be added, if the weather changed to rainy", which corresponds to $Q = P^*(Um_{We=1} \mid We = 0, Ag = 0, Um = 0)$. According to Def. 3, an estimate is ctf consistent with $Q$ if the estimation is within the bound $[0.70, 0.78]$ ( Ex.. 5).

Consider the proxy model $\widehat{\mathcal{M}}$ introduced in Ex. 3. The F-ctf query $P^{\widehat{\mathcal{M}}}(Um_{Z_1=1} \mid We = 0, Ag = 0, Um = 0)$ induced by $\widehat{\mathcal{M}}$ can be calculated as:

$$
\begin{aligned}
&P^{\widehat{\mathcal{M}}}(Um_{Z_1=1} \mid We = 0, Ag = 0, Um = 0) \\
&= P(Z_3 \wedge Z_4 = 1 \mid Z_1 = 0, Z_2 = 0, (\neg Z_3) \wedge Z_4 = 0) = 0.02
\end{aligned}
\tag{112}
$$

Thus, the umbrella would be raised with probability 0.02 due to the statistical correlation between $Um$ and $\{Ag, We\}$. However, the umbrella would be raised at least 0.70 to be ctf-consistent. In other words, naively using the correlation between the intervened feature $Um$ and the descendant $Um$, the amount of descendant change is not guaranteed. ∎

In addition to the discussion between change and invariance in Fig. 3. We provide another graph to illustrate the editing path on $\mathbf{I}$ between causal methods and non-causal methods shown in Fig. S21.

### E.5 Backdoor Disentangled Causal Latent Space - Def. 4 and Thm. 2

We first give an example of BD-CLS.

**Example 8** (continued Ex. 6). Suppose our goal is the edit task formalized as $P^*(\mathbf{i}'_{We=0} \mid \mathbf{i}_1)$ in Ex. 6 given $P(\mathbf{V}, \mathbf{I})$ and the causal diagram $\mathcal{G}_{\mathbf{V},\mathbf{I}}$ shown in Fig. S19(b). Note that $\{Ag\}$ serves as a backdoor set $\mathbf{B}$ in $\mathcal{G}_{\mathbf{V},\mathbf{I}}$ for the intervened variable set $\{We\}$.

Consider the proxy model introduced in Example 3. $\widehat{\mathcal{M}}$ is not an BD-CLS according to Def. 4. To witness, condition (1) is satisfied setting $We = Z_1$, $Ag = Z_2$ and $\mathbf{Z}^{\mathrm{BD-CLS}} = \mathbf{Z}^{\widehat{\mathcal{M}}} \backslash \{Z_1, Z_2\}$. To illustrate, according to Eq. 102 and 105:

$$
We = h^{*-1} \circ \widehat{f}_{\mathbf{I}}(\mathbf{Z}) = Z_1,
\tag{113}
$$

$$
Ag = h^{*-1} \circ \widehat{f}_{\mathbf{I}}(\mathbf{Z}) = Z_2,
\tag{114}
$$

$$
P(\mathbf{I} \mid We, Ag) = P(\mathbf{I} \mid Z_1, Z_2).
\tag{115}
$$

Condition (2) is satisfied since

$$L_2 = \tau_{L_2}(\mathbf{Z}) = f^*_{L_2}(Z_2, Z_5) \tag{116}$$

$$L_3 = \tau_{L_3}(\mathbf{Z}) = f^*_{L_3}(Z_6) \tag{117}$$

$$\tag{118}$$

which implies no $L_j$ is a function of $Z_1$ (which is $\mathbf{X}$ in this context). However, condition (3) is not satisfied since $\mathbf{B} = \{Z_2\}$ is a descendant of $\mathbf{X} = \{Z_1\}$ in this proxy SCM $\widehat{\mathcal{M}}$, which is not compatible with graph shown in Fig. 4.

Now we consider another SCM $\widehat{\mathcal{M}}^{\mathrm{BD}}$ with endogenous variables $\mathbf{Z}^{\mathrm{BD}}$ is exactly the same with $\widehat{\mathcal{M}}$ but different $f_{Z_2}$ as follows:

$$Z_2 \leftarrow \widehat{U}_1 \oplus \widehat{U}_2. \tag{119}$$

Then $\mathbf{Z} = \langle X = \{Z_1\}, B = \{Z_2\}, \mathbf{Z} = \{Z_3, Z_4, \dots\} \rangle$ satisfies the disentanglement requirement (similar to $\widehat{\mathcal{M}}$ illustrated above) and also satisfies the structural requirements since $Z_2$ is not a descendant of $Z_1$ but is now confounded with $Z_1$. ∎

Then, the next example shows how the estimation provided by the above BD-CLS satisfies the editing principles in Thm. 1.

**Example 9** (continued Example 8). Consider $\widehat{\mathcal{M}}^{\mathrm{BD}}$ introduced in Ex. 8. Notice that $\{Z_1\} = \mathbf{X} = \{We\}$ and $\{Z_2\} = \mathbf{B} = \{Ag\}$. Consider the image editing task "change the weather to rainy", which is formalized as the target I-ctf query $P^*(\mathbf{i}'_{we} \mid \mathbf{i}_1)$ in Ex. 8, where $\mathbf{i}_1$ contains the feature $\{We = we = 0, Ag = ag = 0, Um = um = 0\}$ and $\mathbf{i}'$ contains the feature $\{We = we' = 1, Ag = ag' = 0, Um = um' = 1\}$.

First, BD-CLS guarantees interventional consistency.

$$P^{\widehat{\mathcal{M}}^{\mathrm{BD}}}(we'_{We=1} \mid \mathbf{v}, \mathbf{l}) = \mathbf{1}[we' = we] = 1. \tag{120}$$

Since $Ag$ is a non-descendant of $We$, Thm. 2 suggests that

$$P^{\widehat{\mathcal{M}}^{\mathrm{BD}}}(ag'_{We=1} \mid \mathbf{v}, \mathbf{l}) = \mathbf{1}[ag' = ag] = 1. \tag{121}$$

In other words, BD-CLS guarantees that the feature Age is invariant after editing.

Next, consider the descendant $Um$. Thm 2 suggests that the estimation $P^{\widehat{\mathcal{M}}^{\mathrm{BD}}}(um'_{We=1} \mid we, ag, um)$ induced by BD-CLS is ctf-consistent with ground truth $P^*(um' \mid \mathbf{v}, \mathbf{l})$, which means $P^{\mathbf{Z}^{\mathrm{BD}}}(um'_{We=1} \mid we, ag, um)$ is in the optimal bound of $P^*(um'_{We=1} \mid \mathbf{v}, \mathbf{l})$. According to Example 5, this bound is $[0.70, 0.78]$.

### E.6 Difficulties to satisfies constraint Eq. 5 in Alg. 1

In Sec. 4.1, we argued that if one naively updateS learnable parameter $\theta_t$ in the direction shown in Eq. 7, it cannot guarantee the second constraint in the optimization problem (Eq. 5). The following example gives details about how this happens.

**Example 10.** Consider the underlying true ASCM $\mathcal{M}^*_{\mathrm{toy}}$ over labeled generative factors $X$ and unlabeled generative factors $Y$ and $R$, with the following generation mechanism:

$$\mathcal{F}^* = \begin{cases} X \leftarrow U_1 \\ Y \leftarrow X \wedge U_2 \\ R \leftarrow U_3 \\ I_1 \leftarrow X, I_2 \leftarrow Y, I_3 \leftarrow R \end{cases} \tag{122}$$

where $U_1, U_2, U_3$ are binary variables and $P(U_1 = 1) = 0.5, P(U_2 = 1) = 0.4, P(U_3 = 1) = 0.2$. To illustrate, $Y$ is the descendant of $X$ and $R$ is a non-descendant. The generative factors $\{X, Y, Z\}$ are mapped to a 3-bit image $\mathbf{I} = \{I_1, I_2, I_3\}$. The inverse mappings from image to feature are:

$$\begin{cases} X = h^*_X(\mathbf{I}) = I_1 \\ Y = h^*_Y(\mathbf{I}) = I_2 \\ R = h^*_R(\mathbf{I}) = I_3, \end{cases} \tag{123}$$

which implies that feature $X, Y$ and $R$ are shown in pixel $I_1, I_2, I_3$, respectively. Given an image $\{I_1 = 0, I_2 = 0, I_3 = 0\}$, the intervention $do(X = 1)$ is possible to change the factor $Y$ but will never change the non-descendant $R$.

Now consider an SD model $\widehat{\mathcal{M}}^{\text{SD}}_{\text{toy}}$ over prompt $X$ and $\mathbf{N} = \{N_1, N_2, N_3\}$, with mechanism

$$\mathcal{F}^{\text{SD}} = \begin{cases} X \leftarrow U_1 \\ N_1 \leftarrow U_{n_1}, N_2 \leftarrow U_{n_2}, N_3 \leftarrow U_{n_3}, N_4 \leftarrow U_{n_4} \\ I_1 \leftarrow X, I_2 \leftarrow X \wedge N_1, I_3 \leftarrow (\neg X \wedge N_2) \oplus (X \wedge N_3) \wedge N_4 \end{cases} \tag{124}$$

where all $\mathbf{U}$ are binary variables, and $P(U_1 = 1) = 0.5$, $P(U_{n_1} = 1) = 0.4$, $P(U_{n_2} = 1) = 0.2$, $P(U_{n_3} = 1) = 0.2$, $P(U_{n_4} = 1) = 0.6$. It is verified that $\widehat{\mathcal{M}}^{\text{SD}}$ induce the same conditional observational distribution $P(\mathbf{I} \mid \mathbf{X})$ as $\mathcal{M}^*_{\text{toy}}$. According to the mapping of the image $\mathbf{I}$ to features $Y, R$ in Eq. 123, the transformation from $\mathbf{X}$ and $\mathbf{N}$ to features are shown as,

$$\begin{cases} Y = X \wedge N_1 \\ R = (\neg X \wedge N_2) \oplus (X \wedge N_3) \wedge N_4, \end{cases} \tag{125}$$

This indicates that the non-descendant feature $R$ is entangled with $X$. The intervention on $X$ is likely to change $R$. In the process of searching for the transformation $\psi$ based on the given SD models, our aim is to alternate $\boldsymbol{\theta}$ to intervene on $X$ and also intervene on $I_3$ to keep $R$ the same. However, the mapping from image to features $Eq.$ 123 is unknown to SD model ($Y$ and $R$ are unobserved), thus the model itself cannot know to keep $I_3$ the same but allow $I_2$ to change. ∎

Prop. 2 shows that even the unobserved mapping $\tau_R$ (from $\{\mathbf{X}, \mathbf{B}, \mathbf{N}\}$ to non-descendant $R$) and $\tau_Y$ (from $\{\mathbf{X}, \mathbf{B}, \mathbf{N}\}$ to descendant $Y$) are functions with input $\mathbf{X}$, counterfactual behaviors can be used to distinguish $R$ and $Y$. See the next example for illustration.

**Example 11** (continued Example 10)**.** First, notice that an empty set $B$ will serve a backdoor set for $X$ and image $\mathbf{I}$ in the true $\mathcal{M}^*_{\text{toy}}$ and $\mathcal{M}^{\text{SD}}_{\text{toy}}$. Consider the given image is an 3-bit $\{I_1 = 0, I_2 = 0, I_3 = 0\}$ with feature $X = 0, Y = 0, R = 0$. Based on the mapping from Eq. 125, we have

$$P^{\widehat{\mathcal{M}}}(R_{X=1} = 1 \mid X = 0, R = 0, Y = 0) = 0.2 \tag{126}$$

$$P^{\widehat{\mathcal{M}}}(Y_{X=1} = 1 \mid X = 0, R = 0, Y = 0) = 0.4 \tag{127}$$

$$\tag{128}$$

This implies that when $X$ changes from 0 to 1, feature $R$ will change to 1 with probability 0.2. And $Y$ will change to 1 with probability 0.4. Consider another given image $\{I_1 = 1, I_2 = 0, I_3 = 0\}$ that only $I_1$ is different. Similarly,

$$P^{\widehat{\mathcal{M}}}(R_{X=0} = 1 \mid X = 1, R = 0, Y = 0) = 0.2 \tag{129}$$

$$P^{\widehat{\mathcal{M}}}(Y_{X=0} = 1 \mid X = 1, R = 0, Y = 0) = 0 \tag{130}$$

$$\tag{131}$$

This implies that when $X$ changes from 1 back to 0, feature $R$ will change to 1 with probability 0.2. And $Y$ will change to 1 with probability 0. This gives us the opportunity to reduce the change of $R$ but keep the change of $Y$ by comparing these two interventions. ∎

### E.7 Limitation

We discuss several limitations of our approach. First, since our method relies on pre-trained diffusion models, its performance is bound by the capabilities of these models. For example, if a model does not understand the input prompt, interventional consistency may not be achieved, and the edited image $\mathbf{i}$ may not reflect the intended features.

Second, while our theoretical results demonstrate the soundness of BD-CLS-Edit, the practical implementation, particularly Step 4, which searches for $\psi$, relies on the expressiveness of the candidate class $\mu_{\boldsymbol{\theta}}$ (Sect. 4.1). There is no guarantee that this class can disentangle all non-descendants $\mathbf{ND}$, especially in cases involving complex causal relationships with object moving, sizes changing, etc.

For example, editing an image of "a rabbit looking at a carrot in the forest" to "a rabbit looking at a wolf" fails to reflect the expected size differences: the wolf should appear much larger than the rabbit,

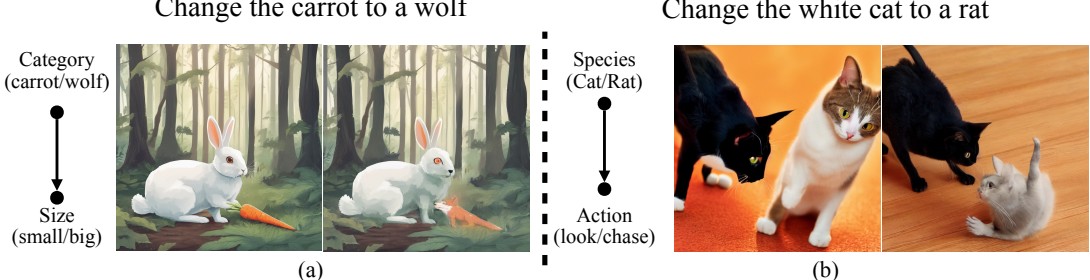

Figure S22: Failure cases of **BD-CLS-Edit**.

but this is not captured by the current edit (Fig.S22(a)). Similarly, replacing a white cat with a rat in an image of "a black cat and a white cat in a room" does not correctly reflect the causal effect of chasing behavior: the black cat should be chasing the rat, and both should be in motion, which is not the case in the result shown in Fig.S22(b). In other words, there are some more complex dynamics in the relationships of these objects that are not captured in a single image.

## E.8  Broader Impact

This paper aims to bridge the gap between causal image editing and the capabilities of large-scale pre-trained text-to-image models. Our work contributes to the growing need for more principled and reliable generative models by introducing a causal framework that respects the underlying structure of the data, rather than relying on correlation-driven editing strategies. A key motivation for this work is to challenge the common practice in current editing methods that prioritize semantic invariance, i.e., preserving as much of the original image as possible, while ignoring the causal effect of the edit on other semantics. This often leads to unrealistic results, particularly when editing should naturally induce downstream changes. By incorporating causal principles into the editing process, our method enables generative models to produce more realistic, consistent, especially in cases involving complex dependencies between visual features, which is beneficial for downstream tasks related to reliability, interpretability, and fairness generation.

## E.9  Safeguards

Similar to previous generative methods, our framework could be misused, for example, to manipulate visual content in ways that appear causally plausible but are misleading, such as in the spread of misinformation or the generation of unsafe content. Since our method builds on pre-trained models rather than creating a new one, existing safety mechanisms developed for diffusion-based models can be applied to enhance the safety of our approach [16, 35, 52, 30].

