# OpenReview forum: "Counterfactual Image Editing with Disentangled Causal Latent Space"
_NeurIPS.cc/2025/Conference — NeurIPS 2025 poster_

### Official Review · Reviewer_UHq7 · 2025-06-24

**Clarity:** 3
**Significance:** 3
**Originality:** 3
**Rating:** 5
**Confidence:** 3

**Summary:**

The paper introduces a framework for performing causally consistent image editing by leveraging pre-trained text-to-image diffusion models, without requiring further training or fine-tuning. The authors model image editing as the estimation of counterfactual queries. They propose a new class of Structural Causal Models (SCMs), called Backdoor Disentangled Causal Latent Space (BD-CLS), which satisfy key properties of causal image editing: intervention consistency, invariance of non-descendants, and controlled change of descendants. The authors also present an algorithm for obtaining a BD-CLS from a pre-trained diffusion model and sampling causally faithful counterfactual images. Lastly, the proposed method is assessed through experiments on both synthetic and real-world image data.

**Questions:**

- In Algorithm 1, Step 1, how do you identify the largest backdoor set?
- In Figures 6 and 9, which present qualitative comparisons, could you clarify which are the source and target prompts for performing the causal edits?
- What diffusion backbone model do you use for the ColorMNIST-Bars dataset? Is it a latent diffusion model or a pixel-based diffusion model?
- I believe the paper would benefit from some form of quantitative evaluation. If this is not feasible, it would still strengthen the work to include some qualitative results on additional known synthetic datasets (e.g., MorphoMNIST) or other real-world datasets such as CelebHQ, where causal graphs can be assumed as in [1] and [2].

**References:**

[1] Yang, Mengyue, et al. *CausalVAE: Disentangled representation learning via neural structural causal models.* Proceedings of the IEEE/CVF Conference on Computer Vision and Pattern Recognition, 2021.

[2] Melistas, Thomas, et al. *Benchmarking counterfactual image generation.* Advances in Neural Information Processing Systems, 37 (2024): 133207–133230.

**Ethical Concerns:**

["NO or VERY MINOR ethics concerns only"]

**Final Justification:**

The authors adequately addressed my concerns regarding the number of real-world images used in the evaluation of their method. That’s why I decided to raise my initial score.

**Limitations:**

The limitations are adequately described in the appendix.

**Paper Formatting Concerns:**

There are no formatting issues with the paper

**Quality:**

3

**Strengths And Weaknesses:**

**Strengths:**
- The paper tackles a significant gap in the literature on how to leverage text-to-image foundation models for performing causally consistent counterfactual image generation.
- The authors provide a solid theoretical framework regarding how to design a BD-CLS SCM. They clearly justify why the proposed BD-CLS achieves the desired properties for faithful causal editing.
- The proposed algorithm 1 for extracting a BD-CLS from a pre-trained diffusion model offers a compelling, causally-aware alternative to existing diffusion-based image editing methods, which may not respect assumed causal relationships when edits are performed.
- The algorithm 1 is efficient, as it does not require updating the diffusion model parameters.


**Weaknesses:**

- The proposed method's qualitative evaluation lacks depth.
  - More qualitative comparisons should have been included in the main paper.
  - The authors qualitatively compare their method with only two baseline image editing techniques.
- The paper lacks a thorough quantitative evaluation.
  - The authors do not include tables with quantitative results or use metrics that are commonly applied in counterfactual image generation [1, 2, 3].
- For real-world images, the authors do not evaluate their approach on a standard benchmark dataset (e.g., the datasets used by compared baselines such as DDPM inversion [4], or alternatives datasets with text annotations like Multi-Modal-CelebA-HQ [5]).
- Although the paper is well-structured, in some parts the notation becomes complex and difficult to follow.

**References:**

[1] Monteiro, Miguel, et al. *Measuring axiomatic soundness of counterfactual image models.* arXiv preprint arXiv:2303.01274, 2023.
[2] Ribeiro, Fabio De Sousa, et al. *High fidelity image counterfactuals with probabilistic causal models.* arXiv preprint arXiv:2306.15764, 2023.
[3] Melistas, Thomas, et al. *Benchmarking counterfactual image generation.* Advances in Neural Information Processing Systems, 37 (2024): 133207–133230.
[4] Huberman-Spiegelglas, Inbar, Vladimir Kulikov, and Tomer Michaeli. *An edit-friendly DDPM noise space: Inversion and manipulations.* Proceedings of the IEEE/CVF Conference on Computer Vision and Pattern Recognition, 2024.
[5] Xia, Weihao, et al. *TediGAN: Text-guided diverse face image generation and manipulation.* Proceedings of the IEEE/CVF Conference on Computer Vision and Pattern Recognition, 2021.

---

> ### Author Rebuttal · Authors · 2025-07-31
>
> Thank you for recognizing the motivation behind our work and for finding its theoretical and practical contributions compelling. We address your concerns in the sequel.
>
> > ***W1. The proposed method's qualitative evaluation lacks depth.***
>
> We appreciate this suggestion and agree that more qualitative comparisons in the main paper would improve clarity. In the revision, we will move additional examples from the appendix (Figs. S12, S13, S14) into the main body.
> We also add a new baseline, **DDIM Inversion**, a popular method for diffusion-based editing. While DDIM Inversion is often strong in perceptual fidelity, it fails to ensure non-descendant invariance (the second principle in Thm. 1, p. 4). For example, changing weather alters the garden layout  (Fig. 9(d)); changing season affects the person’s size  (Fig. 9(e)); Changing scene causes layout shifts (Fig. 9f). We will include these qualitative comparisons in the next revision.
>
> *[6] Mokady, Ron, et al. "Null-text inversion for editing real images using guided diffusion models."*
>
> > ***W2. The paper lacks a thorough quantitative evaluation.***
>
> We appreciate the suggestions of metrics mentioned in [1-3]. However, we found that some metrics are potentially misleading in the context of counterfactual editing; we elaborate a bit on these.
>
> We first highlight the counterfactual editing principles proposed by the paper. Our evaluation framework is theoretically grounded in Section 3 (lines 159–184): (1) **Interventional consistency** – the edited variable must match its intervened value; (2) **Non-descendant invariance** – non-descendants of \mathbf{X} should remain unchanged; (3) **Descendant change** – descendants should change according to the counterfactual distribution $P(\mathbf{De}_{\mathbf{x}’} \mid \mathbf{v, l})$.
> Satisfying these three principles is essential for faithfully modeling the causal effect from $X$ to the other variables, which leads to more causally-realistic image edits. However, metrics in [1-3] are not aligned with our goal.
>
> **LPIPS[3], minimality[3].**
> LPIPS measures perceptual similarity between source and edited images, with lower scores indicating better content preservation. However, this can conflict with the descendant change principle (lines 180–184). For example, in Fig. 9(a, d), editing “weather” should causally introduce an umbrella. Semantic-invariance-based editing that achieves high LPIPS may suppress such changes, producing perceptually similar but causally inaccurate images.
>
> We add evaluation of baselines and BD-CLS-Edit through tasks in large-scale text-to-image settings. DDIM and DDPM Inversion alter non-descendants, resulting in higher LPIPS. BD-CLS-Edit correctly modifies descendants, leading to slightly higher LPIPS than DDS, but this reflects faithful causal editing, not inaccuracy. We will add this clarification in the next revision.
>
> DDIM Inversion| DDPM Inversion | DDS | BD-CLS-Edit |
> |-----------------|-----------------|-----------------|-----------------|
> | 0.32 $\\pm$ 0.09| 0.25 $\\pm$ 0.08     | 0.13 $\\pm$ 0.03    | 0.16 $\\pm$ 0.03     |
>
> **Effectiveness [1-3]**. This metric aligns with our principle of interventional consistency. We add this evaluation to this rebuttal later.
>
> **Composition [1-3]**. This metric evaluates similarity when the intervention matches the factual value $\mathbf{x}’ = \mathbf{x} $ by measuring P(i’_{\mathbf{x}} \mid \mathbf{i}). However, our focus is on counterfactual edits, where $\mathbf{x}’ \ne \mathbf{x}$. Therefore, this metric is orthogonal to our goal.
>
> **Reversibility [1-2]**. This metric is saying that the intervention should be cycle-consistent and relies on the assumption that there exists a function $g$ such that endogenous variables  $\mathbf{u} = g(\mathbf{i}, \mathbf{x})$. However, this assumption does not hold in our non-parametric underlying SCM. Moreover, as discussed in [3], cycle consistency is not generally guaranteed in causal models and may lead to incorrect assumptions about editability.
>
> **To quantitatively validate the ability of BD-CLS to edit images counterfactually, we report estimation results of F-ctf queries as proposed in [7]**. This metric directly evaluates whether the edited outcomes align with the expected counterfactual effects (Fig. 8 and Fig. S7). For instance, in Fig. 8, we measure the probability that the bar becomes thicker after do(D=7). Our method estimates this probability as 0.73, consistent with the theoretical range [0.73, 0.82], reflecting a valid causal effect from “digit → barwidth.” The table version of Fig.8 is as follows.
>
> | Gound truth       | lower bound       | upper bound       | BD-CLS       | CDiffusion       | CGN       |  CGN       |
> |----------------|----------------|----------------|----------------|----------------|----------------|----------------|
> | 0.75    | 0.73    | 0.82    | 0.73 $\pm$ 0.02    | 0.40 $\pm$ 0.02    | 0.0    | 0.75 $\pm$ 0.02    |
>
> We report two additional F-ctf evaluations to demonstrate our method’s ability to edit counterfactually. First, we assess the probability that digit “7” appears in the edited image after the intervention do(D = 7). The ground-truth value is 1, reflecting interventional consistency, similar to effectiveness metrics in [1–3]. Results show that all methods satisfy this property.
>
> | Gound truth       | lower bound       | upper bound       | BD-CLS       | CDiffusion       | CGN       |  CGN       |
> |----------------|----------------|----------------|----------------|----------------|----------------|----------------|
> | 1.000    | 1.000    | 1.000    | 0.980 $\pm$ 0.004    | 0.978 $\pm$ 0.004    | 0.995 $\pm$ 0.001   | 0.988 $\pm$ 0.003    |
>
> Second, we evaluate the probability that the bar color becomes green after the digit changes. This quantity reflects non-descendant invariance. The ground truth should be 0 (line 318-328). The results suggest that BD-CLS is capable of keeping non-descendants invariant after editing (as expected).
>
> | Gound truth       | lower bound       | upper bound       | BD-CLS       | CDiffusion       | CGN       |  CGN       |
> |----------------|----------------|----------------|----------------|----------------|----------------|----------------|
> | 0.0    | 0.0    | 0.0    | 0.0    | 0.15 $\pm$ 0.04    | 0.0  | 0.0    |
>
> We conduct this F-ctf quantitative analysis only on the ColorMNIST datasets because we have full control over the generation process here, allowing us to evaluate the ground truth (and corresponding causal bound) and compare the proposed and baseline methods. In real-world dataset, the ground truth mechanisms for generating data are unknown and we provide a more qualitative type of discussion.
>
> [7] Pan, Yushu, and Elias Bareinboim. "Counterfactual Image Editing." ICML24.
>
> > ***W3. Experiments on other datasets***
>
> Thank you for the suggestion. We agree that additional results on other datasets can strengthen our work.  We will include results on CelebA to highlight causal effects on facial features in the next revision. As for the datasets in [4], while they are suitable for traditional image editing tasks, they lack clear and unified causal structures to demonstrate causal effects. For example, in tasks like editing a cat to a dog, the goal is often to change the animal while keeping the background unchanged and no other causal effects are considered.
>
> > ***W4** Although the paper is well-structured, in some parts the notation becomes complex and difficult to follow.***
>
> Thank you for pointing this out. We have revised the preliminaries for clarity and moved illustrative examples from the appendix into the main paper to better explain the notations used in the theorems.
>
> > ***Q1. In Algorithm 1, Step 1, how do you identify the largest backdoor set?***
>
> By “largest backdoor set,” we mean a maximal backdoor set—one that blocks all backdoor paths from X to the image I and cannot be extended further without violating the criterion. We compute such sets using a polynomial-delay algorithm [8].
>
> While BD-CLS-Edit does not require the backdoor set to be maximal (any valid set suffices theoretically), we use the maximal set in practice to better preserve non-descendant invariance, ensuring that more observed non-descendant variables in B remain unchanged during editing.
>
> *[8] Van der Zander, Benito, Maciej Liśkiewicz, and Johannes Textor. "Efficiently finding conditional instruments for causal inference."*
>
> > ***Q2. In Figures 6 and 9, which present qualitative comparisons, could you clarify which are the source and target prompts for performing the causal edits?***
>
> In Fig. 6 (BarMNIST), we use explicit labels for \mathbf{X} and \mathbf{B}, not text prompts. To illustrate, the task “edit a red ‘1’ with a thin red bar to digit “7”, we use the source labels {D=1, DC=0, BC=0} and target labels {D=7, DC=0, BC=0}.
>
> In Fig. 9 (text-to-image tasks), we use prompts such as:
>
> **Task: Weather edit (Fig. 9a,d; Fig. S10–S11):**
>
> Source prompt: “an old/young woman standing in a garden/street on a sunny day".
>
> Target prompt: “an old/young woman standing in a garden/street on a rainy day".
>
> BD-CLS-Edit edits umbrella/shadow/pose appropriately, unlike baselines.
>
> **Task: Season edit (Fig. 9b,e; Fig. S12):**
>
> Source prompt: “a person in the forest during the summer".
>
> Target prompt: “a person in the forest during the fall".
>
> BD-CLS-Edit correctly adjusts unprompted features like clothing and the shape of the person.
>
> **Task: Scene edit (Fig. 9c,f; Fig. S13):**
>
> Source prompt: “a person standing in the grocery store".
>
> Target prompt: “a person standing in the garden".
>
> BD-CLS-Edit provides causal guarantees for unprompted features such as person details and the grocery bag.
>
> > ***Q3. What diffusion backbone model do you use for the ColorMNIST-Bars dataset? Is it a latent diffusion model or a pixel-based diffusion model?***
>
> It is a pixel-based diffusion model trained to map observed generative factors to image pixels directly.

---

> > ### Comment · Reviewer_UHq7 · 2025-08-02
> >
> > I appreciate the authors’ detailed explanation addressing my questions and concerns. Nevertheless, I am inclined to uphold my initial rating, as the work still lacks a thorough quantitative and qualitative assessment on real-world datasets, such as CelebA for pixel-based diffusion models and Multi-Modal-CelebA-HQ for text-to-image diffusion models, where plausible causal graphs for human faces can be assumed as in [1].
> >
> > [1] Yang, Mengyue, et al. CausalVAE: Disentangled representation learning via neural structural causal models. Proceedings of the IEEE/CVF Conference on Computer Vision and Pattern Recognition, 2021.

---

> > > ### Author Response · Authors · 2025-08-04
> > >
> > > Thank you for taking the time to review our rebuttal and for sharing the reference to CausalVAE, which we will cite in the next revision. We’re glad that our explanation addressed your earlier questions and concerns.
> > >
> > > We’d like to take the opportunity to briefly clarify the role of real-world data in our evaluation. The primary goal of our work is to leverage pre-trained generative models for studying image editing under causal invariances in realistic settings. In particular, our text-to-image experiments use Stable Diffusion, which is trained on large-scale, real-world datasets including LAION-5B [1] and internal data [2].
> > >
> > > To the best of our knowledge, this is the first method to achieve counterfactual consistency with theoretical guarantees at this scale using pre-trained models, following the initial step by Pan & Bareinboim [3], whose method, while sound, operated with raw data and labels at a much smaller scale, including CelebAHQ.
> > >
> > > Our editing tasks reflect plausible real-world scenarios, with qualitative results shown in Fig. 9 and Figs. S10–S14 (pp. 9, 29–32), and quantitative results based on LPIPS included in the rebuttal. We believe these experiments provide strong evidence of the method’s relevance and applicability to real-world conditions.
> > >
> > > That said, we agree that additional experiments on CelebA or Multi-Modal-CelebA-HQ could further enrich the evaluation, and we plan to include such results in the next revision. However, due to the limited rebuttal window, and in accordance with the conference guidance (“Please do not expect the authors to submit updated results with images or any type of rich media”), we did not pursue such additions during this phase. We hope the paper will be evaluated based on its current contributions and supporting experiments, which we believe demonstrate performance superior to comparable state-of-the-art methods. We really appreciate the reviewer’s constructive feedback.
> > >
> > >
> > > *[1] Schuhmann, Christoph, et al. "Laion-5b: An open large-scale dataset for training next generation image-text models." Advances in neural information processing systems 35 (2022): 25278-25294.*
> > >
> > > *[2] Podell, Dustin, et al. "Sdxl: Improving latent diffusion models for high-resolution image synthesis." arXiv preprint arXiv:2307.01952 (2023).*
> > >
> > > *[3] Pan, Yushu, and Elias Bareinboim. "Counterfactual Image Editing." International Conference on Machine Learning. PMLR, 2024.*

---

> > > > ### Comment · Reviewer_UHq7 · 2025-08-04
> > > >
> > > > I thank the authors for their response. I would appreciate further clarification regarding the dataset: specifically, could the authors indicate the size of the real-world image dataset used in their evaluation?

---

> > > > > ### Author Response · Authors · 2025-08-04
> > > > >
> > > > > Thank you for your reply and for the opportunity to clarify!
> > > > >
> > > > > In our real-world experiments, we evaluate three causal editing tasks in the main paper—(1) editing weather, (2) editing season, and (3) editing scene—and one additional task in the appendix: (4) editing sports and surroundings. **Each task evaluates 120 input images, for a total of 480 images**. For comparison, the datasets referenced in [4] (in the review) include one with 212 images and another with 60, totaling 272.
> > > > >
> > > > > Regarding the pre-training of the Stable Diffusion model used in our experiments, the internal dataset details are not disclosed in the technical report [1]. The publicly available LAION-5B dataset [2], used during pre-training, contains 2.32 billion English image-text pairs. To the best of our knowledge, this is the first work to connect counterfactual inference with pre-trained diffusion models at this scale. We believe this is an important, practically meaningful step, as it enables evaluation across diverse, unconstrained real-world scenes rather than narrow or highly curated ones.
> > > > >
> > > > > We are happy to provide any additional counts or protocol details that would be useful.

---

> > > > > > ### Comment · Reviewer_UHq7 · 2025-08-04
> > > > > >
> > > > > > Thanks for the clarification regarding the size of the real-world evaluation dataset used to assess the method. A size of 120 images per causal editing task seems indeed adequate. I also believe it is essential to open-source this dataset, along with details about its source and the assumed causal graphs, as it could serve as a general benchmark for evaluating causal counterfactual image editing.
> > > > > >
> > > > > > With this clarification, I’m inclined to raise my rating.

---

> > > > > > > ### Author Response · Authors · 2025-08-04
> > > > > > >
> > > > > > > Thank you for your thoughtful feedback and for your constructive engagement throughout the reviewing process. We fully intend to open-source the dataset, including the images and the corresponding causal graphs for each task.
> > > > > > > More broadly, we are aligned with your view on the importance of developing benchmarks for counterfactual image editing grounded in real-world tasks, and we hope this work represents a first step toward advancing such resources for the broader Causal AI community.

---

### Official Review · Reviewer_VBDQ · 2025-07-01

**Clarity:** 3
**Significance:** 3
**Originality:** 3
**Rating:** 4
**Confidence:** 3

**Summary:**

This paper tackles the problem of counterfactual image editing—"What would this image look like if feature $X$ were changed from $x$ to $x'$?"—by marrying causal inference with pre-trained diffusion models. The key steps are:

1. Augmented Structural Causal Models (ASCM) formalize the true generative process, partitioning factors into labeled (V) and unlabeled (L) sets (Def.1, p.3).

2. Backdoor Disentangled Causal Latent Space (BD-CLS) is defined (Def.4, p.5) as a proxy SCM over $\{X, B, Z\}$ that (i) matches the true conditional distribution $P(I|X, B)$, (ii) disentangles non-descendant features from interventions, and (iii) respects the causal graph. Theorem 2 (p.6) proves that sampling from any BD-CLS yields intervention consistency, non-descendant invariance, and counterfactual consistency on descendants.

3. BD-CLS-Edit (Alg. 1, p.7) shows how to extract a BD-CLS from a pre-trained Stable Diffusion model without any retraining. By identifying a backdoor set B, inverting the image to noise $n$, and optimizing a prompt-mixing schedule $\theta_t$ via gradient updates (Eqs. 6–8), one can sample counterfactual images $i'\sim P_{\rm{BD-CLS}}(I_{X=x'}|i)$.

4. Experiments:

* Colored MNIST & Bars: BD-CLS yields the correct flip probabilities for descendant features (bar width) within theoretical bounds, while non-causal baselines (CDiffusion, CGN) violate invariance or descendant behaviour (Figs. 6–8).

* Text-to-Image Editing: On real-world prompts (weather → rainy, season → fall, scene → garden), BD-CLS-Edit preserves non-descendants (pose, layout) and induces realistic descendant changes (umbrella, clothing, removal of grocery bag), outperforming DDPM inversion and DDS (Fig. 9).

**Questions:**

1. Robustness: How sensitive is BD-CLS-Edit when the assumed backdoor set B omits a confounder or includes an extraneous node?

2. Inference efficiency: What is the average runtime and GPU/memory footprint for a single 512×512 edit?

3. Generalization: Can the method handle continuous interventions (e.g. brightness levels) or multiple simultaneous edits (changing both season and weather)?

4. Automated graph discovery: Is it possible to infer or verify the necessary causal graph and backdoor sets from data, reducing manual effort?

**Ethical Concerns:**

["NO or VERY MINOR ethics concerns only"]

**Final Justification:**

I believe the contributions and claims of the paper are novel and useful. I will keep my original rating and recommend acceptance.

**Limitations:**

1. Manual causal input: The need for a domain-expert causal graph and feature labels may hinder adoption in uncontrolled settings.

2 .Optimization overhead: Per-instance gradient searches for $\theta_t$ may not suit real-time applications.

3. Evaluation breadth: The paper does not demonstrate performance on large-scale or highly-diverse image datasets beyond a few case studies.

**Quality:**

3

**Strengths And Weaknesses:**

**Strengths**

The ASCM → BD-CLS framework gives formal guarantees on varying and invariant concepts under intervention without costly retraining. Moreover, only target feature $X$ and backdoor set $B$ need labels. All other factors remain label-free. Comprehensive quantitive and qualitative demonstrations via experiments convincingly demonstrate causal consistency where baselines fail.

**Weaknesses**
1. The approach presented assume a fully specified SCM over image factors $V$ and backdoor set $B$, which can be difficult to obtain in practice. The authors should consider ways of automatically extracting or at least validating the causal graph from data.

2. The setup assumes an exact and invertible mapping between image space and diffusion latents, which is not always available.

3. Each edits requires a gradient search over the per-timestep mixing weights which can be quite expensive. There is a lack of discussion in terms of the overhead for inference time.

4. The prompt-mixing schedule $\theta_t$ and its learning rates (Eqs. 6–8) require manual tuning. The paper lacks an extensive ablation on how robust the method is to these settings.

5. Validated mostly on synthetic datasets (Colored MNIST, Bars) and a handful of text-to-image edits. It remains unclear how well the approach scales to complex, real-world scenes with many entangled factors.

6.  Although only weak labels for $X$ and $B$ are needed, it still demands annotated examples of each intervention and backdoor factor. For richly structured images, obtaining such labels can be laborious.

7. Disentanglement emerges indirectly via $\theta_t$ optimization rather than explicit architectural constraints. There’s no guarantee that unrelated factors won’t bleed into each other under challenging edits.

---

> ### Author Rebuttal · Authors · 2025-07-31
>
> Thanks for the valuable suggestions and detailed comments. We appreciate the opportunity to clarify some of the concerns.
>
> > ***W1, L1. The approach presented assume a fully specified SCM over image factors V and backdoor set B, ...The authors should consider ways of automatically extracting or at least validating the causal graph from data.***
>
> **Fundamental point – our approach does not require a fully specified SCM.**
>
> We want to clarify that we do not require the fully specified SCM (a tuple $<U, V, F, P(U)>$), but we do need a causal diagram over V and I. The graph is substantially weaker than (in reality, a coarsening of) the SCM.
>
> **The necessity of diagrams.**
>
> Throughout the literature of causal inference, causal diagram assumptions are made not by choice, but out of necessity when inferring counterfactuals from observational data. To illustrate, any SCM induces families of different distributions related to seeing (observational), doing (interventional), and imagining (counterfactual), which together are known as Pearl Causal Hierarchy [1-2].  According to the Causal Hierarchy Theorem [2, Thm. 1], it is not possible to make a statement about a higher layer (counterfactuals) without more assumptions. We mention this since it also applies to our setting in CV – the graph assumption is made out of necessity and is somewhat expected in the broader causal inference literature. Please see [W2](https://openreview.net/forum?id=u2Lgi4NIe7&noteId=6AEm7uDxbd) for more details.
>
> **Why not learn the causal diagram from the data?**
>
> The complete true diagrams cannot be learned only from the observational data generally. Causal discovery (e.g., PC, FCI algorithms) can recover up to an equivalence class of diagrams. With higher-layer or multi-domain distributions, it is possible to prune the equivalence class further. However, they are often not available in many image editing tasks in practice, especially considering large-scale text-to-image generation. In contrast, causal diagrams can often be specified using human knowledge or common sense (e.g., rain causes umbrella use). LLMs also offer a venue for learning causal facts about the world, which depicts common-sense knowledge.
>
> *[1] Pearl, Judea, and Dana Mackenzie. The book of why: the new science of cause and effect. 2018.*
>
> *[2] Bareinboim, Elias, et al. "On Pearl’s hierarchy and the foundations of causal inference."*
>
> > ***W2. The setup assumes an exact and invertible mapping between image space and diffusion latents, which is not always available.***
>
> We want to clarify that the invertibility assumption in Def 1 refers to the underlying ground truth generative process $f^*_{\mathbf{I}}$, but not to the mapping between the image and the diffusion model’s latent space (e.g., $\mathbf{N}$ or $\mathbf{Z}$). This assumption is standard and widely adopted in the disentangled representation learning literature (line 1033 Remark 4, pp. 32), where it is used to model that true semantic factors can be recovered from images.
> We will make this distinction clearer in the revised version.
>
> > ***W3, Q2, L2. Inference efficiency: What is the average runtime and GPU/memory footprint for a single 512×512 edit?***
>
> Thanks for raising this issue. We agree that optimizing $\theta_t$ for all time steps in [1, 1000] for a stable diffusion training scheduler is costly.  In practice, we set the inference step to 200 for all text-to-image editing in this work (App. D.4, line 1011). The GPU cost of editing one 512*512 image is 28.9G, and the cost of editing a 1024*1024 image is 58.4G (results in the manuscript have a resolution 1024*1024). We add experiments about execution time for BD-CLS-Edit. We compare the excusion time of BD-CLS-Edit and DDPM inversion here for 512*512 edits for the request.
>
> | DDPM inversion 100 steps| DDPM inversion 200 steps | BD-CLS-Edit 100 steps | BD-CLS-Edit  200 steps |
> |-----------------|-----------------|-----------------|-----------------|
> | 0.36 $\\pm$ 0.01  min   | 0.79 $\\pm$ 0.02  min  | 4.88 $\\pm$ 0.06 min     | 9.87 $\\pm$ 0.08 min     |
>
> As shown, BD-CLS-Edit is slower than existing diffusion-based editors (~1 min). However, it remains significantly faster than retraining or fine-tuning full models, which can take hours to days. This was indeed one of the primary motivations of this work: how to leverage pre-existing, non-causal pre-trained models, which are already refined in terms of image quality, towards causally-aligned models. As the first to incorporate counterfactual editing into pre-trained text-to-image models, we prioritize accuracy (i.e., counterfactual consistency) over speed. We will add this discussion to the limitations section and leave inference acceleration to future work.
>
> > ***W4.  \theta and its learning rates (Eqs. 6–8) require manual tuning. The paper lacks an extensive ablation on how robust the method is to these settings.***
>
> Thanks for the helpful suggestions! The hyperparameters in our optimization procedure (Sec. 4.1, Alg. 3), including learning rate $\gamma$, the optimization steps $n_{\max}$, and the clipping value $\theta_{\max}$, require manual tuning. We have added ablation studies over $\gamma \in \{10^{-1}, 10^{-2}, 10^{-3}, 10^{-4}\}$, $n_{\max} \in \{2, 10, 20\}$, and $\theta_{\max} \in \{1.2, 1.5, 2.0\}$.
> We find that $\gamma = 10^{-2}$ and $n_{\max} = 10$ yield the best balance between performance and efficiency, and BD-CLS-Edit is relatively robust to  $\theta_{\max}$.
>
> > ***W5, L3. Datasets for Evaluation breadth***
>
> Thanks for the suggestions. We agree that including additional experiments can strengthen our work and will add results on CelebA in the next revision. However, we want to clarify that our real-world experiments (e.g., Fig. 9, Fig. S10–S14) do involve many entangled factors even though only 3–5 variables are explicitly shown. There are many implicit factors that are omitted from the graph but are still modeled. BD-CLS-Edit provides causal guarantees for these as well, for example, editing “weather” affects lightness (a descendant) while leaving pose and hair color (non-descendants) unchanged.
>
> > ***W6, L1. ..., it still demands annotated examples of each intervention and backdoor factor. For richly structured images, obtaining such labels can be laborious.***
>
> While our method does require labels for the intervention variable (X) and backdoor variables (B), we would like to emphasize that the labeling requirement is significantly lighter than in existing counterfactual editing methods. Specifically, BD-CLS-Edit only requires labels for X and B in the image to be edited. In contrast, prior methods typically require dense annotations of all generative factors across the entire training dataset, which is substantially more expensive.
>
> > ***W7. Disentanglement emerges indirectly via optimization rather than explicit architectural constraints...***
>
> Thanks for mentioning this. Whenever the edited concept sets X and backdoor sets B become complex and larger, BD-CLS-Edit performance will be limited by the given pre-trained model. We show failure cases in the limitation discussion (App. E.6). We believe encoding explicit architectural constraints in the model is also valuable, especially when concepts are more complex and the pre-trained model is not given. Encoding explicit architectural constraints for text-to-image generation could be an interesting research direction.
>
> > ***Q1.Robustness: How sensitive is BD-CLS-Edit when the assumed backdoor set B omits a confounder or includes an extraneous node?***
>
> We clarify that BD-CLS-Edit exhibits some robustness. First, note that the backdoor set B is not unique. For example, in Fig. 7 (p.8) for MNIST, both DC and {DC, BC} are valid, thus BC is redundant. If a mis-specified graph (e.g., both BC and DC are regarded as parents of D) lead to extra variables like BC in the backdoor set, BD-CLS still yields causal editing (see column 1; effect from D to BW remains within ground truth optimal [0.73, 0.82]). However, if a true confounder like DC is mistakenly excluded, e.g., treated as a descendant of D, editing may fail. The resulting bias violates non-descendant invariance and leads to incorrect causal estimates (column 2, not within bound).
>
> | BD-CLS (correct B but wrong graph)       | BD-CLS (wrong graph and wrong B)      |
> |----------------|----------------|
> | 0.73 $\pm$ 0.03    | 0.45 $\pm$ 0.06    |
>
> > ***Q3. Can the method handle continuous interventions (e.g. brightness levels) or multiple simultaneous edits (changing both season and weather)?***
>
> This is a good question, thanks. This work assumes discrete generative factors (not necessarily binary). For continuous variables, non-descendant invariance extends naturally, and they should be invariant. However, for descendants, in order to provide theoretical guarantees, the theory should be extended carefully, and this requires some non-trivial work since new machinery based on measure theory should be evoked.
> On the practical side, the algorithmic machinery proposed in the paper can be leveraged, and it should work in continuous settings.
>
> Additionally, BD-CLS-Edit is built to perform multiple simultaneous edits. The intervention set $\mathbf{X}$ can be a set including several variables. In experiments, we show multiple simultaneous edit results in Fig. S14 (App. E, pp. 32).
>
> > ***Q4. Automated graph discovery: Is it possible to infer or verify the causal graph and backdoor sets from data, reducing manual effort?***
>
> We agree that reducing reliance on manually specified graphs is important for practical use. However, as discussed earlier, with only observational data, the true causal graph is generally unidentifiable.
> Some methods aim to learn backdoor sets under assumptions, but they often require prior causal knowledge. Formalizing counterfactual editing with learned backdoor sets remains an open challenge, especially with unobserved factors, and it's an interesting research direction.

---

> > ### Comment · Reviewer_VBDQ · 2025-08-02
> >
> > I thank the authors for their thorough responses. I will however keep my rating as it currently stands.

---

> > > ### Author Response · Authors · 2025-08-04
> > >
> > > Thank you for taking the time to read our rebuttal. Since we believe we have addressed the concerns raised, we would be grateful for the opportunity to clarify any remaining issues, should anything still be unclear.

---

### Official Review · Reviewer_hSEs · 2025-07-01

**Clarity:** 2
**Significance:** 3
**Originality:** 3
**Rating:** 4
**Confidence:** 3

**Summary:**

This paper aims to study the counterfactual image editing problem with pretrained diffusion models. Current image editing works often ignore causal consistency, where some works change non-relevant parts and others fail to edit logically relevant parts. The authors claim there lack of scalability and existing methods rely on full supervision. On the contrary, the authors propose a theoretically grounded (based on Structural Causal Models) image editing method, which (1) considers the causal relationship between different factors; (2) is efficient without training the full model; (3) can work with weak supervision. The proposed method, BD-CLS-Edit, learns to optimize the linear combination parameters of conditional variables $c$ and $c'$ during the denoising process, under the constraints grounded by SCM. The method is demonstrated on Colored MNIST and Bars dataset and stable diffusion, where qualitative examples show better performance than baselines: LS inversion and DDS.

**Questions:**

See weakness. Major points are 1 and 2. It would be better if the paper could provide quantitative measurements and compare the proposed method with the baseline. It would be better if the evaluation dataset could practically cover diverse editing scenarios. It would also be better to provide robustness analysis on the scene graphs. Other points are minor and would be better clarified.

**Ethical Concerns:**

["NO or VERY MINOR ethics concerns only"]

**Final Justification:**

The authors' response has addressed most of my concerns, especially for the "assumed scene graph".

**Limitations:**

yes

**Quality:**

2

**Strengths And Weaknesses:**

Strengths
1. The method is highly theoretically grounded and novel.
2. The motivation is clear and interesting, counterfactual image editing.
3. The method is shown to be working on both toy models trained on MNIST and the stable diffusion model.
4. The actual implementation of BD-CLS-Edit by optimizing linear combination coefficients guided by the gradients is novel.

Weaknesses
1. Although some qualitative examples are shown, the paper shows no quantitative measurements (LPIPS, FID, human evaluation, etc) with large-scale evaluation datasets. This is a key limitation since the method's effectiveness may not be valid.
2. Although novel, the method requires a predefined causal graph as input. In practice, such a graph may not be easy to obtain. The paper does not provide a practical solution to get such graphs. Furthermore, in practice, the gained graphs may be incomplete or noisy, and it is worth analyzing whether the method is robust to such causal graphs.
3. The paper lacks some experimental details, such as how $\theta_i$ is regularized, whether it is learned per edited image, or can be shared across certain groups of images.
4. Current illustrations of the method are still constrained within simple causal relationships and do not seem scalable to more complex editing scenarios. For example, whether emotional relationships can be reflected as well?
5. Some writings are not clear, for example, the term LS inversion are used several times before defining it. The introduction of SCM related parts is also not clear. For example, the notion $\mathbf{P a}_{V_j}$ (Parents of $V_j$) is never introduced but used as granted.

---

> ### Author Rebuttal · Authors · 2025-07-31
>
> We thank the reviewer for recognizing the theoretical strength and novelty of our work. We feel that a few misreadings of our paper make the evaluation overly harsh, and hope you can reconsider the paper based on the clarifications provided below.
>
> > ***W1. the paper shows no quantitative measurements with large-scale evaluation datasets.***
>
> We appreciate the suggestions of using metrics such as **LPIPS, FID**. However, we found that these metrics are potentially misleading in the context of counterfactual editing; we elaborate a bit on these.
>
> We first highlight the counterfactual editing principles proposed by the paper. Our evaluation framework is theoretically grounded in Section 3 (lines 159–184): (1) **Interventional consistency** – the edited variable must match its intervened value; (2) **Non-descendant invariance** – non-descendants of $\mathbf{X}$ should remain unchanged; (3) **Descendant change** – descendants should change according to the counterfactual distribution $P(\mathbf{De}_{\mathbf{x}’} \mid \mathbf{v, l})$.
> Satisfying these three principles is essential for faithfully modeling the causal effect from X to the other variables, which leads to more causally-realistic image edits. However, LPIPS and FID are not aligned with the causal editing principles.
>
> **LPIPS.** This metric is typically used to assess the perceptual distance between the source and edited images. The lower distance implies better content preservation. However, this contradicts the descendant change principle described above (lines 180-184).
> LPIPS measures perceptual similarity between source and edited images, with lower scores indicating better content preservation. However, this can conflict with the descendant change principle (lines 180–184). For example, in Fig. 9(a, d), editing “weather” should causally introduce an umbrella. Semantic-invariance-based editing that achieves high LPIPS may suppress such changes, producing perceptually similar but causally inaccurate images.
>
> We evaluate LPIPS of baselines and BD-CLS-Edit through tasks in text-to-image settings. We add another baseline, DDIM inversion, which is widely used in Diffusion-based methods. DDIM and DDPM Inversion alter non-descendants, resulting in higher LPIPS. BD-CLS-Edit correctly modifies descendants, leading to slightly higher LPIPS than DDS, but this reflects faithful causal editing, not inaccuracy. We will add this clarification in the next revision.
>
> | DDIM Inversion| DDPM Inversion | DDS | BD-CLS-Edit |
> |-----------------|-----------------|-----------------|-----------------|
> | 0.32 $\\pm$ 0.09| 0.25 $\\pm$ 0.08     | 0.13 $\\pm$ 0.03    | 0.16 $\\pm$ 0.03     |
>
> **FID**. FID measures the distance between the distributions of the original observational dataset and the edited results of this dataset. However, in causal modeling, counterfactual images over the dataset follow the distribution $\sum_i P(i’_{x’} \mid i)P(i)$, which is theoretically different from the observational distribution $P(i)$. Therefore, a lower FID score does not indicate a more accurate counterfactual distribution and may misrepresent the quality of the generated edits.
>
> **To quantitatively validate the effectiveness of counterfactually editing provided by BD-CLS, we report estimation results of the F-ctf query as proposed in [1]**. This metric directly evaluates whether the edited outcomes align with the expected counterfactual effects (Fig. 8 and Fig. S7). For instance, in Fig. 8, we measure the probability that the bar becomes thicker after editing the digit. Our method estimates this probability as 0.73, consistent with the theoretical range [0.73, 0.82], reflecting a valid causal effect from “digit → barwidth.” The table version of Fig.8 is as follows.
>
> | Gound truth       | lower bound       | upper bound       | BD-CLS       | CDiffusion       | CGN       |  CGN       |
> |----------------|----------------|----------------|----------------|----------------|----------------|----------------|
> | 0.75    | 0.73    | 0.82    | 0.73 $\\pm$ 0.02    | 0.40 $\\pm$ 0.02    | 0.0    | 0.75 $\\pm$ 0.02    |
>
> [1] Pan, Yushu, and Elias Bareinboim. "Counterfactual Image Editing.".
>
> > ***W2. Although novel, the method requires a predefined causal graph as input. ... it is worth analyzing whether the method is robust to such causal graphs.***
>
> We appreciate raising this point. This concern touches on foundational principles in causal inference – why **assume a given causal graph?**
>
> Throughout the literature of causal inference, causal diagram assumptions are made not by choice, but out of necessity when inferring counterfactuals from observational data. To illustrate, any SCM induces families of different distributions related to the activities of seeing (observational), doing (interventional), and imagining (counterfactual), which together are known as Pearl Causal Hierarchy [2-3]. In this context, the passive collected images with labels are observational data, and the image editing query is a counterfactual quantity. According to the Causal Hierarchy Theorem [3, Thm. 1], distributions from the lower layers almost always underdetermine higher layers. In other words, in a measure-theoretic sense, it is not possible to make a statement about a higher layer (counterfactuals) without more assumptions; we refer to Sec. 1.3 in [3] for a formal discussion. We mention this since it also applies to our setting in computer vision – the graph assumption is made out of necessity and is somewhat expected in the broader causal inference literature.
>
> **Why not learn the causal diagram from the data?**
> The complete true underlying causal diagrams cannot be learned only from the observational data in general. Causal discovery methods (e.g., PC, FCI algorithms) can recover up to an equivalence class of diagrams. With higher layer (such as interventional) or multi-domain distributions, it is possible to prune the equivalence class further. However, these distributions are often not available in image editing tasks in practice, especially considering large-scale text-to-image generation. In contrast, the diagram itself can be more easily obtained from human knowledge in some common-sense tasks. For example, one knows that rain will cause the use of an umbrella. LLMs also offer a venue for learning causal facts about the world, which depicts common-sense knowledge.
>
> **Next, we elaborate on the sensitivity and robustness of the input graph and provide additional experimental results.**
> BD-CLS-Edit demonstrates robustness in many practical scenarios. According to Thm 2, the set $B$ is required to be a backdoor set in the true graph $G$. If a misspecified or incomplete graph $G’$ is used, BD-CLS still provides causal guarantees as long as the chosen backdoor set $B’$ remains valid in $G$.
>
> For example, consider the true $G$ in Fig. 7 (p. 8), where D is confounded with DC, and BC is a descendant of DC. Even if one mistakenly assumes both DC and BC are parents of D, DC plus BC still satisfies the backdoor criterion from D to image in both the incorrect and true graphs. Therefore, BD-CLS-Edit remains valid under this mis-specification and counterfactual editing is unaffected.
>
> To ground this, we practically evaluate F-ctf query $P(BW_{D=7}= 1 | D = 1, DC = 0, BC = 0, BW = 0)$ (same query in Fig.8) when the misspecified graph $G’$ is given.
> The results, as shown in the first column, suggest that the F-ctf estimation remains within the bounds, indicating that the counterfactual editing provided by BD-CLS is still effective and robust to the misspecified graph.
>
>
> | BD-CLS (correct B but wrong graph)       | BD-CLS (wrong graph and wrong B)      |
> |----------------|----------------|
> | 0.73 $\\pm$ 0.03    | 0.45 $\\pm$ 0.06    |
>
> If the backdoor set from the input graph is invalid in the true graph, causal editing may fail. For instance, if the misspecified graph wrongly treats DC as a descendant of D and excludes it from the backdoor set, an intervention on D may incorrectly influence DC, violating non-descendant invariance. This leads to biased estimation of D’s effect on BW, compromising counterfactual validity.
> To illustrate, we evaluate the same F-ctf query using this misspecified graph (results in column 2), and observe that the effect from D to BW is underestimated.
>
> **[2] Pearl, Judea, and Dana Mackenzie. The book of why: the new science of cause and effect. 2018.**
>
> **[3] Bareinboim, Elias, et al. "On Pearl’s hierarchy and the foundations of causal inference."**
>
> > ***W3. experimental details***
>
> During optimization, we apply a penalty when $\theta_t$ becomes too large to avoid unstable updates (Step 10 of Algorithm 3 on p. 24). This regularization, along with other algorithmic details and experimental parameters, is described in App. C and D. $\theta_t$ is optimized individually for each input image. We will make this design choice explicit in the next revision.
>
> > ***W4.***
>
> While our real-world results (e.g., Fig. 9, Fig. S10–S14) show only a few variables for clarity, many additional factors (e.g., pose, hair color, lightness) are implicitly modeled. BD-CLS-Edit still provides causal guarantees for these, e.g., editing “weather” changes lightness (a descendant) while keeping pose and hair color (non-descendants) unchanged. (see [Q1](https://openreview.net/forum?id=u2Lgi4NIe7&noteId=WttAWh3uHg) here for more details)
>
> We agree that handling richer, more abstract concepts (e.g., emotion) is challenging. As $X, B$ grows in size and complexity, performance may be limited by the pre-trained model (see App. E.6 limitation). Improving the causal guarantees when involving more complex graph structures and more variables could be an interesting future work.
>
>
> > ***W5.***
>
> Thanks for raising this. We add an illustration for LS inversion before using it and rephrase the preliminary for better notation illustration and explicitly add “the parents of $V_i$ are denoted as $Pa_{v_i}$ in SCMs.” into the manuscript.

---

> > ### Comment · Reviewer_hSEs · 2025-08-04
> > **Reply to Rebuttal**
> >
> > Thanks for the response, which addressed my questions. I would like to increase my score.

---

> > > ### Author Response · Authors · 2025-08-04
> > >
> > > Thank you for the opportunity to clarify these issues and elaborate on our work. We will review this thread again and use it to improve the manuscript.

---

### Official Review · Reviewer_NNpN · 2025-07-02

**Clarity:** 2
**Significance:** 3
**Originality:** 3
**Rating:** 5
**Confidence:** 2

**Summary:**

This paper introduces Backdoor Disentangled Causal Latent Space (BD-CLS), a method that enables edits to reflect causal effects without altering unrelated features, even under weak supervision. It proposes BD-CLS-Edit, an algorithm that extracts this structure from pre-trained diffusion models like Stable Diffusion without retraining. The approach ensures edits are causally consistent, scalable, and effective even when only partial labels are available. Results on synthetic and real-world tasks show improved causal fidelity compared to existing baselines.

**Questions:**

While results on real-world images are promising, most examples involve relatively small-scale causal graphs (3–5 variables). How does BD-CLS-Edit scale when editing in scenes with richer semantics?

**Ethical Concerns:**

["NO or VERY MINOR ethics concerns only"]

**Final Justification:**

Based on the overall quality of the submission and the authors’ clarification regarding the scale of the causal graph, in the rebuttal, which addressed my key comments, I’m inclined to recommend acceptance for this paper.

**Limitations:**

Yes, but only in the supplementary materials.

**Paper Formatting Concerns:**

No concerns.

**Quality:**

3

**Strengths And Weaknesses:**

The paper presents a solid theoretical framework for counterfactual image editing. It integrates well with pre-trained diffusion models and requires no retraining. The structure is clear however the paper relies heavily on the supplementary material which make reading harder. The work addresses a critical gap in generative modeling. The ability to edit images in a causally grounded way under weak supervision is valuable. The formulation is original and timely.

---

> ### Author Rebuttal · Authors · 2025-07-31
>
> We thank the reviewer for the positive evaluation of our work and the thoughtful comments. Below, we address your feedback and questions in detail.
>
> > ***W1. the paper relies heavily on the supplementary material which make reading harder.***
>
> Thanks for pointing this out. We have plenty of examples and experimental results to validate our theorem and algorithmic contributions. Due to space constraints, we had to move some of them to the supplementary material. If the paper is accepted, we will utilize the additional page to relocate some content from the supplementary material into the main paper, hopefully translating into improved readability.
>
> > ***Q1. While results on real-world images are promising, most examples involve relatively small-scale causal graphs (3–5 variables). How does BD-CLS-Edit scale when editing in scenes with richer semantics?***
>
> We appreciate this important question and would like to provide further elaboration.
>
> First, we note that even though our real-world experiments (e.g., Fig. 9, Fig. S10–S14, p. 9, 29-32) explicitly show only 3–5 variables in the demonstrated graph, **many additional variables are implicitly present in the causal graph, and BD-CLS-Edit also provides counterfactual guarantees for these implicit variables** (as long as they obey certain topological requirements as delineated by Thm. 2). This is an essential, compelling feature of the proposed framework.  To illustrate, generative factors are divided into two buckets (1) observed ones and (2) unobserved ones. Real-world images often involve numerous concepts and fine-grained details, many of which are either unlabeled or not explicitly described in the prompts. As a result, unobserved factors can involve a wide range of latent concepts.
> **We drew several unobserved factors in the graph for discussion. However, our theoretical results and algorithm are not limited to the specific unobserved factors shown; they remain valid for the other unobserved factors.**
>
> For concreteness, consider the weather-editing task (Fig. 9 (a)), where we explicitly draw variables such as weather, age, umbrella, shadow, and scene. However, other factors, such as the person’s pose, scene layout, hair color, and scene lightness, are also part of generative factors in images but are not drawn explicitly for the clean presentation.
>
> BD-CLS-Edit also gives causal guarantees for these variables. As shown in Fig. 9, edits to the “weather” do not affect the pose, scene layout, and hair color (non-descendants of weather), while descendant features such as the lightness changes appropriately. We will clarify this point to make the scope of the settings more explicit in the next revision.
>
> Second, we note that as the number of variables grows, the performance of BD-CLS-Edit may be increasingly constrained by the capacity of the pre-trained backbone model. Theoretically, BD-CLS-Edit assumes access to a pretrained model that can fit the conditional distribution $P(\mathbf{I} \mid \mathbf{x, b})$. In practice, modeling this distribution is more challenging as the size of the set $\{\mathbf{X}, \mathbf{B})$ increases, and the concepts become more complex. Improving the causal guarantees when involving more complex graph structures and more variables could be an interesting question for future work.
>
> In terms of the larger context and history, we note that, in a novel way, the contributions of this paper constitute an important step toward enabling large-scale generative models to exhibit causal capabilities and perform counterfactual reasoning. To the best of our knowledge, the theoretical contributions presented here tie these very different modalities together in a fundamental way. We hope this connection, now formally established, can be leveraged to build not only larger, but also more realistic models. In other words, we view this paper as a first step in initiating a conversation with the community about the necessary developments on the causal-generative front, including further scalability challenges, as noted by the reviewer.

---

### Decision · Program_Chairs · 2025-09-17

**Decision:**

Accept (poster)

**Comment:**

This paper proposes a counterfactual image editing method using a pre-trained diffusion model. Specifically, they are interested in causal consistency, i.e., only edit the logically relevant parts. All the reviewers are supportive of this paper. The main discussion points during the discussion period are the empirical results and many of the implementation details. The authors have successfully addressed the reviewers’ concerns. The AC does not see reasons to overrule the reviewer’s recommendation.